# Expansion microscopy of *C. elegans*

**Chih-Chieh (Jay) Yu**[1,2,3], **Nicholas C Barry**[2,3], **Asmamaw T Wassie**[1,3], **Anubhav Sinha**[2,3,4], **Abhishek Bhattacharya**[5], **Shoh Asano**[2†], **Chi Zhang**[2,3], **Fei Chen**[6], **Oliver Hobert**[5], **Miriam B Goodman**[7], **Gal Haspel**[8,9], **Edward S Boyden**[1,2,3,10,11]*

[1]Department of Biological Engineering, Massachusetts Institute of Technology, Cambridge, United States; [2]Media Lab, Massachusetts Institute of Technology, Cambridge, United States; [3]McGovern Institute, Massachusetts Institute of Technology, Cambridge, United States; [4]Division of Health Sciences and Technology, Massachusetts Institute of Technology, Cambridge, United States; [5]Department of Biological Sciences, Howard Hughes Medical Institute, Columbia University, New York, United States; [6]Broad Institute of MIT and Harvard, Cambridge, United States; [7]Department of Molecular and Cellular Physiology, Stanford University, Stanford, United States; [8]Federated Department of Biological Sciences, New Jersey Institute of Technology and Rutgers University-Newark, Newark, United States; [9]The Brain Research Institute, New Jersey Institute of Technology, Newark, United States; [10]Koch Institute, Massachusetts Institute of Technology, Cambridge, United States; [11]Department of Brain and Cognitive Sciences, Massachusetts Institute of Technology, Cambridge, United States

**Abstract** We recently developed expansion microscopy (ExM), which achieves nanoscale-precise imaging of specimens at ~70 nm resolution (with ~4.5x linear expansion) by isotropic swelling of chemically processed, hydrogel-embedded tissue. ExM of *C. elegans* is challenged by its cuticle, which is stiff and impermeable to antibodies. Here we present a strategy, expansion of *C. elegans* (ExCel), to expand fixed, intact *C. elegans*. ExCel enables simultaneous readout of fluorescent proteins, RNA, DNA location, and anatomical structures at resolutions of ~65–75 nm (3.3–3.8x linear expansion). We also developed epitope-preserving ExCel, which enables imaging of endogenous proteins stained by antibodies, and iterative ExCel, which enables imaging of fluorescent proteins after 20x linear expansion. We demonstrate the utility of the ExCel toolbox for mapping synaptic proteins, for identifying previously unreported proteins at cell junctions, and for gene expression analysis in multiple individual neurons of the same animal.

**\*For correspondence:**
esb@media.mit.edu

**Present address:** †Internal Medicine Research Unit, Pfizer Inc, Cambridge, United States

## Introduction

*Caenorhabditis elegans* is an important model system in biology, because of its tractable size (959 somatic cells in adult hermaphrodites), its genetic manipulability, and its optical transparency, which yields the possibility of whole-organism imaging of biological processes and signals. Perhaps not surprisingly, therefore, super-resolution microscopy has been useful to the analysis of *C. elegans*, with studies applying STORM, PALM, SR-SIM, and STED to *C. elegans* to investigate cells and tissues in both intact or dissected *C. elegans* (*Rankin et al., 2011*; *Gao et al., 2012*; *Vangindertael et al., 2015*; *He et al., 2016*; *Köhler et al., 2017*; *Krieg et al., 2017*). However, the depths of imaging of such studies were largely physically limited to a few microns to tens of microns, insufficient to map the entire depth of an adult animal, and the hardware required for super-resolution microscopy is not available in all laboratories, and can be slow and/or expensive to deploy. Furthermore, the tough

cuticle of *C. elegans* presents a barrier to immunostaining in the intact animal, important for STORM and STED imaging and for the general labeling of proteins in a variety of scientific contexts.

Recently, we discovered that it is possible to isotropically expand biological specimens by permeating them evenly and densely with a swellable hydrogel polymer network, anchoring key biomolecules or labels to the hydrogel, softening the tissue through a chemical process, and then adding water, which swells the polymer and in turn the tissue (*Chen et al., 2015*). This technique, expansion microscopy (ExM), is now being adapted and improved by many groups, and has been applied to tissues of mice, human patients, and in many other biological contexts (*Chen et al., 2015*; *Chen et al., 2016*; *Chozinski et al., 2016*; *Ku et al., 2016*; *Tillberg et al., 2016*; *Chang et al., 2017*; *Zhao et al., 2017*; *Park, 2018*; *Truckenbrodt et al., 2018*; *Gambarotto et al., 2019*; *Wassie et al., 2019*). However, *C. elegans* is wrapped in a multi-layer cuticle, which is well known to be impermeable to many small molecules and all antibodies, and mechanically stiff to the point where physical expansion would be expected to proceed poorly (*Duerr, 2006*; *Page and Johnstone, 2007*; *Chisholm and Xu, 2012*). Thus, we set out to develop an ExM protocol customized for the *C. elegans* context that would overcome these barriers.

To achieve this goal, we modified previously published protocols in a number of ways (*Figure 1*, green steps) to generate a new protocol which we call expansion of *C. elegans* (or ExCel). This protocol results in high signal-to-background antibody staining against protease-resistant fluorescent proteins, low-distortion (~1–6% over length scales of 0–100 µm) physical expansion by ~3.3x, and both protein and RNA detection with sub-cellular resolution. Using ExCel, we were able to resolve synaptic and gap junction proteins better than with ordinary confocal microscopy, and simultaneously image proteins, RNA, and DNA location within the same specimen. In particular, such multiplexed capability has not been demonstrated with previous super-resolution methods in *C. elegans*, and facilitates nanoscale-precise analyses of how multiple molecular types are spatially organized in the context of an entire animal.

The standard ExCel protocol visualizes fluorescent reporters, such as those fused to proteins of interest, which requires transgenesis, and could in principle affect the function and localization of the target protein. Thus, we additionally developed an alternative ExCel protocol, which we call epitope-preserving ExCel, that enables detection of untagged, completely endogenous proteins, using off-the-shelf primary antibodies. The epitope-preserving ExCel protocol replaces the use of Proteinase K, a general protease that disrupts most epitopes in the standard ExCel protocol, with an epitope-preserving cuticle-permeabilization treatment that we identified in a systematic screen of chemical treatments. This protocol enables antibody staining of protein epitopes at the expense of a slightly reduced expansion factor (~2.8x) and lower expansion isotropy (~8–25% error over length scales of 0–100 µm). We showed that epitope-preserving ExCel allows multiplexed readout of multiple native proteins at super-resolution, a capability that we used to identify a previously unreported protein localization at the junctions between developing vulval precursor cells, and to resolve the peri-active and active zones of chemical pre-synapses.

Lastly, we developed a third protocol, iterative ExCel (iExCel), which enables two successive rounds of hydrogel-mediated expansion of a given worm, by incorporating the previously validated strategies of iterative expansion microscopy into the ExCel context (*Chang et al., 2017*). iExCel brings the expansion factor from ~4x to ~4x *~4x = ~20x, and the theoretical limit of resolution down to ~25 nm, at a low level of distortion (~1.5–4.5% over length scales of 0–100 µm), on par with that of standard ExCel, on which it builds. With iExCel, we were able to resolve fluorescent puncta that may represent individual GFP molecules expressed in the neuronal cytosol.

Each of these ExCel protocols highlights some of the challenges remaining in deploying ExM in *C. elegans*, including distortion in the gonad and mouth regions, reduced general isotropy with epitope-preserving ExCel, and the ability to only detect fluorescent proteins with the current form of iExCel, which provide grounds for further optimization in the future.

## Results

### Design of the ExCel protocol

In one popular form of ExM, protein retention ExM (proExM; *Tillberg et al., 2016*), a formaldehyde fixed specimen is labeled with fluorescent antibodies, and then the sample is treated with a protein-

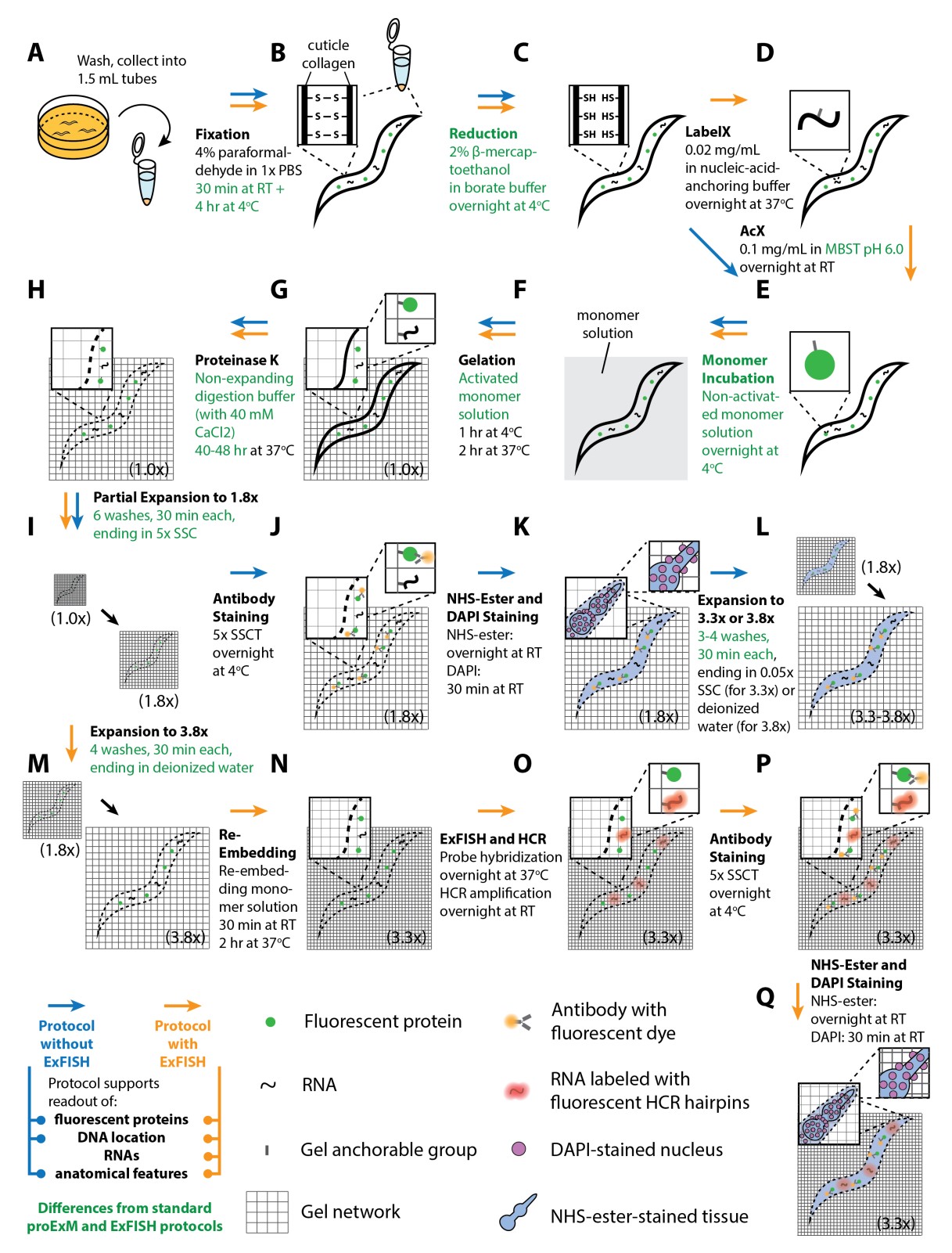

**Figure 1.** Workflow for expansion of *C. elegans* (ExCel) sample processing. A method for expanding cuticle-enclosed intact *C. elegans*, extending published proExM and ExFISH protocols with specific modifications (shown in green text; full key in lower left). Depending on whether the user intends to visualize RNAs or not, the protocol branches into two forms. The protocol without ExFISH, which supports the readout of fluorescent proteins, DNA location (in the form of DAPI staining), and anatomical features, is indicated with blue arrows, ending in Panel L. The protocol with ExFISH, which

*Figure 1 continued on next page*

*Figure 1 continued*

additionally supports readout of RNAs, is indicated with orange arrows, ending in panel Q. For all steps after hydrogel formation (Panels G-Q), the linear expansion factor of the hydrogel-specimen composite is shown in parentheses. (A–Q) Steps of the protocol, with the bold text indicating the title of the step; see text for details of each step.

binding anchor, AcX, which equips the proteins (including the fluorescent antibodies) with a polymer-binding handle; then the sample is evenly permeated by the monomer sodium acrylate (plus a crosslinker). The monomers self-assemble into a dense polymer matrix, a hydrogel of swellable sodium polyacrylate, to which AcX (and thus proteins) are bound. Treatment with a strong protease, Proteinase K, softens the sample by destroying most proteins, but sparing the fluorescent antibodies. Then, adding water swells the hydrogel-specimen composite by ~4.5x in linear extent, for mouse brain specimens. A related process, expansion microscopy followed by in situ hybridization (ExFISH; *Chen et al., 2016*), uses an RNA-binding anchor, LabelX, to couple RNA molecules to the hydrogel, with similar monomer-infusion and proteolysis steps as proExM.

We first designed an ExM protocol for *C. elegans* (schematized in *Figure 1*), extending the published proExM and ExFISH protocols (*Chen et al., 2015*; *Chen et al., 2016*; *Tillberg et al., 2016*) with several design choices that we reasoned would help with isotropic expansion of fixed, intact, cuticle-enclosed *C. elegans*. In outline (see Methods for details), we first collect animals in 1.5 mL Eppendorf tubes to facilitate centrifugation and solution exchange (*Figure 1A*). We fix animals with 4% paraformaldehyde for 30 min at room temperature, and then for 4 hr at 4°C (*Figure 1B*). We then incubate the animals in a buffer containing 2% β-mercaptoethanol overnight at 4°C to chemically reduce the disulfide bonds between collagen fibers in the cuticle, as in *Finney and Ruvkun (1990)*, *Garriga et al. (1993)*, and *Duerr (2006)* (*Figure 1C*). We thought that incorporation of this step into the protocol design could enhance diffusion of chemical reagents for ExM into the specimen.

Next, we add AcX and/or LabelX to the specimen to equip proteins and nucleic acids, respectively, with a polymer-anchorable moiety (*Figure 1D–E*). For applications that involve RNA readout, we treat the specimen with both LabelX and AcX, sequentially and in that order. For other applications, we treat the specimen solely with AcX. The AcX treatment is performed with a low-pH buffer (pH 6.0 MBST, i.e. MES-buffer saline with Triton X-100) building from our prior observation that such a condition helped with deep AcX permeation into specimens (*Tillberg et al., 2016*). We then perform monomer infusion (*Figure 1F*) by incubating the specimen in non-activated monomer solution (7.5% (w/w) sodium acrylate, 2.5% (w/w) acrylamide, 0.5% (w/w) N,N'-diallyl-tartardiamide (DATD; a crosslinker that can be chemically cleaved by periodate ions [*Späth and Koblet, 1979*], which we reasoned would make the hydrogel compatible with ExM-related downstream applications that require the disintegration of this initial hydrogel; we employ this hydrogel-disintegration procedure in two other *C. elegans* expansion protocols that we describe later in this manuscript), in a solution of 50 mM MOPS pH 7.0 and 2 M NaCl) overnight at 4°C. We reasoned that this overnight incubation could promote more thorough diffusion of hydrogel monomers into the specimen, compared to the 30 min incubation in the published proExM or ExFISH protocols. This infusion is followed by polymerization (*Figure 1G*), an extensive Proteinase K digestion over 2 days at 37°C (*Figure 1H*), longer than in previous protocols (*Chen et al., 2015*; *Chen et al., 2016*; *Tillberg et al., 2016*; *Chang et al., 2017*; *Zhao et al., 2017*), to allow a thorough digestion of the cuticle.

Usually when Proteinase K is added to a gelled specimen, a moderate amount (~1.7-fold) of expansion begins, even before the main water-addition swelling step. We thought that distortions might occur if the internal tissue starts partially expanding before the cuticle is thoroughly digested. To prevent premature hydrogel expansion during the digestion process, we designed a digestion buffer containing 40 mM $CaCl_2$, since such divalent cations prevent polyacrylate gel expansion (*Tillberg et al., 2016*). We further considered that slower speeds of expansion could potentially result in more isotropic expansion even after Proteinase K digestion, by permitting time for the gel network to more uniformly stretch out the embedded tissue. Thus, we designed a protocol to partially expand the Proteinase K digested specimens in 6 serial washes lasting 30 min each, incrementally reducing salt concentrations each time (i.e., lowering Tris, NaCl, and $CaCl_2$ concentrations bit by bit throughout each of the 6 washes, ending with 5x saline-sodium citrate buffer (SSC)) (*Figure 1I*). The final expansion factor at the end of the 6 washes is 1.8-fold.

Although the original proExM protocol preserves, to some extent, fluorescent proteins (*Tillberg et al., 2016*), some of the fluorescent proteins are destroyed, a problem exacerbated by a strong Proteinase K digestion protocol such as utilized here. We thus add fluorescent antibodies that bind to the remaining fluorescent proteins, to amplify their fluorescence (*Figure 1J*); the thorough cuticle digestion, we reasoned, would increase the normally poor permeability of the *C. elegans* cuticle to antibodies so that staining would be possible. We next perform optional steps, such as the adding of an N-hydroxysuccinimide ester (NHS ester) of a fluorescent dye, which binds to amines and thus enables the visualization of general anatomical features, or the adding of DAPI to visualize DNA location (*Figure 1K*). Finally, we expand the tissue-hydrogel composite yet further, again at a slow speed by using four serial washes of 30 min each, reducing the amount of SSC each time, until the final specimen is immersed in deionized water (*Figure 1L*). This results in a final expansion of 3.8-fold. However, we found that 0.05x SSC, i.e. a low-salt environment, promotes stability of antibody staining signals better than deionized water, i.e. a no-salt environment, and may be favorable for long-term (>1–3 hr) imaging; to reach this state, we skip the last wash (with deionized water) in the serial washes, so the final specimen ends up immersed in 0.05x SSC, with a final expansion factor of 3.3-fold.

If LabelX was included as an anchor in the corresponding step mentioned above, post-expansion fluorescent in situ hybridization (FISH), followed by hybridization chain reaction (HCR), is possible, as we previously described for single molecule resolution RNA imaging (*Choi et al., 2014*; *Chen et al., 2016*). In the original ExFISH protocol, a linear expansion factor of ~3.3x was achieved, for mouse brain specimens. This expansion factor resulted from first performing RNA detection in the specimen with hybridization probes, then performing HCR to amplify the hybridized products, and finally hydrogel expansion in 0.05x SSC, which has a sufficient salt concentration to maintain stable hybridization of the RNA detection probes, but which results in a lower expansion factor than deionized water. However, newer ('version 3.0') HCR strategies, which offer higher signal-to-background ratio and reduced necessity of optimizing probe sequences on a per-RNA-target-basis (*Choi et al., 2018*) use split-initiator probes that carry half the HCR initiator length of previous versions of HCR probes (18 instead of 36 nucleotides), and thus require a greater salt concentration to maintain hybridization stability between the HCR hairpins and the initiator, which in turn would result in linear expansion factors below ~3.3x. For *C. elegans*, we thought that the tradeoff between hybridization stability and expansion factor could be ameliorated if we first expand the tissue-hydrogel sample to ~3.8x (*Figure 1M*), then re-embed it into a non-expandable hydrogel to lock its size at the expanded state (*Figure 1N*), as described in the published ExFISH and iterative expansion protocols (*Chen et al., 2016*; *Chang et al., 2017*). In practice, the re-embedding monomer solution contains some ions, and thus the linear expansion factor slightly drops from 3.8x to 3.3x during this step (*Figure 1N*). But, after re-embedding, the linear expansion factor remains at ~3.3x, regardless of the salt concentration. Afterwards, we can then perform probe hybridization and HCR amplification (*Figure 1O*), and the resulting hydrogel can be constantly maintained in high-salt environments (such as 5x SSC) that stabilize hybridized and amplified products, while the expansion factor can be maintained at ~3.3x, as previously reported. We finally perform antibody staining (*Figure 1P*), as well as optional NHS-ester staining and DAPI staining (*Figure 1Q*), to visualize fluorescent proteins, anatomical features and DNA location, respectively. For the rest of this paper, we will refer to the protocol outlined above as the 'expansion of *C. elegans*' (ExCel) protocol.

## Visualizing fluorescent proteins via antibody staining with ExCel

We first asked whether the proposed ExCel protocol of *Figure 1* could support immunohistochemistry against fluorescent proteins following the strong Proteinase K digestion of ExCel, in order to achieve signal amplification of retained fluorescent proteins. We applied the steps of *Figure 1A–I* of the ExCel protocol on transgenic animals expressing pan-neuronal cytosolic GFP (*tag-168p::GFP*), and then performed immunostaining with anti-GFP primary antibody and fluorophore (Alexa Fluor 546)-conjugated secondary antibody in the step of *Figure 1J*, followed by DAPI staining (*Figure 1K*) and expansion (*Figure 1L*). We observed bright, uniform anti-GFP staining of the nervous system in ExCel-processed, but not pre-ExCel, animals (*Figure 2A*), suggesting that ExCel could enable good antibody access to the inside of *C. elegans*, and that GFP epitopes could survive the Proteinase K digestion sufficiently to serve as targets for antibody labeling. To quantify these observations, we computed the signal-to-background ratio for antibody staining with ExCel vs. standard processing,

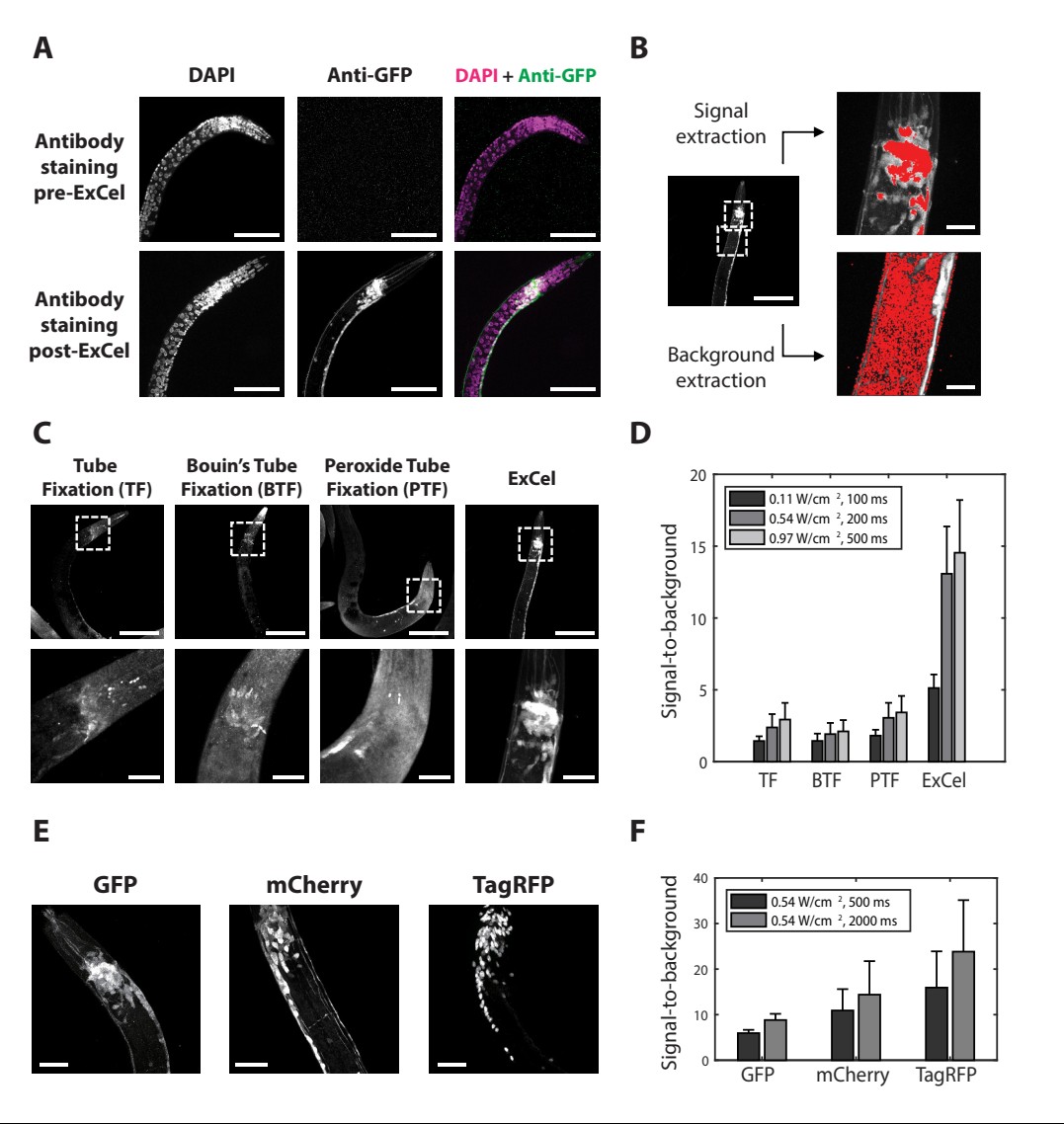

**Figure 2.** ExCel enables antibody-mediated visualization of fluorescent proteins. (**A**) Representative images of immunohistochemistry against GFP in paraformaldehyde-fixed, β-mercaptoethanol-reduced (as in *Figure 1A–C*) hermaphrodite animals, on which the antibody staining was performed without AcX treatment, hydrogel-embedding, Proteinase K digestion and partial expansion ('pre-ExCel') or with such treatments ('post-ExCel'). The strain used had pan-neuronal cytosolic expression of GFP (*tag-168p::GFP*). Images throughout this figure are max-intensity projections of confocal stacks acquired through the entire animal. Brightness and contrast settings: DAPI (left) and the post-ExCel anti-GFP (lower center) images, individually set by the automatic adjustment function in Fiji; the pre-ExCel anti-GFP image (upper center), has the same settings as the post-ExCel anti-GFP image, to facilitate direct comparison. Linear expansion factor: lower images, 3.1x. Scale bars: 50 µm (in biological units, e.g. post-expansion lengths are divided by the expansion factor, used throughout this study unless otherwise noted). (**B**) Quantification of signal-to-background ratio for immunohistochemistry. Image shows a representative transgenic hermaphrodite animal (*tag-168p::GFP*), immunostained with anti-GFP post-ExCel, as in lower images of A. For quantitation, specimens were shrunk back to original size after antibody staining. Images at right are magnified views of the boxed regions at left (centered on the nerve ring and the upper body). Area masks (red) were generated to capture areas corresponding to neurons in the nerve ring region (representing the signal) and non-neuronal tissue (representing the background), using a semi-automated algorithm (see Methods for details). Scale bars: left image, 50 µm; right images, 10 µm. (**C**) Representative images of transgenic hermaphrodite *C. elegans* (*tag-168p::GFP*), immunostained with anti-GFP after various immunohistochemistry protocols (n = 11–17 animals from 3 separately fixed-and-stained populations for each protocol). For purposes of quantitation, the ExCel-processed sample was shrunk back to its original size after antibody staining. Lower panels are magnified views of the boxed regions (centered on the nerve ring) in upper panels. Brightness and contrast settings: each panel is individually set by the automatic adjustment function in Fiji. Scale bars: upper images, 100 µm; lower images, 20 µm. (**D**) Signal-to-background ratio of anti-GFP, computed as in B, for the immunohistochemistry methods performed on worms as in C, for various laser intensities (561 nm, since Alexa Fluor 546 was being imaged) and camera exposure times. Bars indicate mean + / - standard deviation. n = 15, 17, 15, 11 animals, from 3 separately fixed-and-stained populations for each protocol. Source data of the intensity measurements (signal and background), whose population statistics are summarized with the

*Figure 2 continued on next page*

*Figure 2 continued*

bar graph, are available in *Figure 2—source data 1*. (E) Representative images for post-ExCel immunohistochemistry against different fluorescent proteins (n = 7 animals from 2 separately processed sets of animals for each strain) in hermaphrodite animals. The strains expressed one of the following gene constructs: *tag-168p::GFP*, *rab-3p::mCherry*, or *rab-3p::NLS::TagRFP* (NLS, nuclear localization sequence). Signals were from antibody staining (Alexa Fluor 546 for anti-GFP; Alexa Fluor 647 for anti-mCherry and anti-TagRFP). Brightness and contrast settings: individually set by the automatic adjustment function in Fiji. Linear expansion factors: 3.0–3.2x. Scale bars: 20 µm. (F) Signal-to-background ratio, plotted as in D (except with 561 and 647 nm lasers as appropriate, and using post-expansion images), of post-ExCel immunohistochemistry against fluorescent proteins from worms stained as in E. Linear expansion factors, 3.0–3.2x. n = 7 animals from 2 separately stained groups of animals for each strain. Source data of the intensity measurements (signal and background), whose population statistics are summarized with the bar graph, are available in *Figure 2—source data 2*.

The online version of this article includes the following source data for figure 2:

**Source data 1.** Intensity measurements for the signal-to-background ratios shown in *Figure 2D*.
**Source data 2.** Intensity measurements for the signal-to-background ratios shown in *Figure 2F*.

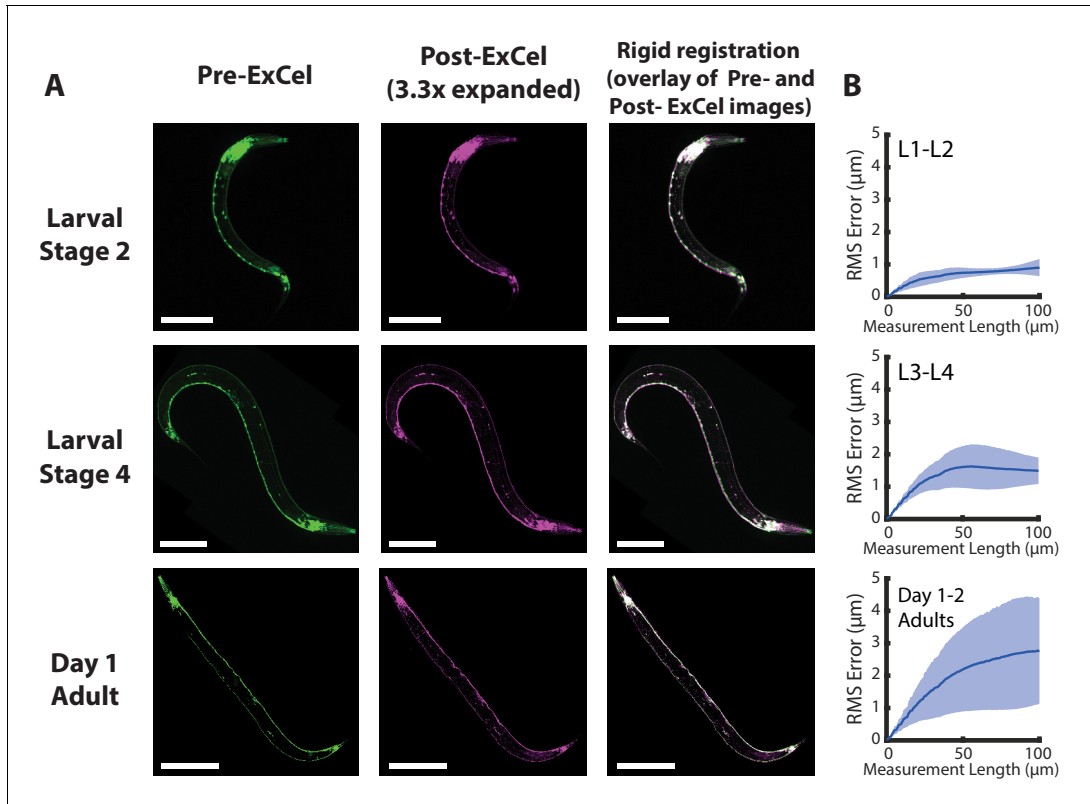

**Figure 3.** Isotropy of ExCel. (A) Representative images of paraformaldehyde-fixed, β-mercaptoethanol-reduced, AcX-treated, and hydrogel-embedded (as in *Figure 1A–C, E–G*) hermaphrodite animals in the second larval stage ('Larval Stage 2'; L2), the fourth larval stage ('Larval Stage 4'; L4) and day 1 adulthood ('Day 1 Adult') before Proteinase K digestion, partial expansion to 1.8x, antibody staining, and expansion to 3.3x ('pre-ExCel') or after such treatments ('post-ExCel'). Pre- and post- ExCel images were rigidly registered with scaled rotation. Strain expressed *tag-168p::GFP*. Signals in the pre-ExCel images were from native GFP; signals in the post-ExCel images were from antibody staining against GFP. Images are max-intensity projections of confocal stacks acquired through the entire animal. Brightness and contrast settings: pre- and post-ExCel images (left and center), first individually set by the automatic adjustment function in Fiji, and then manually adjusted (raising the minimum-intensity threshold and lowering the maximum-intensity threshold) to improve contrast. Linear expansion factor: post-ExCel images, 3.3x. Scale bars: L2, 50 µm; L4, 100 µm; day 1 adult, 200 µm. (B) Root-mean-square length measurement error ('RMS Error') computed from pre- and post- ExCel images, as acquired in A, for L1-L2 larvae (top), L3-L4 larvae (middle), and day 1 – day 2 adults (bottom). Blue line, mean; shaded area, standard deviation. n = 3, 4, 2 animals, from 2 separately processed populations for each age group. Source data of the RMS length measurement errors are available in *Figure 3—source data 1*.

The online version of this article includes the following source data and figure supplement(s) for figure 3:

**Source data 1.** Root-mean-square (RMS) length measurement errors plotted in *Figure 3B*.
**Figure supplement 1.** Local distortion at the gonad region of day 1 – day 2 adult hermaphrodites.
**Figure supplement 2.** Local distortion at the mouth region.

as shown in *Figure 2B*. In short, we defined signal-to-background as the ratio between the average fluorescent intensity of the nerve ring region in the pan-neuronal cytosolic GFP animal (representing the signal), and that of the non-neuronal tissue (e.g. gonad and intestine tissue; representing the background) in the same animal. Signal-to-background ratios obtained with ExCel were several fold higher than those obtained using prior immunohistochemistry protocols (*Finney and Ruvkun, 1990*; *Li and Chalfie, 1990*; *Nonet et al., 1997*; *Duerr, 2006*; *Figure 2C and D*). ExCel worked not just for GFP, but also worked for other fluorescent proteins such as mCherry and TagRFP (*Figure 2E–F*). Thus, ExCel could support antibody-enhanced visualization of fluorescent proteins in *C. elegans*.

## Validation of ExCel isotropy via non-rigid registration analysis

We next examined whether ExCel expanded *C. elegans* evenly. We examined *tag-168p::GFP* animals at different stages of development, and compared pre-expansion to post-expansion images of the same animals, and observed that visible distortion was for the most part quite low (*Figure 3A*). We did observe local distortions, however, at two sites in the worm: around the gonads of adult hermaphrodite animals (*Figure 3—figure supplement 1*) and around the mouth (*Figure 3—figure supplement 2*). In short, for adult gonads, distortions were centered around the vulva, and caused variable tissue displacements in the range of 10 µm (5th percentile) to 25 µm (95th percentile). For mouth regions, distortions were centered around the buccal cavity and procorpus, i.e. the region frontal to the anterior pharyngeal bulb, which resulted in a mouth tip that looks locally compressed, and tissue displacements in the range of 5–20 µm.

To quantify the overall distortion of ExCel-expanded animals, we calculated the root-mean-square (RMS) error of feature measurements from the non-rigidly registered pre- and post-ExCel images (*Figure 3B*), as done previously (*Chen et al., 2015*). We observed that RMS errors positively correlated with animal age. RMS errors over the entire animals were ~1–3% over length scales from 0 to 100 microns for L1-L2 animals,~2–5% for L3-L4 animals, and ~3–6% for day 1 and day 2 adults.

These error measurements are not quite as low as those of published expansion protocols on other tissue types, e.g. <1% on cultured cells (*Chen et al., 2015*; *Tillberg et al., 2016*) and ~2–4% on mouse brain slice (*Chen et al., 2015*), but are in a similar range (1–6% versus 1–4%). Thus, for many biological questions, ExCel exhibits high enough isotropy for making conclusions on par with previously published ExM protocols applied to other tissue types.

## N-hydroxysuccinimide esters (NHS esters) is a novel stain for anatomical features in ExCel-processed animals

We next asked whether ExCel could support multiplexed in situ analyses in *C. elegans*. In situ analyses generally benefit from spatial context, which allow probed features (e.g. protein and nucleic acid targets) of unknown localization to be spatially attributed to known features or compartments (e.g. an identified cell, a tissue type, or an anatomical landmark). We reasoned that a label for spatial context could be particularly useful in an intact, multi-cellular organism that densely contains many tissue types (e.g. nervous, alimentary, reproductive, epithelial, muscle tissues) and anatomical structures (e.g. pharynx, nerve ring, intestines, gonads). In principle, it would be possible to create transgenic animals with multiple fluorescent markers labeling each structure, but here we asked whether a simple labeling strategy, which ideally requires a single, commercially available stain, and which could be rapidly applied, would suffice to provide sufficient contrast to differentiate tissue types throughout the whole animal. The chemical moiety N-hydroxysuccinimide ester (NHS ester) could potentially satisfy these criteria, by reacting to primary amines on proteins. To determine whether the concentration of reactive amines after ExCel processing (which will digest many proteins, but may leave fragments of proteins attached to the hydrogel for staining) varies sufficiently across tissue types and anatomical features to produce sufficient contrast, we incubated ExCel-processed animals in a buffer containing 2 µM Atto 647N NHS Ester (as in *Figure 1K*), and then expanded the animals in deionized water (as in *Figure 1L*). We observed that this post-ExCel NHS-ester staining strategy achieved contrasts sufficient to differentiate between, and to clearly identify, many anatomical features (*Figure 4*, *Videos 1–2*), including (but not limited to) the pharyngeal bulbs, the nerve ring, striated muscles (*Figure 4A*), intestine (*Figure 4B*), gonads (*Figure 4C*), and outlines of many cells (*Figure 4A–C*, *Videos 1–2*). Thus, as a novel tissue staining strategy, NHS esters of fluorescent dyes

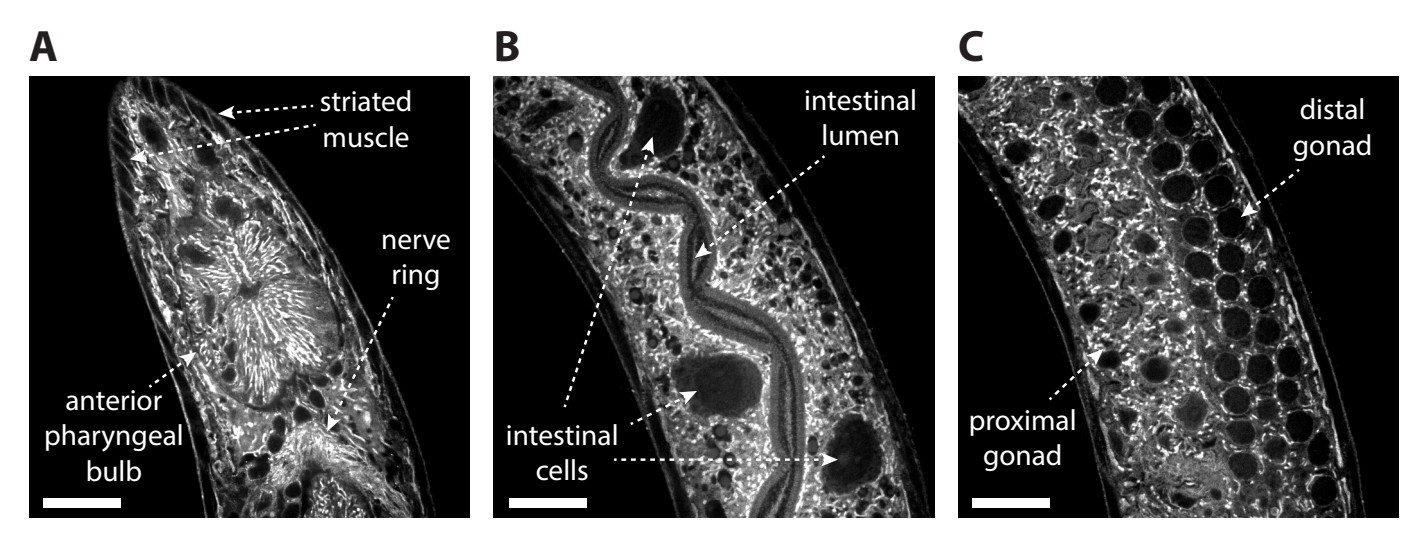

**Figure 4.** Post-ExCel NHS-ester staining reveals anatomical structures. Representative images of (**A**) pharyngeal region, (**B**) intestinal tissue and (**C**) gonad tissue of ExCel-processed (formaldehyde-fixed, β-mercaptoethanol-reduced, AcX-treated, hydrogel-embedded, Proteinase-K digested, partially expanded; as in *Figure 1A–C, E–G*) L3-L4 hermaphrodite animals, stained with Atto 647N NHS ester, which is an NHS ester of a fluorescent dye (as in *Figure 1K*), and then expanded in deionized water (as in *Figure 1L*). The strain used had pan-neuronal expression of RAB-3::GFP (*rab-3p::GFP::rab-3*); not visualized in this specific set of images. Images are confocal micrographs at a single z-plane. Brightness and contrast settings: first set by the automatic adjustment function in Fiji, and then manually adjusted (raising the minimum-intensity threshold and lowering the maximum-intensity threshold) to improve contrast. Linear expansion factor: 4.1–4.2x. Scale bars: 10 μm.

can be applied to ExCel-processed animals to support single-color identification of many anatomical features, which could in turn provide spatial context for other biological signals.

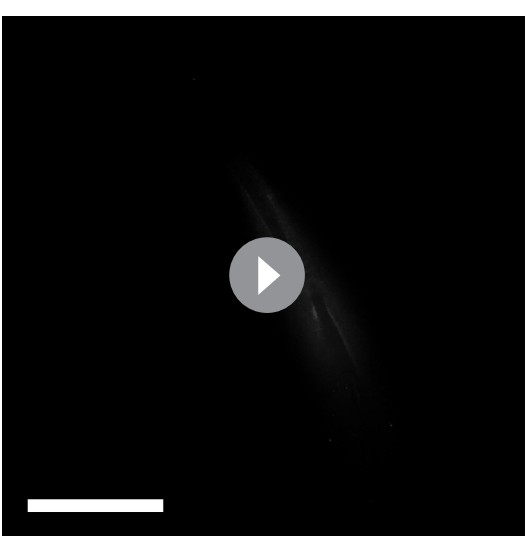

**Video 1.** Post-ExCel NHS-ester staining of the pharyngeal region. Confocal stack of the pharyngeal region of the L3 hermaphrodite animal shown in *Figure 4A*. Scale bar: 20 μm.

https://elifesciences.org/articles/46249#video1

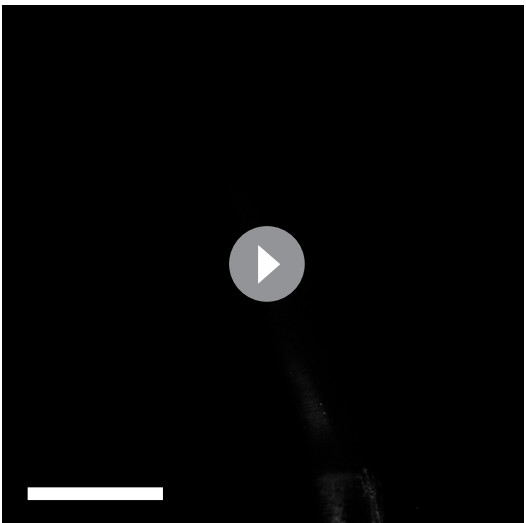

**Video 2.** Post-ExCel NHS-ester staining of the gut and germline tissue. Confocal stack of the upper body region (between the pharynx and the vulva) of the L4 hermaphrodite animal shown in *Figure 4B–C*. Scale bar: 20 μm.

https://elifesciences.org/articles/46249#video2

## Simultaneous readout of fluorescent proteins, RNAs, DNA location and anatomical features

We next explored whether multiple kinds of stain could be combined in the ExCel protocol, for simultaneous readout of fluorescent protein and nucleic acid information. To enable readout of RNA molecules via ExFISH and HCR, without trading off the expansion factor, we performed the re-embedding strategy that we discussed earlier in the Results section. Specifically, we expanded and re-embedded the hydrogel to maintain its expanded state (as in *Figure 1M–N*), which allows the hydrogel to be maintained at an expansion factor of ~3.3x, while immersed in a high-salt environment (5x SSC, which would otherwise shrink the non-re-embedded hydrogel to an expansion factor of ~2.0x) that supports signal stability of the ExFISH-HCR hybridized products. Afterwards, we sequentially performed 4 types of staining procedures on the re-embedded samples. First, we hybridized RNA detection probes to GFP mRNA transcripts, and amplified the detected signal with hybridization chain reaction (HCR) following the published ExFISH and HCR protocols (*Chen et al., 2016*; *Choi et al., 2018*). Second, we used immunohistochemistry to label GFP. Third, we used an NHS ester of a fluorescent dye to stain anatomical features. Fourth, we used DAPI to stain for DNA location. We observed that simultaneous readout of GFP, the mRNA of GFP, DNA and anatomical features could be achieved (*Figure 5*, *Video 3*). From these images, we observed that, as expected, GFP mRNA was found in GFP-labeled neurons (compare *Figure 5A and B*); the nerve ring was visible clearly in both the GFP and the NHS-ester channels (compare *Figure 5A and D*); DAPI-labeled nuclei spatially complemented the dark ellipsoidal spaces with length scales of 2–3 μm observed in the NHS-ester staining (compare *Figure 5C and D*); and relatively little GFP mRNA is found inside the DAPI-labeled nuclei compared to the peri-nuclear space (compare *Figure 5B and C*). Thus, these four labeling modalities can be used in combinations to provide spatial contexts for one another.

## Super-resolution imaging of synaptic puncta

ExCel could potentially achieve a linear expansion factor of ~3.8x after hydrogel expansion in deionized water (*Figure 1L*). Although we rarely used pure water as such, because it encouraged antibody dissociation away from antigens (e.g., anti-GFP signal would drop by ~30% over a 3 hr imaging session), we did try this near-maximal expansion for the purposes of seeing how images would look. At 3.8x linear expansion, the diffraction limited resolution of a typical confocal microscope, using a standard 40x water immersion lens with a numerical aperture of 1.15, would effectively be approximately ~250 nm / 3.8 = ~65 nm (where the 250 nm value is derived from the Abbe diffraction limit [*Abbe, 1873*; *Lipson et al., 1995*]). We applied ExCel to transgenic animals expressing GFP-labeled synaptic proteins RAB-3, SNB-1 or GLR-1, and expanded them to linear expansion factors of 3.8x with deionized water. ExCel resolved nearby puncta that were blurred into a single punctum at the pre-ExCel stage (*Figure 6A*). For the acquisition of quantitative data, we used the more conservative 0.05x SSC expansion protocol (*Figure 1L*), which results in an expansion factor of 3.3x (and yields an effective resolution of ~75 nm, instead of ~65 nm in the fully expanded case); in 0.05x SSC, antibody brightness was stable. We measured the number of synaptic puncta detected from the RAB-3::GFP intensity profiles along segments of ventral nerve cord or SAB axonal processes, in pre- and post- ExCel images (*Figure 6B*), and observed 2.1-fold more detected puncta in post-ExCel images than in pre-ExCel images (*Figure 6C*). Thus, ExCel may be useful for mapping synapses and other key structures in neural circuits of *C. elegans*.

## Super-resolution imaging of innexin, the electrical synapse constituent, at endogenous levels of expression

Recent advances in genome-editing tools, such as CRISPR/Cas9, allow generation of transgenic strains in which a fluorescent reporter is directly fused into the endogenous locus of a target protein. This is in contrast to classical transgenic methods that introduce multiple additional copies of a reporter-fused gene into the animal, which can yield over-expression artifacts, and also can complicate quantitative analysis, because the co-existence of the endogenous, un-tagged gene and the exogenous, tagged gene means that not 100% of the target protein is tagged (*Dickinson et al., 2013*). To test whether ExCel can detect endogenously-tagged proteins expressed at native levels, we applied ExCel to a transgenic animal in which endogenous *che-7*, an innexin that forms electrical synapses in the nervous system (*Bhattacharya et al., 2019*), is fused to a fluorescent protein

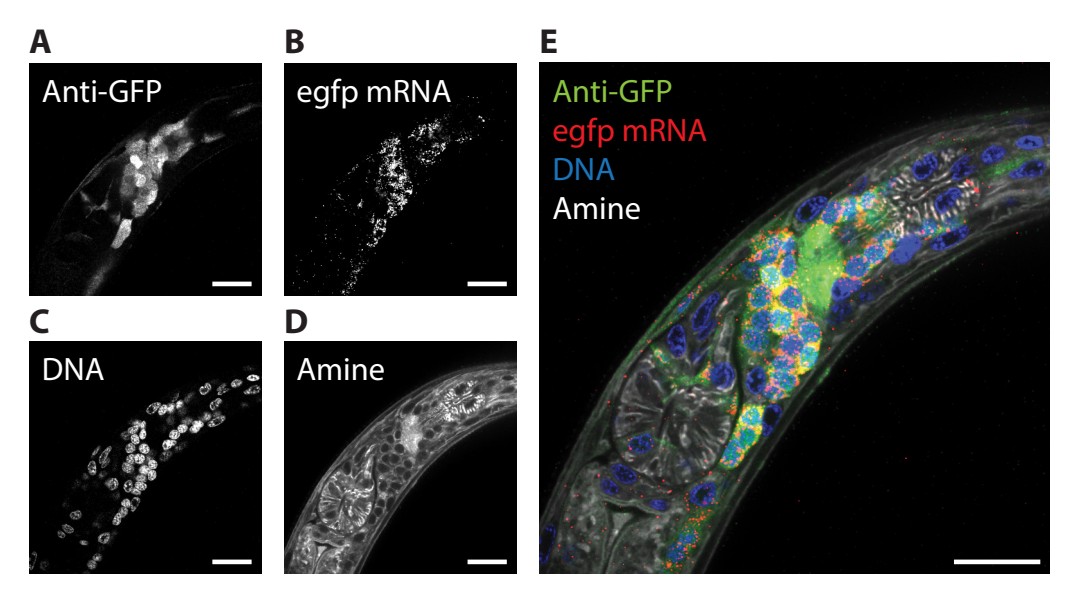

**Figure 5.** ExCel enables simultaneous readout of fluorescent proteins, RNA, DNA location, and anatomical features. The pharyngeal region of a representative ExCel-processed (formaldehyde-fixed, β-mercaptoethanol-reduced, LabelX- and AcX-treated, hydrogel-embedded, Proteinase-K digested and re-embedded; as in *Figure 1A–I, M, N*) L2 hermaphrodite animal, stained sequentially with ExFISH-HCR against the *egfp* mRNA, antibody against GFP, NHS ester of a fluorescent dye (Atto 647N NHS ester; against amines; for anatomical features) and DAPI (for DNA location), as schematized in *Figure 1N–Q*. (A–D) Single-channel images of each staining modality. (E) Merged composite image from combining A-D. Strain expressed *tag-168p::GFP*. Images are single-z-plane confocal micrographs. Brightness and contrast settings: each channel was first set by the automatic adjustment function in Fiji, and then manually adjusted (raising the minimum-intensity threshold and lowering the maximum-intensity threshold) to improve contrast. Linear expansion factor: 3.3x. Scale bars: 10 μm.

(TagRFP) by CRISPR/Cas9-mediated homologous recombination (*Figure 7A*). We observed that CHE-7::TagRFP can be clearly visualized after ExCel, and that its appearance corresponds nearly identically between pre-ExCel and post-ExCel states (*Figure 7B*). Consistent to results obtained using chemical synaptic markers (*Figure 6*), ExCel resolved nearby CHE-7::TagRFP puncta that were blurred at the pre-ExCel stage. Thus, ExCel has sufficient sensitivity to detect fluorescent-reporter-tagged proteins expressed at endogenous levels, and can potentially facilitate mapping of both chemical and electrical synapses in the *C. elegans* nervous system.

## Super-resolved RNA detection at single- and sub- neuronal resolution

It was previously demonstrated that the combination of ExFISH and HCR amplification (ExFISH-HCR) can achieve identification of mRNA transcripts with nanoscale precision and single-molecule resolution (*Chen et al., 2016*). The ExCel protocol uses the same chemistry to covalently retain nucleic acids in the expandable hydrogel, and could in theory allow detection of any anchored RNA in the animal, at single- or sub-cellular resolution. To validate whether ExCel can indeed support general detection of mRNA transcripts, we applied ExFISH-HCR against the mRNAs for *egfp*, mouse parvalbumin, *unc-25*, *cat-2*, *tph-1*, *cho-1*, *eat-4* and *rab-3* on separate groups of ExCel-processed and re-embedded animals in developmental stages between L1 to day 2 adulthood with pan-neuronal cytosolic GFP expression (*tag-168p::GFP*). We then applied immunostaining against GFP to provide spatial

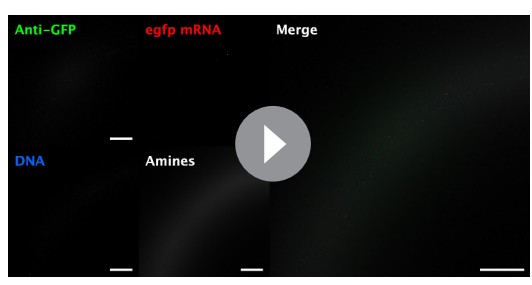

**Video 3.** ExCel 4-color readout. Full confocal stack of the L2 hermaphrodite shown in *Figure 5*. Scale bars: 10 μm.
https://elifesciences.org/articles/46249#video3

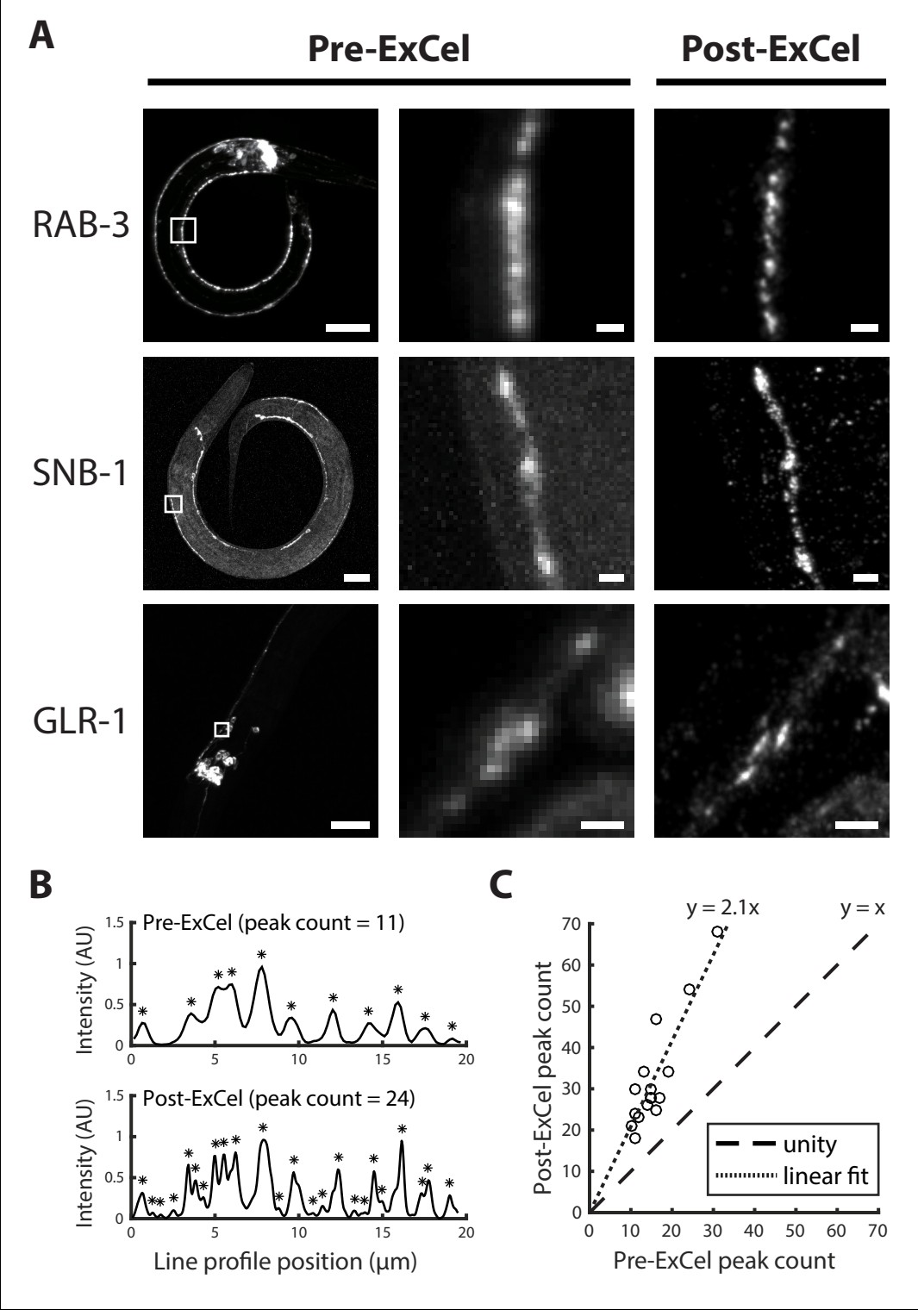

**Figure 6.** Super-resolution imaging of synaptic proteins with ExCel. (**A**) Representative images of GFP-fused synaptic proteins RAB-3, SNB-1 and GLR-1 in paraformaldehyde-fixed, β-mercaptoethanol-reduced, AcX-treated, and hydrogel-embedded (as in *Figure 1A–C, E–G*) hermaphrodite animals, before Proteinase K digestion, partial expansion to 1.8x, antibody staining, and expansion to 3.8x ('pre-ExCel') or after such treatments ('post-ExCel'). Middle images are magnified views of the boxed regions in the left images. Strains express GFP fusions of pre-synaptic proteins RAB-3 (*rab-3p::GFP::rab-3*) or SNB-1 (*unc-25p::snb-1::GFP*), or post-synaptic protein GLR-1 (*glr-*

*Figure 6 continued on next page*

*Figure 6 continued*

*1p::glr-1::GFP*). Signals in the pre-ExCel images were from native GFP; signals in the post-ExCel images were from antibody staining against GFP. Images are max-intensity projections of confocal stacks acquired through the regions of interest. Brightness and contrast settings: left images, individually set by the automatic adjustment function in Fiji; center and right images, first set by the automatic adjustment function in Fiji, and then manually adjusted (raising the minimum-intensity threshold and lowering the maximum-intensity threshold) to improve contrast for the synaptic puncta. Linear expansion factors: post-ExCel images, 3.8x. Scale bars: left images, 20 µm; middle and right images, 1 µm. (B) Representative line intensity profiles of RAB-3::GFP along a section of the ventral nerve cord, from pre- (top) and post- (bottom) ExCel images acquired as in the top row of A (except that the last wash in deionized water is skipped, resulting in improved stability of antibody stained signal and 3.3x linear expansion, as discussed in Main Text). Fluorescent intensity values were linearly normalized to arbitrary units between 0 and 1. Expansion factors of the analyzed post-ExCel image: 3.3x. Asterisks, detected peaks. (C) Peak counts of the line intensity profiles of RAB-3::GFP along sections of ventral nerve cord or SAB axonal processes, pre- and post-ExCel, as plotted in B. Each dot represents a single line profile. Expansion factors of the analyzed post-ExCel images: 3.3x. Dashed line, unity; dotted line, linear fit. n = 16 line profiles from 7 animals in 2 separately processed populations. Source data of the line intensity profiles and their peak counts are available in *Figure 6—source datas 1* and *2*, respectively.

The online version of this article includes the following source data for figure 6:

**Source data 1.** Line intensity profiles for all data points plotted in *Figure 6C*.
**Source data 2.** Peak counts for all data points plotted in *Figure 6C*.

---

context of the entire nervous system, and imaged them with confocal microscopy. As a positive control, we confirmed that nearly all egfp mRNA were localized inside GFP neurons (*Figure 8A*). As a negative control, we found that the mRNA of parvalbumin, a mammalian GABAergic neuronal protein with no known orthologs in *C. elegans* (*Hobert, 2013*), was not detectable in *C. elegans* (*Figure 8B*). We next examined a group of transcripts which encode proteins to synthesize or transport specific neurotransmitters, and which have been described for marking neurotransmitter identity in *C. elegans* neurons, including *unc-25* for GABA synthesis (*Figure 8C*), *cat-2* for dopamine synthesis (*Figure 8D*), *tph-1* for serotonin synthesis (*Figure 8E*), *cho-1* for acetylcholine transport (*Figure 8F*) and *eat-4* for glutamate transport (*Figure 8G*; *Lints and Emmons, 1999*; *Sze et al., 2000*; *Serrano-Saiz et al., 2013*; *Pereira et al., 2015*; *Gendrel et al., 2016*). We also examined *rab-3* (*Figure 8H*), a GTPase that is involved in pre-synaptic vesicle release and was previously reported to express pan-neuronally (*Stefanakis et al., 2015*). For all of the probed targets, we observed that the vast majority (~80–100%) of the ExFISH-HCR signal was detected inside neurons (GFP-filled regions). To ask the question of whether the ExFISH-HCR signal was detected exclusively in neurons with previously reported expression, it would be necessary to perform neuron identification on all ExFISH-positive neurons using an ExFISH-independent identification method, such as through transgenic animals that express deterministic combinations of fluorescent proteins via promoter specification (*Yemini et al., 2019*), an experiment that we did not perform in the present study. However, for each probed transcript target, a subset (3-10) of ExFISH-positive neurons could be feasibly identified based on their unique somatic location, and all of those identified neurons had previously reported expression of the probed transcript. Taken together, these results suggested that ExCel can detect general mRNA transcript targets with single-neuron precision.

In mammalian cells, such as HeLa cells and mouse hippocampal neurons, it has been observed that the vast majority of mRNA transcripts are located in the peri-nuclear space of the cell body (*Chen et al., 2016*; *Samacoits et al., 2018*). We asked the question of whether this general description of mRNA sub-cellular localization also applies to *C. elegans*. To address this question, we performed ExFISH-HCR against mRNA transcripts for *egfp*, *unc-25* and *cat-2* on separate re-embedded samples (each sample was labeled with one transcript target), followed by immunohistochemistry against GFP to visualize neurons, and DAPI staining to visualize nuclei. For all the mRNA transcripts that we imaged, we observed that the vast majority (~80–90%) of transcripts were found in the peri-nuclear space of the cell body (*Figure 9A*, *Video 4*). Overall, we observed a relatively small number of transcripts (~5–20%) in the nuclei, which may indicate transcription sites (*Ji and van Oudenaarden, 2012*). For the three transcript targets we examined, we observed nearly no transcripts in neural processes, especially in regions distant to the soma. As a related observation, nearly no

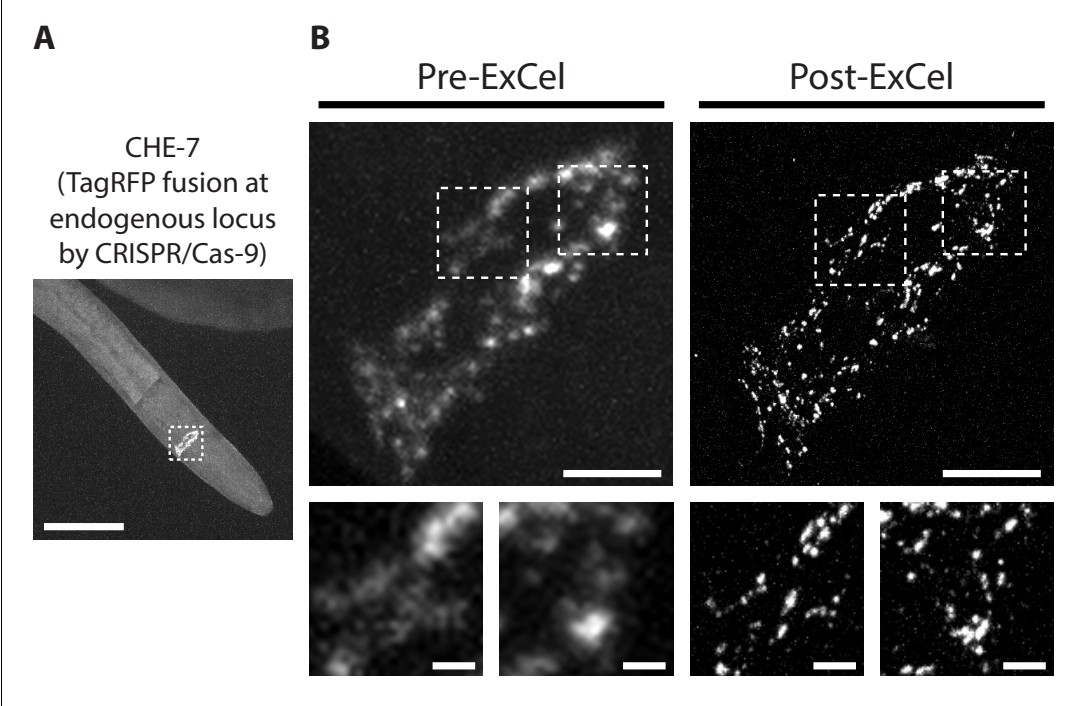

**Figure 7.** Super-resolution imaging of electrical synapses with ExCel. Representative images of TagRFP-fused innexin protein CHE-7 in a paraformaldehyde-fixed, β-mercaptoethanol-reduced, AcX-treated, and hydrogel-embedded (as in *Figure 1A–C, E–G*) L4 hermaphrodite animal, before Proteinase K digestion, partial expansion to 1.8x, antibody staining, and expansion to 3.8x ('pre-ExCel') or after such treatments ('post-ExCel'). Strain expresses innexin protein CHE-7 that is fused to TagRFP at its endogenous locus, via CRISPR-Cas9-mediated homologous recombination. (**A**) Pharyngeal region of the animal. The nerve ring is marked by the dotted box and shown in magnified views in the top panels of B. (**B**) Top panels, nerve ring of the animal, as marked in the dotted box in A. Lower panels, magnified views of the dotted regions in the top panels. Signals in the pre-ExCel images were from native TagRFP; signals in the post-ExCel images were from antibody staining against TagRFP. Images are max-intensity projections of confocal stacks acquired through the depth of the entire animal. Brightness and contrast settings: first set by the automatic adjustment function in Fiji, and then manually adjusted (raising the minimum-intensity threshold and lowering the maximum-intensity threshold) to improve contrast for the synaptic puncta. Linear expansion factors: post-ExCel images, 4.0x. Scale bars: (**A**) 50 μm; (**B**) top panels, 5 μm; bottom panels, 1 μm.

transcripts were detected inside the nerve ring (*Figure 9B*), which is a dense bundle of neural processes and does not contain neuronal soma (*Zallen et al., 1999*). Taken together, these results demonstrate that ExCel enables detection of single mRNA transcripts at spatial resolutions of not only individual cells, but also sub-cellular compartments within a cell.

## RNA quantification at single-neuron resolution, on multiple neurons in the same animal

We next explored the application of ExCel to mapping of a transcript throughout multiple identified neurons in a single animal, performing ExCel (including the re-embedding steps, *Figure 1M–N*) on animals in developmental stages between L1 to day 2 adulthood with pan-neuronal cytosolic GFP expression (*tag-168p::GFP*). We performed ExFISH-HCR against mRNA transcripts of *unc-25*, *cat-2* and *tph-1* on separate re-embedded samples (each sample was labeled with one transcript target), followed by immunohistochemistry against GFP to visualize all neurons in the animal, and DAPI staining to visualize the nuclei (as in *Figure 1O–Q*). We imaged the anti-GFP, ExFISH-HCR and DAPI signals from these animals under confocal microscopy. On the imaged animals, we identified individual neurons based on their stereotypical somatic location, using GFP and DAPI signals, and prior knowledge of endogenous RNA expression pattern, using ExFISH-HCR (*Figure 10A*). We segmented out 3-D stacks that each enclose an identified neuron (*Figure 10B*), using a custom 3-D ROI selection algorithm in MATLAB (see Methods for details), and then detected the ExFISH-HCR amplicons inside each segmented neuron (*Figure 10C*), using a previously described 3-D spot-finding algorithm (*Chen et al., 2016*). After spot detection, we quantified the number of RNA molecules for each

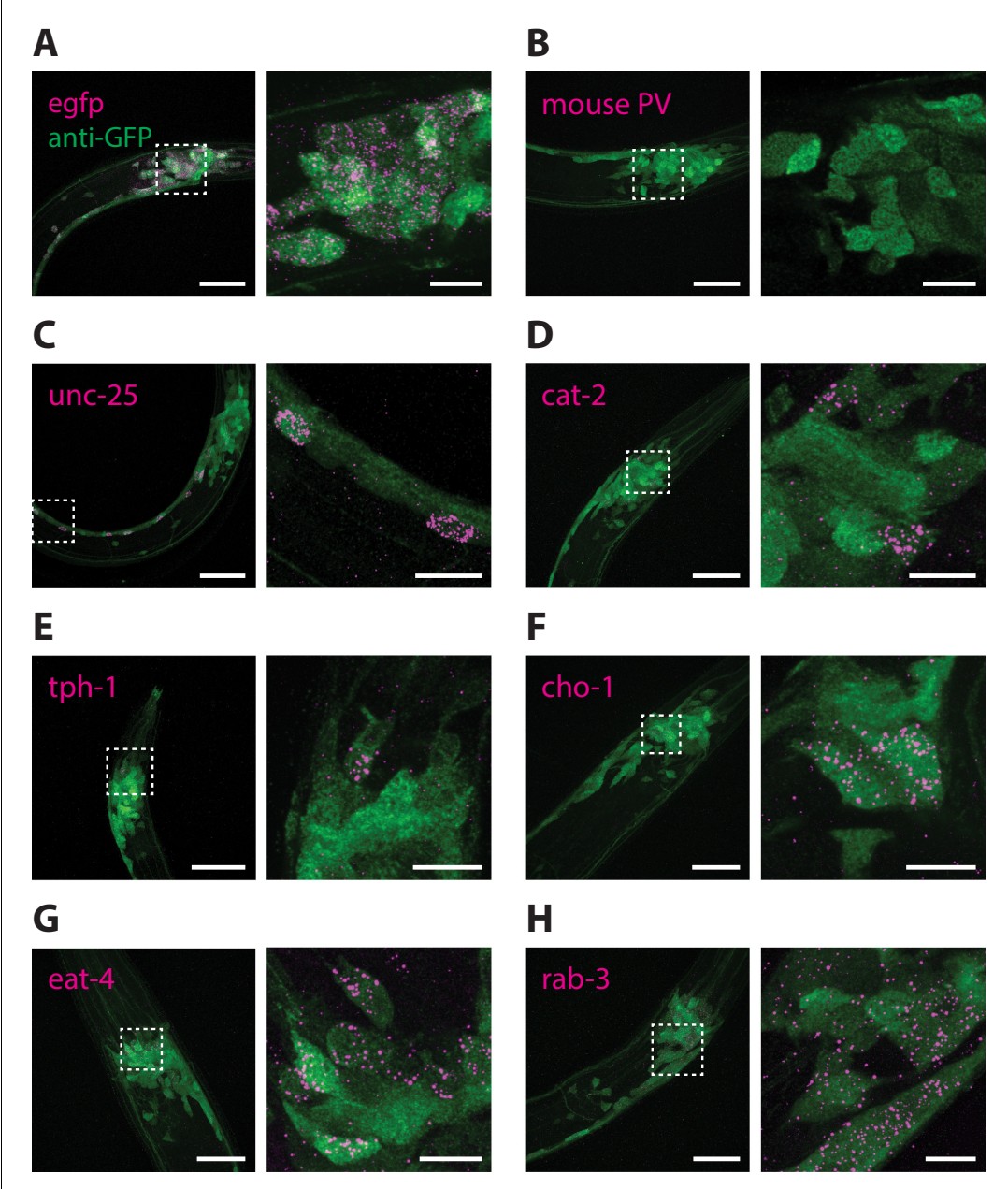

**Figure 8.** RNA detection in neurons. ExCel-processed (formaldehyde-fixed, β-mercaptoethanol-reduced, LabelX- and AcX-treated, hydrogel-embedded, Proteinase-K digested and re-embedded; as in *Figure 1A–I, M, N*) hermaphrodite animals labeled with antibody staining against GFP (green) and ExFISH-HCR (magenta) against the following RNA transcripts: (**A**) *egfp* (as a positive control), (**B**) mouse parvalbumin (no known ortholog in *C. elegans*; as a negative control), (**C**) *unc-25*, a GABAergic neuronal marker, (**D**) *cat-2*, a dopaminergic neuronal marker, (**E**) *tph-1*, a serotoninergic neuronal marker, (**F**) *cho-1*, a cholinergic neuronal marker, (**G**) *eat-4*, a glutamatergic neuronal marker, (**H**) *rab-3*, a pre-synaptic protein with pan-neuronal expression. Right images are magnified views of the boxed regions in the left images. Strain expressed *tag-168p::GFP*. Images are max-intensity projections of confocal stacks acquired through the entire animal (left images) or just the expressing cells (right images). Brightness and contrast settings: first set by the automatic adjustment function in Fiji, and then manually adjusted (raising the minimum-intensity threshold and lowering the maximum-intensity threshold) to improve contrast for cellular morphology and ExFISH puncta. Selection of displayed image: (**A–B**) both localization and density are representative, (**C–H**) localization is representative; density is close to exemplar, because the authors selected images from animals that have relatively strong expression levels, within the 3–5 animals imaged per transcript target, to facilitate visualization of the expressing cells. Nearly all of the selected images are from L2-L4 stage larvae, which have generally greater expression levels than adults, for the transcript targets that we investigated in these panels. Linear expansion factors: 3.4–3.6x. Scale bars: left images, 20 μm; right images, 5 μm.

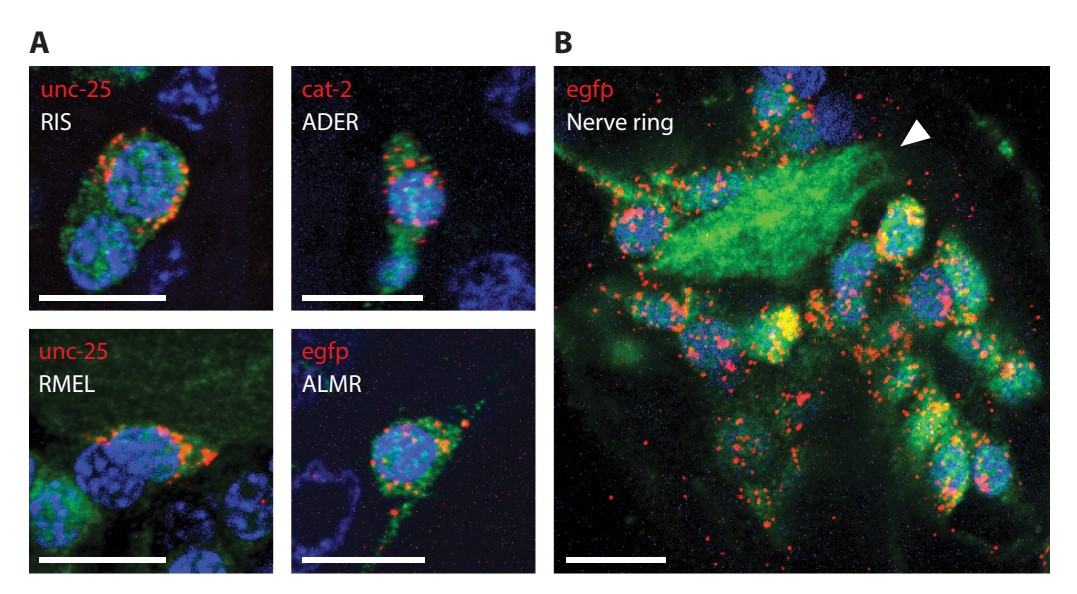

**Figure 9.** RNA detection in neurons, at sub-cellular resolution. Representative images of ExCel-processed (formaldehyde-fixed, β-mercaptoethanol-reduced, LabelX- and AcX-treated, hydrogel-embedded, Proteinase-K digested and re-embedded; as in *Figure 1A–I, M, N*) hermaphrodite animals labeled with anti-GFP (green), DAPI (blue) and ExFISH-HCR (red) against specified mRNA transcripts (red text). Strain expressed *tag-168p::GFP*. Brightness and contrast settings: first set by the automatic adjustment function in Fiji, and then manually adjusted (raising the minimum-intensity threshold and lowering the maximum-intensity threshold) to improve contrast for cellular morphology and ExFISH puncta. Linear expansion factors: 3.1–3.3x. Scale bars: 5 µm. (**A**) Sub-cellular localization of mRNA transcripts in identified single neurons. White text indicates the identity of the displayed neuron, which was determined based on stereotypical somatic location (via GFP signal) and prior knowledge of the expression patterns (via ExFISH signal). Images are max-intensity projections over 5 z-planes (with step size of 0.4 µm, in absolute distance, i.e. post-expansion distance) centered (in z-dimension) at the centerline of the imaged cell. (**B**) Localization of *egfp* mRNA transcript in the head region of an L2 larval animal. White arrowhead, nerve ring.

imaged transcript, in identified single neurons expressing the respective transcript, for *unc-25* (*Figure 10D*), *cat-2* (*Figure 10E*) and *tph-1* (*Figure 10F*). We observed that the expression of the same transcript could differ up to 3.9-fold in expressing neurons, as in the case of *unc-25* mRNA molecule count in RIS (188.3 + / - 72.5; mean + / - standard deviation; n = 3 animals from 1 population) and RMED (48.7 + / - 20.5, mean + / - standard deviation; n = 3 animals from 1 population).

Many *C. elegans* neurons belong to classes of bilaterally or radially symmetric neurons, some of which, such as the bilaterally symmetric ASEL/R and AWCL/R chemosensory neurons, exhibit asymmetries in function and gene expression (*Hobert et al., 2016*). However, differences in the relative expression of genes expressed among members of a single class have not been reported. We used ExCel to fill this gap, comparing single-neuron mRNA counts of probed transcripts in bilaterally and radially symmetric neurons of the same class. We did not observe any significant difference in the expression levels of examined transcripts between L and R neurons of the same class, including *unc-25* levels in RMEL/R (*Figure 10D*), *cat-2* levels in ADEL/R (*Figure 10E*) and *tph-1* levels in NSML/R (*Figure 10F*) (n = 3–7 animals from 1 population for each transcript; 2-sided Wilcoxon rank sum test; alpha = 0.05), which could suggest that the bilaterally symmetric L and R neurons of the same class have generally similar levels of expression. However, when we examined the RME neurons (a class of radially symmetric neurons) inside the same animals, we observed a significant difference in *unc-25* transcript count between RMEV/D and RMEL/R neurons (*Figure 10D*, n = 6 neurons for each sub-class neuron group, over 3 animals from 1 population; p<0.01, two-sided Wilcoxon rank sum test), which provides evidence against the generalization that all neurons of the same class have similar levels of expression. Taken together, these results suggested that ExCel can achieve single-cell-resolution quantification of mRNA transcripts in identified single neurons of the same animal, which can resolve differential expression within the same neuronal class, and which allows collection of linked expression data over the same animal. Given that correlations of gene expression between identified neurons in a neural circuit may reveal principles of how neural circuits are organized and develop

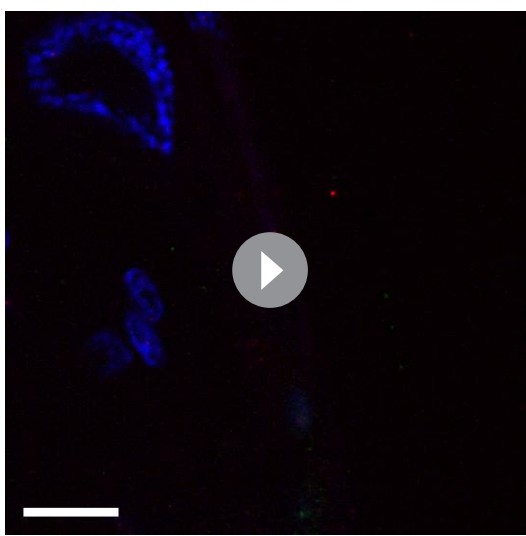

**Video 4.** Sub-cellular localization of unc-25 transcripts in motor neurons. The retrovesicular ganglion of a representative ExCel-processed (formaldehyde-fixed, β-mercaptoethanol-reduced, LabelX- and AcX-treated, hydrogel-embedded, Proteinase-K digested and re-embedded; as in *Figure 1A–I, M, N*) L4 hermaphrodite animal labeled with anti-GFP (green), DAPI (blue) and ExFISH-HCR (red) against unc-25. The unc-25-expressing cells are motor neurons DD1 (bottom), VD1 (center) and VD2 (top). Strain expressed *tag-168p::GFP*. Brightness and contrast settings: first set by the automatic adjustment function in Fiji, and then manually adjusted (raising the minimum-intensity threshold and lowering the maximum-intensity threshold) to improve contrast for cellular morphology and ExFISH puncta. Linear expansion factor: 3.2x. Scale bar: 5 μm.

https://elifesciences.org/articles/46249#video4

(*Schulz et al., 2006*), this may be very useful for the mapping of plasticity and homeostasis mechanisms in *C. elegans*.

## Development of an alternative ExCel protocol for detecting general endogenous epitopes

In the standard ExCel protocol (*Figure 1*), a strong treatment with Proteinase K (*Figure 1G–H*) is used to both permeabilize the cuticle for antibody access (*Figure 2A*), and to confer high isotropy of expansion. Such treatment disrupts most protein epitopes, except for protease-resistant ones, such as fluorescent proteins. While the versatile genetic toolbox of *C. elegans* makes it relatively feasible to fuse target proteins of interest with fluorescent proteins, transgenesis could still be a labor-intensive process (e.g. compared to applying an affinity probe against a target), especially for investigating multiple protein targets. More importantly, fusion with fluorescent proteins could in principle disrupt expression level, localization and/or function of target proteins (*Snapp, 2005*; *Hammond et al., 2010*; *Crivat and Taraska, 2012*). It would therefore be useful to develop an alternative ExCel protocol that supports detection of arbitrary, untagged endogenous protein targets, via completely post hoc methods such as immunohistochemistry.

There are two strategies for introducing general-target immunohistochemistry into the ExCel pipeline – either prior to gelation and partial expansion (e.g. at the state of *Figure 1C*), or afterwards. We decided to proceed with the latter option, because our results with anti-GFP indicated that the former option resulted in lower uniformity and intensity of staining (*Figure 2A,C, D*), due to the insufficiently permeabilized cuticle. While stronger cuticle permeabilizations, such as physical cracking of the frozen cuticle, and protease digestion, can improve staining (*Duerr, 2006*), such treatments can also adversely affect the physical integrity of the worms. In principle, the strategy of performing immunostaining after gelation and partial expansion (*Figure 1A–G*) could solve this problem, by first covalently anchoring proteins to the hydrogel network, which preserves their spatial organization. Then, strong permeabilization treatments could be applied.

Therefore, we sought to identify an epitope-preserving post-gelation treatment that could replace the Proteinase K digestion. Such a treatment would need to fulfill two criteria. First, like Proteinase K, the treatment would need to mechanically homogenize the tissue, since otherwise fixed worm tissue could only be expanded by a limited amount (as in the first row of *Figure 11A*; without any treatment, the worm expanded only by 1.4x linearly whereas the surrounding hydrogel expanded by 3.5x linearly). We developed an assay to score such capability to confer tissue expansion (*Figure 11A*). In this assay, hydrogel-embedded animals (as prepared by steps shown in *Figure 1A–C, E–G*) first undergo a candidate treatment (i.e. a single entry among all the post-gelation treatments in this screen), and then are expanded by deionized water, until the hydrogel reaches ~3.5x linear expansion factor (as shown in the third column in *Figure 11A*). At this stage, the animal expansion factors are measured, and normalized by the hydrogel expansion factor. (The normalization step removes sample-to-sample variability in hydrogel expansion factors, which had a standard deviation of 0.16x around the mean of 3.53x; n = 52 hydrogel samples from 3 separate

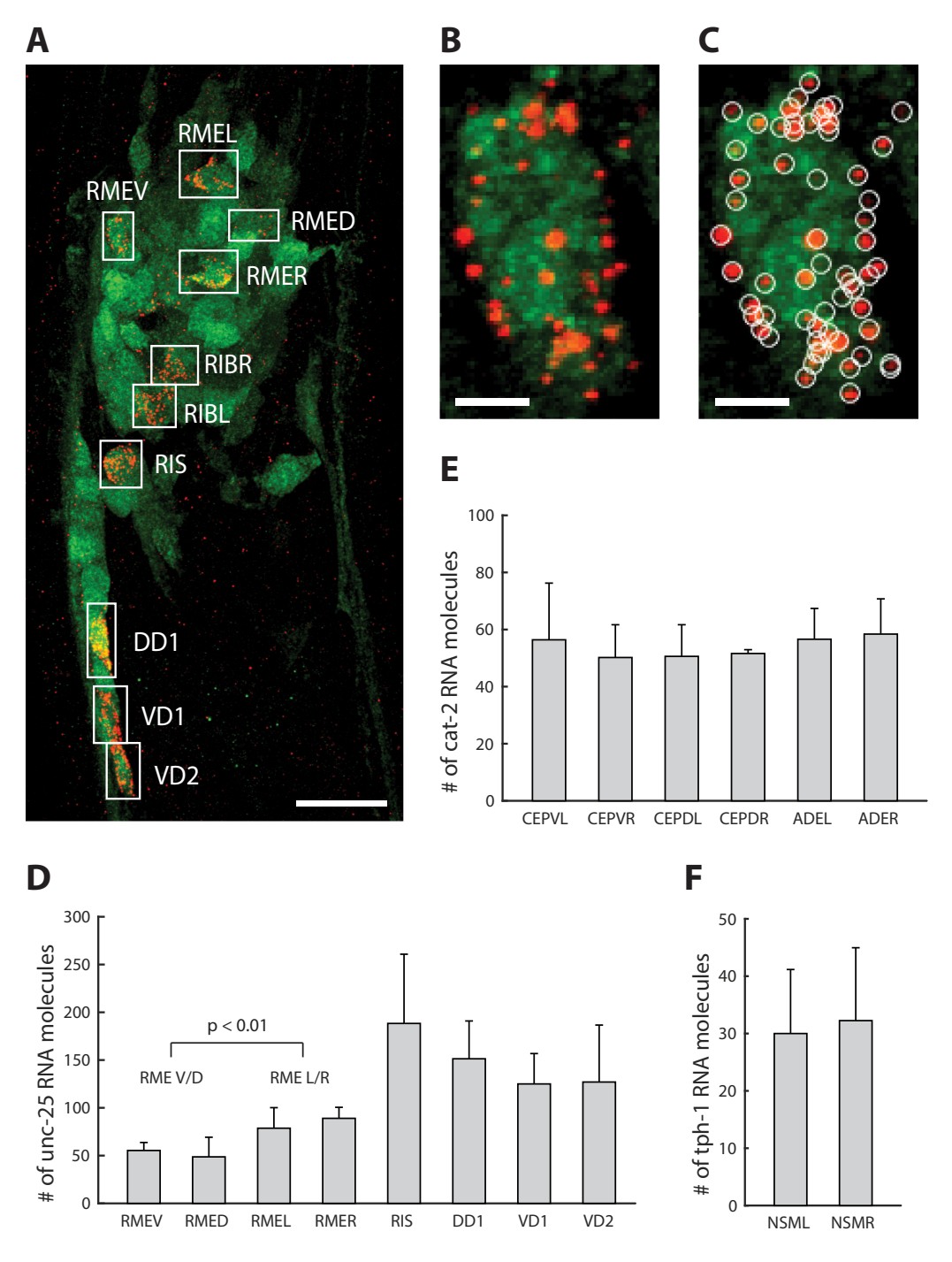

**Figure 10.** Single-neuron resolution RNA quantification. (**A**) A representative ExCel-processed (formaldehyde-fixed, β-mercaptoethanol-reduced, LabelX- and AcX-treated, hydrogel-embedded, Proteinase-K digested and re-embedded; as in *Figure 1A–I, M, N*) L4 hermaphrodite animal labeled with anti-GFP (green), ExFISH-HCR against unc-25 mRNA transcript (red) and DAPI (not shown for image clarity). Boxes are manually selected ROIs enclosing single neurons that were identified based on stereotypical somatic location (via GFP and DAPI signal) and prior knowledge of unc-25 expression pattern (via ExFISH signal). Strain expressed *tag-168p::GFP*. Image is a max-intensity projection of a confocal stack acquired through the entire animal. Brightness and contrast settings: first set by the automatic adjustment function in Fiji, and then manually adjusted (raising the minimum-intensity threshold and lowering the maximum-intensity threshold) to improve contrast for cellular morphology and ExFISH puncta. Linear expansion factor: 3.2x. Scale bar: 10 μm. (**B**) Magnified view of the region of interest enclosing RMEV from A. Scale bar: 1 μm. (**C**) Same image as in B, with detected ExFISH-HCR spots (white circle) that correspond to *unc-25* mRNA molecules. Spot detection was performed on the confocal stack by a 3D spot-finding algorithm (see Methods for details). (**D–F**) mRNA molecule count of (**D**) *unc-25*, (**E**) *cat-2* and (**F**) *tph-1* in each expressing neuron around

*Figure 10 continued on next page*

*Figure 10 continued*

the nerve ring, by applying the analytical workflow shown in A-C to separate groups of ExCel-processed animals singly labeled for each specified transcript target. Bar height, mean; error bars, standard deviation. Analyzed animals were between L2-L4 stages. n = 3–7 animals from 1 population, for each transcript-neuron combination. Values for neurons with n < 3 (due to cases in which a neuron cannot be confidently identified) were not shown. The RMEV/D group pools single-neuron mRNA molecule count of *unc-25* from RMEV and RMED (n = 6 neurons from 3 animals from 1 population), whereas the RMEL/R group pools from RMEL and RMER (n = 6 neurons from 3 animals from 1 population). P-value, two-sided Wilcoxon rank sum test. Source data of the single-neuron RNA counts, whose population statistics are summarized with the bar graphs shown in D-F, are available in *Figure 10—source data 1*.

The online version of this article includes the following source data for figure 10:

**Source data 1.** Count of unc-25, cat-2 and tph-1 RNA molecules within single neurons, whose population statistics are shown in *Figure 10D-F*.

experiments) If a treatment sufficiently homogenized the worm tissue, the worm would expand by nearly the same extent as the surrounding hydrogel does, and result in a normalized expansion factor of unity. As we expected, we observed that a collagenase treatment, which digests collagens, abundant structural proteins in the worm cuticle, resulted in improved normalized expansion factor (66%; third row in *Figure 11A*) compared to the no-treatment condition (40%; first row in *Figure 11A*), and that the Proteinase K treatment from the standard ExCel protocol, which we added as a positive control known to achieve thorough tissue homogenization (as shown in *Figure 3*), resulted in a normalized expansion factor close to unity (97%; second row in *Figure 11A*).

As a second criterion, such a treatment would need to permeabilize the cuticle for antibody access, as did Proteinase K, but without disrupting the protein epitopes required for antibody binding. We developed an assay to screen for the combination of these two effects, by evaluating the quality of immunostaining on the treated animals. As different epitopes could respond differently to any given treatment, we utilized a panel of multiple targets to estimate the generality of epitope stainability (GFP, expressed pan-neuronally; LMN-1, a nuclear envelope protein; myotactin, a protein in the hypodermis; DLG-1, an adherens junction marker; and acetylated tubulin, a touch-receptor-neuron marker [*Siddiqui et al., 1989*; *Hadwiger et al., 2010*]). These antibodies were selected because the authors of this paper were familiar with the expected patterns of staining, which were distributed into four spectral channels. We then scored the quality of immunostaining for each channel (*Figure 11B*), and then used the average score across all channels to represent the general epitope stainability for a given treatment. As we expected, the no-treatment negative control resulted in poor staining over all channels due to insufficient cuticle permeabilization. Proteinase K yielded excellent GFP staining, as expected, but low staining in most other channels (except for anti-myotactin, which yielded the expected pharyngeal staining with a moderate signal). Lastly, collagenase type II treatment, which could potentially permeabilize the cuticle (as the cuticle is majorly composed of collagen) while preserving general antigens (as collagenase recognizes a glycine-proline-hydroxyproline sequence that is highly specific to collagens), showed strong immunostaining results in 3 out of the 4 channels, and received an average immunohistochemistry (IHC) score of 0.67.

We used these two assays (*Figure 11A and B*) to screen through 22 different candidate postgelation treatments, which are summarized in *Table 1*. We explored using non-Proteinase-K proteases with different substrate specificity, treatments derived from the antigen retrieval literature (which has been described to reverse formaldehyde-mediated crosslinks introduced by fixation, and could thus potentially improve tissue expandability and epitope stainability), and treatments that were derived from the magnified analysis of the proteome (MAP) protocol, which had been shown to permit a wide range of antibody stains following tissue expansion in other tissue types (*Ku et al., 2016*).

The results of the screen are summarized in *Figure 11C*. As we expected, the no-treatment condition resulted in lower expandability (represented by the normalized expansion factor, on the x-axis) and stainability (represented by the average IHC score, on the y-axis) than nearly all other treatments. The Proteinase K treatment from the standard ExCel protocol resulted in nearly the highest expandability among tested treatments, and a near-median stainability (due to high quality of anti-GFP staining and moderate quality of anti-myotactin staining). Several substrate-specific proteases (including collagenase type II, sequencing-grade trypsin, alpha-chymotrypsin and elastase) outperformed Proteinase K in stainability, but at the expense of expandability. Treatments adopted from the antigen-retrieval literature performed similarly to the no treatment control, suggesting that

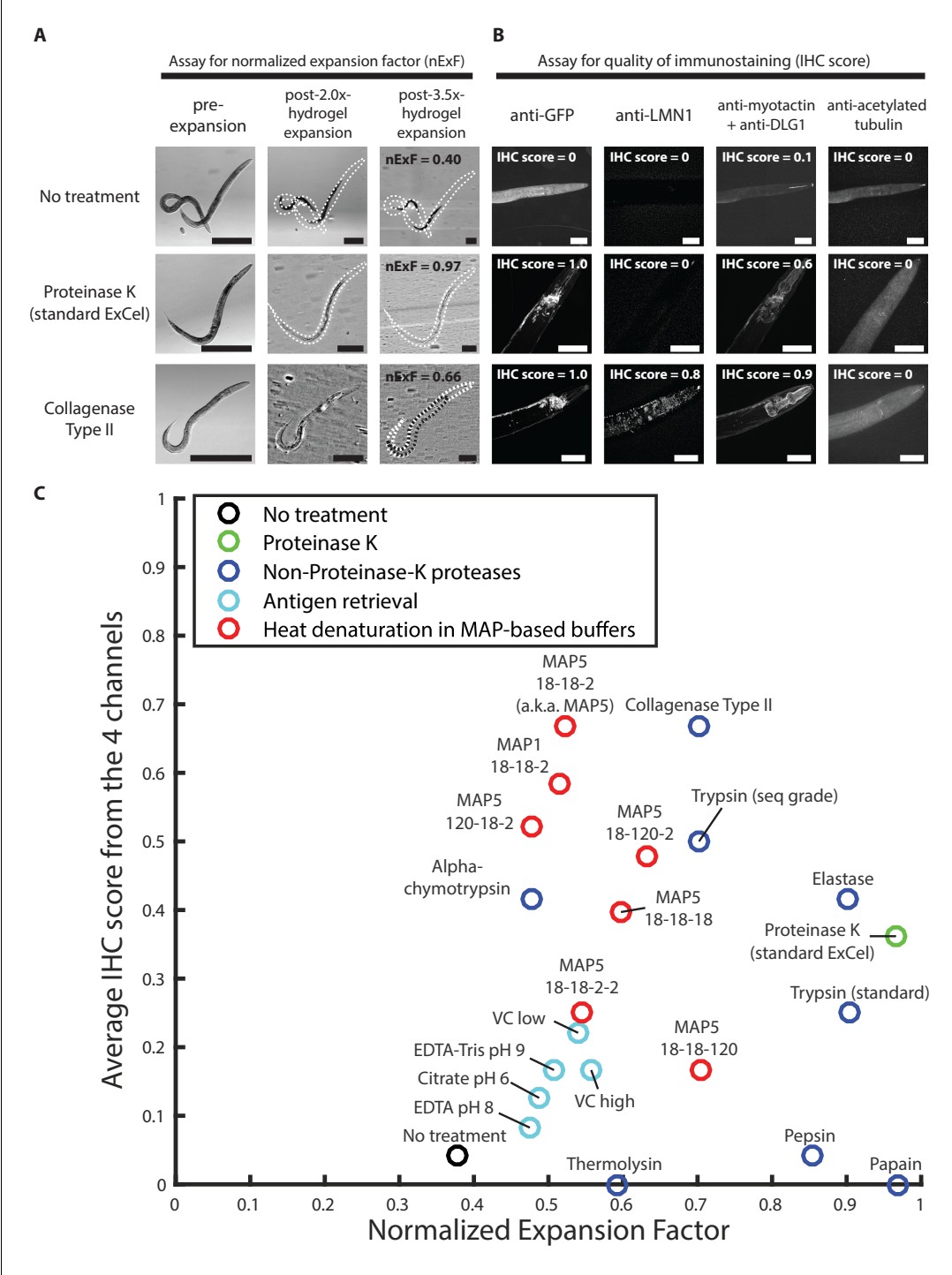

**Figure 11.** Screen of post-gelation treatments that confer tissue expandability and general stainability of epitopes. (**A**) Representative transillumination images of paraformaldehyde-fixed, β-mercaptoethanol-reduced, AcX-treated, and hydrogel-embedded hermaphrodite animals, (left column) right after hydrogel embedding and prior to any hydrogel expansion, (middle column) after 1.9x-2.1x hydrogel expansion, by incubating the gelled sample in 1x PBS, or (right column) after 3.3–3.7x hydrogel expansion, by sequentially washing the gelled sample with 0.5x PBS, 0.1x PBS, and 0.01x PBS. After hydrogel embedding, gelled samples either are left in TNT buffer (top row; no treatment), processed with a 2 day 37°C Proteinase K digestion, as in the standard ExCel protocol (middle row), or processed with a 5 day 37°C collagenase type II digestion (bottom row). Transillumination images provide visualization to both the contour of the worm (traced under high digital magnification in black dotted lines, in cases where direct observation is difficult due to reduced tissue scattering after hydrogel expansion), and also the contour of the mold in the embedding hydrogel (traced in white dotted lines, in cases where direct observation is difficult to reduced gel-boundary scattering after hydrogel expansion). For each treatment, the expansion factor of

*Figure 11 continued on next page*

*Figure 11 continued*

the worm (measured as the length ratio of a worm in the pre-expansion and the post-3.5x-expansion (hydrogel) state) is normalized by the expansion factor of the embedding hydrogel, which results in a normalized expansion factor (abbreviated as nExF), to remove the variation on worm expansion factor due to inter-sample variation in the hydrogel expansion factor. For the no-treatment condition (top row) and the collagenase type II condition (bottom row), where the normalized expansion factors are markedly less than unity (0.40 and 0.66, respectively), the hydrogel-embedded worm tissue detaches from the surrounding hydrogel, due to tissue mechanical hindrances against expansion that are incompletely removed by the post-hydrogel-embedding treatment, and can be visualized by the extent of mismatch between the worm contour and the hydrogel-mold contour. Images are single-plane wide-field acquisitions. For post-2.0x- and 3.5x- images, in cases where uneven illumination from the bright-field light source strongly affects contour visualization, a band-pass filtering with the boundary of 3 and 30 pixels was performed with the Fiji function 'Bandpass Filter' to remove the illumination artifact, and to improve contour visualization. Brightness and contrast settings: first set by the automatic adjustment function in Fiji, and then manually adjusted (raising the minimum-intensity threshold and lowering the maximum-intensity threshold) to improve contrast for the boundaries of the worm and the mold. Scale bars: 300 µm in actual units (not converting to biological units here, since the two features (worm and hydrogel-mold) are associated with different expansion factors). (B) Representative images of the immunostaining of hydrogel-embedded animals in (A), via a panel of 5 primary antibodies with known patterns of staining. Due to spectral limitation, the five antibodies were separated into four spectrally separable channels (DyLight 405 for anti-GFP, Alexa 488 for anti-LMN1, Alexa 546 for anti-myotactin and anti-DLG1, Alexa 647 for anti-acetylated tubulin). An IHC score from 0 to 1 was manually assigned to each channel, based on the estimated signal-to-noise ratio of the expected pattern of staining, and thereby provides a rough quantification for the quality of immunostaining of each antibody (or pair of antibodies) following the specified post-hydrogel-embedding treatment. The strain used had pan-neuronal cytosolic expression of GFP (*tag-168p::GFP*). A few patterns of channel crosstalk, such as the anti-GFP signal observed in the anti-myotactin + anti-DLG1 channel, were observed but do not affect the scoring process, because the known patterns of staining for each of the five antibodies were spatially separable (GFP, pan-neuronal by promoter choice; LMN1 (lamin), nuclear; myotactin, periphery of pharyngeal muscle and beneath cuticle; DLG1 (disc large), adherens junctions that form characteristic thread-like patterns across the length of the worm; acetylated tubulin, touch-receptor neurons). Images are max-intensity projections of confocal stacks acquired through the entire animal. Brightness and contrast settings: individually set by the automatic adjustment function in Fiji. Linear expansion factors of the hydrogel: 1.9–2.1x (after immunostaining, the samples were left in 1x PBS and imaged in that state, without further expansion in deionized water; we decided to use this procedure here, because we observed that even at this partially expanded state, we could already evaluate whether the staining against protein targets yielded the expected patterns of localization, as demonstrated by the images in this panel, without the additional improvements in resolution that would result from further expansion of the samples). Linear expansion factors of the worm: no treatment, 1.1x; Proteinase K (standard ExCel), 1.9x, Collagenase Type II, 1.6x. Scale bars: left images, 50 µm (in biological units, i.e. post-expansion lengths are divided by the expansion factor of the worm). (C) Summary of the screen of 22 post-hydrogel-embedding treatments, each of which is characterized by (X axis) the post-treatment expandability of the worms, as quantified by the normalized expansion factor analysis as performed in A, and (Y axis) the post-treatment quality of immunohistochemistry, as quantified by the average of IHC scores across the four channels in the immunostaining assay as performed in B. Each dot represents a single treatment. See Methods for the protocol performed for each treatment. Treatments are grouped based on the nature of the protocol, and colored according to the group they belong to (legend). X- and Y- coordinates of each treatment represent the mean values of all animals analyzed in the expandability assay (which quantifies the normalized expansion factor, as in A) and the immunostaining assay (which quantifies the IHC score, as in B), respectively. Number of animals analyzed in the assay: expandability assay, 3–4 animals from 1 hydrogel sample; 4-channel immunostaining assay, 2–4 animals from 1 hydrogel sample, except for the papain treatment (1 animal). The condition displayed as MAP5 18-18-2 (heat denaturation in MAP5 buffer for 18 hr at 37°C, 18 hr at 70°C, and 2 hr at 95°C) is abbreviated as simply 'MAP5' in later figures. Source data of the measurements made in the expandability and the stainability assays are available in *Figure 11—source data 1*.
The online version of this article includes the following source data for figure 11:

**Source data 1.** Measurements for the expandabiliy and stainability assays, whose population statistics are summarized in *Figure 11C*.

---

fixative reversal itself might be insufficient to grant cuticle permeabilization and tissue homogenization, both of which could be more related to the mechanical properties of native tissue, rather than to covalent crosslinks made by the fixative. Lastly, heat-mediated denaturation treatments in buffers that we adopted from the MAP protocol (i.e., MAP1 buffer, which is the original buffer from the protocol and contains 50 mM Tris pH 9.0, 200 mM sodium dodecyl sulfate (abbreviated as SDS; a protein denaturant), and 200 mM NaCl; and MAP5 buffer, which is a modified buffer containing 200 mM of additional NaCl (for 400 mM total) and 20 mM $CaCl_2$ (to reduce tissue expansion during treatment, as employed in the standard ExCel protocol)) resulted in modest expandability but high stainabilities (relative to the rest of the treatments) under certain temperature profiles (e.g. MAP5 18-18-2, which denotes a treatment in MAP5 buffer for 18 hr at 37°C, 18 hr at 70°C, and 2 hr at 95°C).

None of the screened treatments landed in the upper right corner, which represents the ideal of both full expandability and high-quality immunostaining. We reasoned that expansion factor could always be improved later, for example by applying expansion iteratively (*Chang et al., 2017*), whereas permanent loss of epitopes could not be compensated for by a downstream step. Therefore, we focused on the treatments that in our screen scored highest on stainability, including

**Table 1.** Screened post-gelation treatments.

Experimental parameters used for each post-hydrogel-embedding treatment that was screened and shown in *Figure 11C*.

| Treatment type | Treatment name | Treatment buffer | Buffer pH | Protease concentration | Duration (triple numbers, times at 37°C-70°C-95°C, in hr) |
|---|---|---|---|---|---|
| No treatment | No treatment | TNT (Tris, 1M NaCl, Triton) | 8.0 | N/A | 0-0-0 (kept at RT) |
| Proteinase K | Proteinase K (standard ExCel) | 50 mM Tris pH 8 + 0.5M NaCl + 40 mM CaCl$_2$ + 0.1% Triton | 8.0 | 8 U/mL | 2 days 37°C* |
| Non-Proteinase-K proteases | Trypsin (standard) | 50 mM Tris pH 8 + 0.5M NaCl + 40 mM CaCl$_2$ | 8.0 | 1 mg/mL | 5 days 37°C* |
| | Trypsin (seq grade) | 50 mM Tris pH 8 + 0.5M NaCl + 40 mM CaCl$_2$ | 8.0 | 0.1 mg/mL | 5 days 37°C** |
| | Elastase | 50 mM Tris pH 9 + 0.5M NaCl + 40 mM CaCl$_2$ | 9.0 | 0.5 mg/mL | 5 days 37°C* |
| | Pepsin | 3 mM HCl + 0.5M NaCl + 40 mM CaCl$_2$ | 2.5 | 1 mg/mL | 5 days 37°C* |
| | Thermolysin | 50 mM Tris pH 8 + 0.5M NaCl + 40 mM CaCl$_2$ | 8.0 | 0.5 mg/mL | 5 days 70°C* |
| | Papain | 1x PBS pH 6.5 + 5 mM L-cysteine + 5 mM EDTA + 2M NaCl | 6.5 | 1 mg/mL | 5 days 70°C* |
| | Alpha-chymotrypsin | 50 mM Tris pH 8 + 0.5M NaCl + 40 mM CaCl$_2$ | 8.0 | 1 mg/mL | 5 days 25°C* |
| | Collagenase Type II | 50 mM Tris pH 8 + 0.5M NaCl + 40 mM CaCl$_2$ | 8.0 | 1 U/mL | 5 days 37°C* |
| Antigen retrieval | EDTA pH 8 | 10 mM Tris + 1 mM EDTA + 2M NaCl | 8.0 | N/A | 18-18-2 |
| | EDTA-Tris pH 9 | 50 mM Tris + 1 mM EDTA + 0.05% Tween + 2M NaCl | 9.0 | N/A | 18-18-2 |
| | Citrate pH 6 | 10 mM citrate pH 6 + 0.05% Tween + 2M NaCl | 6.0 | N/A | 18-18-2 |
| | VC low | 5% (w/v) ascorbic acid + 2M NaCl | 3.0 | N/A | 1-1-1 |
| | VC high | 5% (w/v) ascorbic acid + 2M NaCl | 3.0 | N/A | 3-24-2 |
| Heat denaturation in MAP-based buffers | MAP1 18-18-2 | 50 mM Tris + 200 mM SDS + 200 mM NaCl | 9.0 | N/A | 18-18-2 |
| | MAP5 18-18-2 | 50 mM Tris + 200 mM SDS + 400 mM NaCl + 20 mM CaCl$_2$ | 9.0 | N/A | 18-18-2 |
| | MAP5 120-18-2 | 50 mM Tris + 200 mM SDS + 400 mM NaCl + 20 mM CaCl$_2$ | 9.0 | N/A | 120-18-2 |
| | MAP5 18-120-2 | 50 mM Tris + 200 mM SDS + 400 mM NaCl + 20 mM CaCl$_2$ | 9.0 | N/A | 18-120-2 |
| | MAP5 18-18-120 | 50 mM Tris + 200 mM SDS + 400 mM NaCl + 20 mM CaCl$_2$ | 9.0 | N/A | 18-18-120 |
| | MAP5 18-18-18 | 50 mM Tris + 200 mM SDS + 400 mM NaCl + 20 mM CaCl$_2$ | 9.0 | N/A | 18-18-18 |
| | MAP5 18-18-2-2 | 50 mM Tris + 200 mM SDS + 400 mM NaCl + 20 mM CaCl$_2$ | 9.0 | N/A | 18-18-2, and 2 hr at 121°C |

\* Multi-day protease treatments are refreshed with newly prepared solutions every day, to partially compensate for loss of enzyme activity over time.

\*\* Refreshment for Trypsin (seq grade) was performed only on Day 1, 3, 5, instead of daily, due to limits on reagent availability.

denaturation treatment in MAP5 buffer (MAP5 18-18-2, abbreviated as 'MAP5' in following texts) and collagenase type II treatment (abbreviated as 'Coll II'), for a more thorough analysis of general epitope stainability, by testing the quality of immunostaining (via the same method as in the initial screen, i.e. estimating the signal-to-noise ratio of the expected pattern of staining) over an expanded panel of 16 antibodies (*Figure 12A*). We found that although these two treatments yielded similar average IHC scores (~0.67) as in the initial screen, MAP5, with which ~6–8 antibodies worked (~40%), moderately outperformed Coll II, with which ~4–5 antibodies worked (~28%).

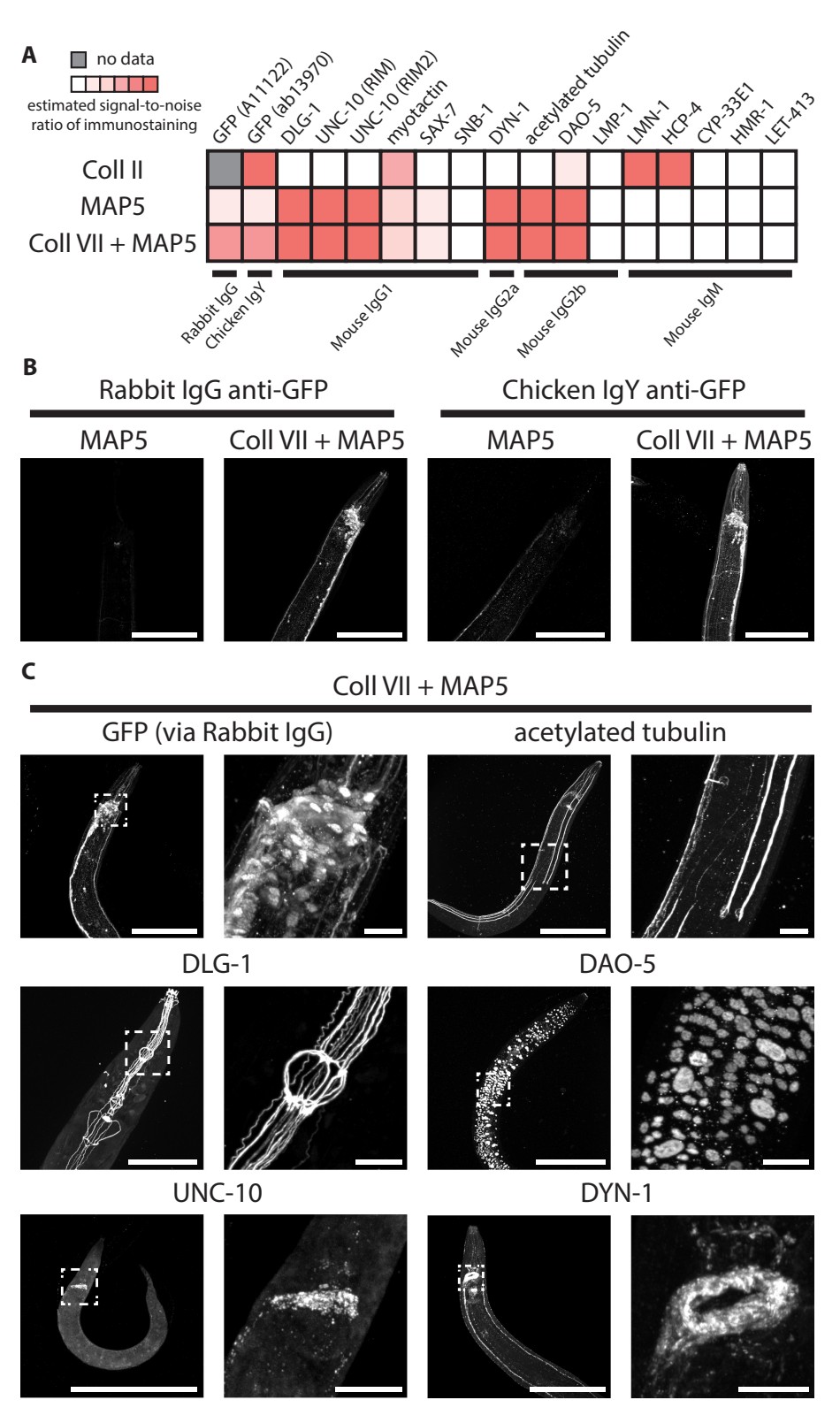

**Figure 12.** Immunohistochemistry after selected post-gelation treatments. (**A**) Estimated signal-to-noise ratio of immunostaining in paraformaldehyde-fixed, β-mercaptoethanol-reduced, AcX-treated, and hydrogel-embedded hermaphrodite animals, where the immunostaining was performed after the gel-embedded sample was processed with either a collagenase type II treatment (Coll II; 24 hr at 37°C), a denaturation treatment in MAP5 buffer (MAP5; 18 hr at 37°C + 18 hr at 70°C + 2 hr at 95°C), or the same denaturation treatment but additionally preceded by a collagenase type VII treatment

*Figure 12 continued on next page*

*Figure 12 continued*

(Coll VII + MAP5; the collagenase VII treatment was performed for 24 hr at 37°C). See Methods for detailed descriptions of each treatment. Top labels, target of the antibody; bottom labels, class of the antibody. Signal intensities were manually scored from confocal stacks acquired from at least three animals from one or more gel samples. (B) Representative images of anti-GFP staining on paraformaldehyde-fixed, β-mercaptoethanol-reduced, AcX-treated, and hydrogel-embedded hermaphrodite animals, where the immunostaining was performed after the gel-embedded sample was processed with either a denaturation treatment in MAP5 buffer (MAP5; 18 hr at 37°C + 18 hr at 70°C + 2 hr at 95°C), or the same denaturation treatment but additionally preceded by a collagenase type VII treatment (Coll VII + MAP5; the collagenase VII treatment was performed for 24 hr at 37°C). The strain used had pan-neuronal cytosolic expression of GFP (*tag-168p::GFP*). Images are max-intensity projections of confocal stacks acquired through the entire animal. Brightness and contrast settings: Coll VII + MAP5 images (right), individually set by the automatic adjustment function in Fiji; MAP5 images (left), have the same settings as the corresponding Coll VII + MAP5 image, to facilitate direct comparison. Linear expansion factors: worm, 1.1–1.3x; surrounding hydrogel, 1.9–2.1x. Scale bars: 100 µm (in biological units, i.e. post-expansion lengths are divided by the expansion factor of the worm). (C) Representative images of immunostaining on paraformaldehyde-fixed, β-mercaptoethanol-reduced, AcX-treated, and hydrogel-embedded hermaphrodite animals, where the immunostaining was performed after the gel-embedded sample was processed first with a collagenase type VII treatment (24 hr at 37°C) and then with a denaturation treatment in MAP5 buffer (18 hr at 37°C + 18 hr at 70°C + 2 hr at 95°C) (Coll VII + MAP5). The strain used had pan-neuronal cytosolic expression of GFP (*tag-168p::GFP*). Applied antibodies were against (upper left) GFP, (upper right) acetylated tubulin, a marker of touch-receptor neurons, (middle left) DLG-1, a scaffolding protein that localizes to adherens junctions, (middle right) DAO-5, a nuclear protein, (lower left) UNC-10, a homolog of the vertebrate Rim protein that involves in synaptic vesicle release, and (lower right) DYN-1, a dynamin, which is involved in clathrin-mediated endocytosis. Images are max-intensity projections of confocal stacks acquired through the entire animal. Right images are magnified views of the boxed regions in the left images. Brightness and contrast settings: first set by the automatic adjustment function in Fiji, and then manually adjusted (raising the minimum-intensity threshold and lowering the maximum-intensity threshold) to improve contrast for stained structures. Linear expansion factors: worm, 1.1–1.3x; surrounding hydrogel, 1.9–2.1x. Scale bars: left images, 100 µm; right images, 10 µm (in biological units, i.e. post-expansion lengths are divided by the expansion factor of the worm).

Two groups of antibodies failed to stain well after the MAP5 treatment. The first group was antibodies against GFP, which resulted in dim and uneven (i.e. consistently brighter around the vulva but dimmer in the head and tail) signal across the animal (*Figure 12B*). We hypothesized that as MAP5 functions through protein denaturation, which does not specifically degrade the cuticle, the rate of antibody diffusion across the cuticle could limit the effective concentration of antibody that reaches the internal tissue, which could preferentially affect staining of high-copy-number targets, such as GFP expressed at a high level. We thus attempted to further permeabilize the cuticle by preceding the MAP5 treatment by a collagenase type VII pre-treatment (abbreviated as 'Coll VII'); this collagenase is a chromatography-purified collagenase that removes a number of non-specific proteases found in other collagenase preparations, such as Coll II. Indeed, anti-GFP staining became stronger and more uniform across animals, for both of the GFP antibodies that we tested (*Figure 12B*), and without affecting the quality of staining for other epitopes (*Figure 12A*; compare second and third rows).

The second group that failed to stain after MAP5 treatment was antibodies of the IgM class, which yielded nearly no signal inside the worm tissue, for all 5 of the IgM-class antibodies tested. As discussed above, MAP5 does not specifically degrade the cuticle. Since IgM-class antibodies are pentamers of 180-kDa-sized subunits (i.e. ~900 kDa total, compared to the ~150 kDa IgG-class antibodies), their inability to diffuse across the relatively under-permeabilized cuticle following MAP5 could explain why internal tissues were never stained, whereas collagenase type II achieved staining with a few (2/5) IgM antibodies, likely because of the ability of collagenase II to digest the cuticle. We observed that while the Coll VII pre-treatment improved anti-GFP staining, it did not rescue the stainability of epitopes detected by IgM-class antibodies, which suggests that the Coll VII pre-treatment does not permeabilize the cuticle as extensively as Coll II does (but critically preserves many antigens that Coll II failed to preserve; compare rows 1 and 3 in *Figure 12A*).

In summary, the combination of Coll VII and MAP5 treatments resulted in strong signal-to-noise ratios and expected patterns of localization for many epitope targets (*Figure 12C*), such as GFP (expressed pan-neuronally in the cytosol), acetylated tubulin (expressed specifically in touch-receptor neurons), DLG-1 (adherens junction marker), DAO-5 (a nuclear protein), UNC-10 (a GTPase for synaptic vesicle release; localized to synapse-dense regions such as the nerve ring) and DYN-1 (a dynamin, for clathrin-mediated endocytosis; also localized to synapse-dense regions) (*Siddiqui et al., 1989*; *Hadwiger et al., 2010*). The Coll VII + MAP5 treatment achieved high signal-to-noise ratios of staining for ~70% (~8 out of 11) of epitopes that we attempted to detect using non-IgM class antibodies (including IgG, the most prevalent class that accounts for >95% of all commercially available

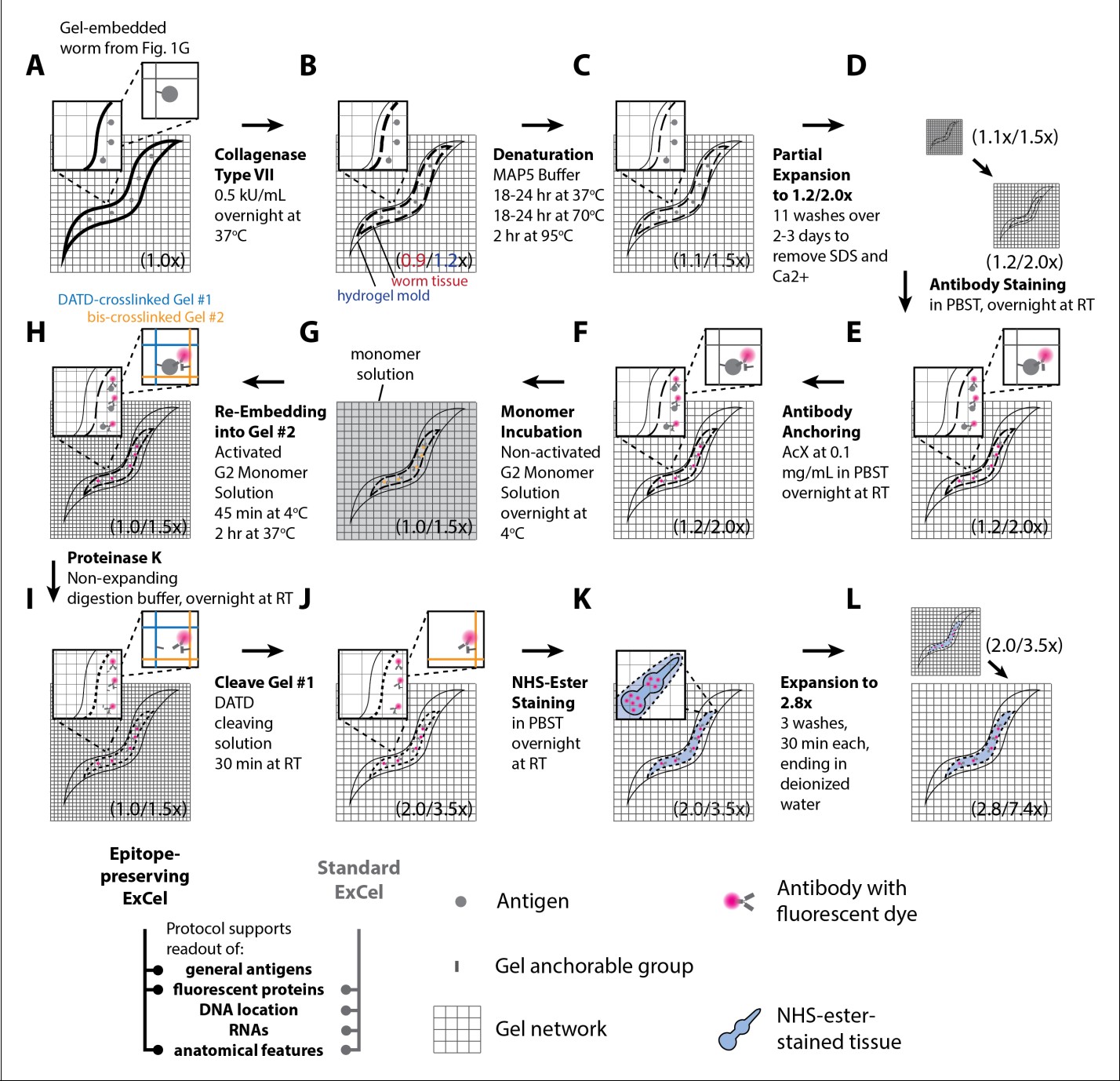

antibodies). Thus, the Coll VII/MAP5 dual treatment enables immunostaining of general epitopes in hydrogel-embedded, morphology-intact whole animals of *C. elegans*.

Having established the Coll VII/MAP5 treatment as good for epitope preservation, we next incorporated strategies from iterative expansion microscopy (*Chang et al., 2017*) to boost the expansion factor, resulting in an epitope-preserving ExCel protocol (schematized in *Figure 13*) capable of achieving expansion factors comparable to those of the standard ExCel protocol (*Figure 1*). In this protocol, we first use the same procedures as in the standard ExCel protocol to prepare hydrogel-embedded worms (*Figure 1A–C,E–G*). Then, instead of the Proteinase K treatment, we apply Coll VII and MAP5 treatments (*Figure 13B–C*). As in standard ExCel, we employ a strategy to minimize premature sample expansion via adding calcium ions during mechanical homogenization; the collagenase type VII treatment is performed in a calcium-containing buffer (100 mM Tris pH 8.0, 500 mM NaCl, 40 mM $CaCl_2$). Despite the use of calcium, small mismatches between the expansion factors of the worm and surrounding gel (which are ~0.9x and ~1.2x, respectively) can still cause the worm to detach from, but physically stay within, its own cast in the hydrogel (as illustrated in *Figure 13B*). We speculate that the divalent calcium ions confer their sample-shrinking effect by crosslinking, via ionic interaction, polymers with repeating negative charges, including the polyacrylate network that constitutes the hydrogel, and perhaps the phosphate backbone of nucleic acids. Thus, a potential explanation for why the gel-embedded worm tissue is more responsive to the shrinking effect of the calcium-containing buffer, than the surrounding pure hydrogel, is that the worm tissue additionally contains negatively charged biological polymers such as DNA and RNAs, which could potentially get retained within the hydrogel even without explicit anchoring by LabelX (especially for chromosomal DNA, whose molecular weight would likely prevent it from free diffusion in the hydrogel network). After collagenase treatment, we perform denaturation (*Figure 13C*) for 18–24 hr at 37°C, 18–24 hr at 70°C, and then 2 hr at 95°C, in MAP5 buffer, again with high salt to minimize premature hydrogel expansion. NaCl and $CaCl_2$ concentrations in MAP5 buffer are lower (400 mM and 20 mM, respectively) compared to those in the Proteinase K or Collagenase VII buffers (500 mM and 40 mM, respectively), due to the limited solubility of these ions in the presence of 200 mM SDS, a key component of the denaturation buffer.

Following the MAP5 treatment, we treat the samples by a total of 11 washes (*Figure 13D*) to completely remove SDS, which could in principle disrupt downstream immunostaining, and calcium ions, which can precipitate out with phosphate ions present in staining buffer (performed in 1x PBS + 0.1% Triton X-100, as described next) and antibody stock solutions (because commercially available antibody stocks are commonly distributed in 1x PBS). Specifically, samples are first washed four times with a buffer formulated similar to the denaturation buffer, but without the SDS component (50 mM Tris pH 8, 400 mM NaCl, 20 mM $CaCl_2$), to remove SDS from the sample. Next, the samples are washed four times in buffers with reduced (10 mM, for the first wash) or no calcium ions (for the next three washes) to remove calcium ions from the sample. Then, the samples are washed in phosphate-based buffers with reducing salt concentration, ending with PBST (1x PBS + 0.1% Triton X-100).

At this stage, the samples are immunostained against the target epitopes, with standard primary and fluorophore-conjugated secondary antibodies, in PBST (*Figure 13E*). As we re-embed the samples into another hydrogel, in a step later in the protocol, to boost the net expansion factor, the fluorophores on the secondary antibodies must be able to survive the free-radical-mediated hydrogel polymerization process. Many fluorophores that fit such criterion were described in the proExM study (*Tillberg et al., 2016*), including common fluorescent dyes such as Alexa 488, Alexa 546, and Atto 647N. After immunostaining, samples are treated with AcX (*Figure 13F*), with the goal of adding a gel-anchorable moiety to the fluorophore-conjugated antibodies.

For the second expandable gel, in which we re-embed our samples for extra physical magnification, we designed a second-gel ('G2') monomer solution (50 mM MOPS pH 7.0, 0.15 M NaCl, 40 mM $CaCl_2$, 7.5% (w/w) sodium acrylate, 2.5% (w/w) acrylamide, 0.15% (w/w) N,N'-methylenebisacrylamide), whose formulation is similar to the monomer solution used in the first gelation, but replacing the gel crosslinker from the periodate-cleavable DATD with the periodate-resistant N,N'-methylenebisacrylamide (abbreviated as 'bis'). To perform the re-embedding, we use a scheme similar to how the initial gelation was performed (as in *Figure 1F–G*): samples are first pre-incubated

**Figure 13.** Workflow for epitope-preserving expansion of *C. elegans* (epitope-preserving ExCel) sample processing. A method for expanding cuticle-enclosed intact *C. elegans*, while permitting readout of a majority of antigens that are detectable through non-IgM-class antibodies (~70%; estimated from the immunostaining results from the panel of 12 non-IgM antibodies in *Figure 12A*). Sample processing prior to Panel A is identical to the workflow for the standard ExCel protocol without ExFISH (as shown in blue arrows in *Figure 1*) until, and including, the gelation step (*Figure 1A–C, E–G*). The linear expansion factor of the hydrogel-specimen composite is shown in parentheses. For stages in which the worm tissue expands to a less extent than the surrounding hydrogel, which occurs due to incomplete homogenization of mechanical strength of the fixed worm tissue, the expansion factors of the worm and of the hydrogel are shown in front of and after a slash sign, respectively. (A–L) Steps of the protocol, with the bold text indicating the title of the step. (A) Hydrogel polymerization is performed on the specimen, by first incubating the specimens in activated monomer solution (0.015% 4-hydroxy-TEMPO, 0.2% TEMED, 0.2% APS, in addition to the non-activated monomer solution) for 1 hr at 4°C, transferring the specimens into a gelation chamber, and incubating the chamber for 2 hr at 37°C. During polymerization, AcX-modified proteins are covalently anchored to the hydrogel network. (B) Specimens are treated with chromatography-purified collagenase type VII at 0.5 kU/mL, in a calcium-containing tris-buffered saline (100 mM Tris pH 8.0, 500 mM NaCl, 40 mM CaCl₂) overnight (18–24 hr) at 37°C. During this treatment, the hydrogel expands by ~1.2x

*Figure 13 continued on next page*

*Figure 13 continued*

linearly, whereas the worm slightly reduces in size to ~0.9x linearly. Due to the mismatch in expansion factor between the worm and the gel, the worm tissue detaches from the surrounding hydrogel, but physically remains in the hydrogel mold that was made of its own shape during the gelation step in A. (C) Specimens are processed with a denaturation treatment, in which they are incubated in a minimally-expanding protein-denaturing buffer (MAP5 buffer; 50 mM Tris pH 9.0, 5.72% (w/w) sodium dodecyl sulfate (SDS), 400 mM NaCl, 20 mM $CaCl_2$) overnight (18–24 hr) at 37°C, overnight (18–24 hr) at 70°C, and 2 hr at 95°C. Reduced calcium and NaCl concentrations are used in this buffer, compared to other non-expanding buffers designed in this paper, due to their incompatible solubility with SDS at higher concentrations. (D) Specimens are washed four times in a tris-buffered saline (TNC40020 (acronyms are used in the supplementary protocols in Appendix 1) buffer; 50 mM Tris pH 8.0, 400 mM NaCl, 20 mM $CaCl_2$) to remove SDS from the hydrogel sample. Specimens are then washed four times in tris buffered saline with reducing calcium concentration (once with TNT Buffer + 10 mM $CaCl_2$, and then three times with TNT Buffer; TNT Buffer is 50 mM Tris pH 8.0, 1M NaCl, 0.1% Triton X-100) to remove calcium ions from the hydrogel sample. Finally, specimens are washed with phosphate-buffer saline with reducing NaCl concentration (once with PBST + 500 mM NaCl, twice with PBST; PBST is 1x PBS + 0.1% Triton X-100). (E) Specimens are immunostained with fluorescent antibodies against the target antigens. (F) Specimens are incubated with AcX at a concentration of 0.1 mg/mL in PBST (1x PBS + 0.1% Triton X-100) overnight at RT. This step equips proteins, including the fluorescent antibodies introduced in E, with a polymer-anchorable moiety. (G) Specimens are incubated in non-activated G2 monomer solution (50 mM MOPS pH 7.0, 2 M NaCl, 7.5% (w/w) sodium acrylate, 2.5% (w/w) acrylamide, 0.15% (w/w) N,N'-methylene-bis-acrylamide) overnight at 4°C, to ensure complete diffusion of the monomer solution throughout the specimen, prior to the gelation reaction. (H) Specimens are re-embedded into a second expandable hydrogel, by incubating the specimens in activated monomer solution (0.015% 4-hydroxy-TEMPO, 0.2% TEMED, 0.2% APS, in addition to the non-activated monomer solution) for 45 min at 4°C, transferring the specimens into a gelation chamber, and incubating the chamber for 2 hr at 37° C. During polymerization, AcX-modified fluorescent antibodies are covalently anchored to the hydrogel network of the second hydrogel (orange grids). We use blue grids to represent the hydrogel network of the first, DATD-crosslinked hydrogel (i.e., the network synthesized in Panel A), to differentiate it from the network of the re-embedding second hydrogel. (I) Specimens are treated with Proteinase K at 8 U/mL, in non-expanding digestion buffer (50 mM Tris pH 8.0, 500 mM NaCl, 40 mM $CaCl_2$, 0.1% Triton X-100) overnight (18–24 hr) at RT, to further reduce the mechanical strength of the original worm tissue and permit greater expansion. During this proteolytic treatment, most proteins lose antigenicity, but some of the fluorescent signals from AcX-anchored fluorescent proteins are retained, as utilized by the original ProExM protocol. (J) Specimens are treated with DATD cleaving solution (20 mM sodium meta-periodate in 1x PBS, pH 5.5) for 30 min at RT, to chemically disintegrate the first hydrogel, which contains the periodate-cleavable crosslinker N,N'-diallyl-tartardiamide (DATD), while sparing the second hydrogel, which contains a periodate-resistant crosslinker, N,N'-methylene-bis-acrylamide (bis). (K) To visualize anatomical features, specimens can be stained with an N-hydroxysuccinimide ester (NHS ester) of fluorescent dye (introduced in Main Text, *Figure 4*, *Videos 1* and *2*). NHS-ester staining is performed at 5 µM in PBST (1x PBS + 0.1% Triton X-100) overnight at RT. DAPI staining can be applied at this stage, but the result does not correspond to the expected nuclear pattern as observed in *Figures 2*, *5* and *9* (see *Figure 13—figure supplement 1*). (L) Specimens are expanded with one wash in 0.1x PBS and two washes in deionized water. At this stage, the hydrogel expands by ~7.4x linearly, whereas the worm tissue expands by ~2.8x linearly, within a range from 2.5 to 3.5x (median, 2.78x; mean, 2.83x; n = 10 independently processed hydrogels from 2 sets of experiments). After expansion, specimens are ready for imaging.

The online version of this article includes the following figure supplement(s) for figure 13:

**Figure supplement 1.** DAPI staining after epitope-preserving ExCel.

with the non-activated form of the G2 monomer solution overnight (to ensure equilibration of monomer concentration; *Figure 13G*), and then re-embedded with the activated form of the G2 monomer solution (*Figure 13H*). During this step, AcX-modified fluorescent antibodies are covalently linked into this second, bis-crosslinked hydrogel.

We then perform a mild (1 day at RT) Proteinase K treatment to further homogenize the worm tissue and grant greater expandability (*Figure 13I*). We chose to use this version of the Proteinase K treatment, instead of the stronger version used in standard ExCel, because we reasoned that this version has been reported in our earlier proExM study to preserve a fair portion of the fluorescent signal (~50%) from fluorophore-conjugated antibodies (*Tillberg et al., 2016*).

As in iterative expansion microscopy, the second expandable gel could reach its maximal expansion factor after the crosslinker of the first expandable gel was cleaved, thus chemically disintegrating the first gel. Therefore, we treat the samples with sodium periodate (1x PBS pH 5.5, 200 mM sodium periodate; *Figure 13J*), which cleaves the DATD crosslinker that was used to compose the first expandable gel, based on a previously characterized chemistry of cis-diol cleavage by periodate (*Späth and Koblet, 1979*).

At this stage, NHS-ester staining could be applied to enable visualization of general anatomical features (*Figure 13K*), as described in the standard ExCel protocol (*Figure 4*). While DAPI staining could also be applied here, we observed that the staining pattern looked very different from the regular, nucleus-contained pattern (*Figure 13—figure supplement 1*). Specifically, while DAPI staining still seemed to localize to regions near nuclei, staining was no longer spatially restricted to being inside nuclei. In addition, we observed ellipsoidal blobs that localized to the periphery of the nuclei, had length scales of roughly 1/10 – 1/3 of a nucleus, and had brighter intensities than the nuclei

(*Figure 13—figure supplement 1B and D*). These observations are in contrast to DAPI staining with standard ExCel (e.g. *Figures 5* and *9*, *Videos 3* and *4*) or previous protocols for applying DAPI in fixed worm tissue (*Shaham, 2006*). A potential reason for this observation is that the heat-denaturation treatment, which combines non-physiologically high temperature (70–95˚C), alkaline environment (pH 9), and a protein-denaturing agent (i.e. SDS) could have disrupted histone-DNA interactions, causing structural disintegration of nucleosomes. Under this hypothesis, genomic DNA, which was not covalently anchored to the hydrogel network (as no LabelX treatment was applied) would no longer be in its native, tightly packed state, and could potentially unwind, expand in physical size, and spill out from the nucleus (as the nuclear membrane would also be disrupted by the denaturation treatment). Of course, in standard ExCel, Proteinase K treatment did not alter DAPI staining from the expected nuclear-restricted staining profile. Perhaps Proteinase K does not access the histones, which may be sterically protected by DNA, whereas heat and small-molecule denaturants like SDS can access and exert disruptive effects on histones. In future studies, including LabelX to covalently anchor genomic DNA to the hydrogel may enable better preservation of genomic integrity.

Finally, we expand the samples in buffers with reducing salt, ending in deionized water, to fully expand the hydrogel-embedded worms (*Figure 13L*). At this stage, the hydrogel expands by ~7.4x linearly, whereas the worm tissue expands by ~2.8x linearly, within a range from 2.5 to 3.5x (median, 2.78x; mean, 2.83x; n = 10 independently processed hydrogels from 2 sets of experiments). As the expansion factor of the worm tissue determines the effective resolution (which differs from ~250 nm / 3.5 = ~70 nm, to ~250 nm / 2.5 = ~100 nm, accordingly), we hypothesized two potential reasons why such variation could have occurred. First, during the execution of these experiments, we switched the vendor for sodium acrylate (from Millipore Sigma, product #408220, to Santa Cruz Biotechnology, product # sc-236893B), a key ingredient of the monomer solution, because we noticed an abrupt change in quality of the sodium acrylate powder supplied by the original vendor; these differences included changes in the powder texture (which increased distinctly in 'fluffiness'), solubility (e.g. powders no longer dissolved at the stock concentration of 33%, and instead left insoluble, string-like precipitates, which might indicate premature polymerization within the source), and color of solution (from near-complete clearness to moderately yellow). We have since then screened through sodium acrylate powders from various vendors, and developed a strategy to identify, and to stock up on, quality-screened powders, which we describe in detail in Appendix 1 – 'Notes on sodium acrylate quality'. After switching to the second vendor, however, we noticed that the quality of sodium acrylate did not exactly match that of the original batches from the first vendor (i.e., the batches from the first vendor prior to aforementioned quality decline), based on slight but perceivable differences in the intensity of the yellow-ness of the stock solutions, as well as the timing of gelation after monomer activation. Such variations could indicate differences in the effective concentration of sodium acrylate, whose ionic interaction is the mechanism by which the hydrogel expands, and could explain differences in the linear expansion factor.

In epitope-preserving ExCel, tissue homogenization is not complete, as evident from the strong mismatch in the expansion factors of the worm tissue (2.8x) and the hydrogel (7.4x). Since homogenization is incomplete, unintended variations in the parameters of homogenizing treatments (which affect their strengths; e.g. temperature, duration, effective enzyme concentrations) could thus more strongly impact the final expansion factor. Consistent with this hypothesis, we observe less variation in the final-state expansion factor in the standard ExCel protocol (3.1–3.6x, for the re-embedding protocol; 3.8–4.2x, for the protocol without re-embedding), compared to the variation associated with epitope-preserving ExCel (2.5–3.5x). Thus, the greater variability in the final expansion factor of epitope-preserving ExCel could be due to insufficient homogenization, here deliberately pursued so that we can preserve epitopes and fluorophore-conjugated antibodies. We note that thorough homogenization could be potentially feasible with this protocol (and would grant more consistent expansion factor under this hypothesis), if in the future one were to replace the fluorophore-conjugated antibody with a DNA-oligo-conjugated antibody (as utilized in the original expansion microscopy publication [*Chen et al., 2015*]), precisely for the purpose of de-coupling tissue homogenization from signal retention), although this also means added complexity for the end user, who will have to synthesize the DNA-conjugated antibody.

## Validation of epitope-preserving ExCel isotropy

Using a similar analysis to that used for characterizing the isotropy of standard ExCel (as in *Figure 3*), we examined *tag-168p::GFP* animals (which has pan-neuronal GFP expression in the cytosol) at different stages of development, and compared rigidly registered (via scaled rotation) images of the same animal at four stages (*Figure 14A*): (1) pre-expansion, for which the image was acquired right after animals are embedded into the first hydrogel (as in *Figure 13A*); (2) after the embedded sample underwent collagenase type VII digestion, heat denaturation, partial expansion and immunostaining (as in *Figure 13E*; as discussed previously, starting with the collagenase treatment, a slight sample expansion occurs and causes a mismatch in the expansion factors of the worm and the surrounding hydrogel, which in turn results in detachment of the worms); (3) after the immuno-stained sample is re-embedded into the second hydrogel (as in *Figure 13H*); (4) after mild homogenization with the 1 day RT Proteinase K treatment and full expansion in deionized water (as in *Figure 13L*), which corresponds to the state where the final imaging occurs. We observed a noticeable change in the overall body posture between images from the first two stages (pre-expansion, vs. after Coll VII, MAP5, and partial expansion), which could be likely due to the detachment of the worm tissue from the surrounding hydrogel, and release of the worm from external mechanical forces exerted by the gel on the worm. In contrast, the body posture stays relatively consistent through the three later stages (after Coll VII, MAP5 and partial expansion, vs. after re-embedding, vs. after entire protocol), suggesting that the majority of body-posture level distortions occur during the initial stages (i.e. Coll VII, MAP5 and partial expansion) of the epitope-preserving ExCel protocol.

To quantify the overall distortion associated with the entire protocol, we apply non-rigid registration between the pre-expansion and post-entire-protocol images, and calculated the root-mean-square (RMS) error of feature measurements (*Figure 14B*), as performed in *Figure 3B*. We observed that the error over 0 to 100 microns was ~25–32% for L1-L2 animals,~32–34% for L3-L4 animals, and ~34–36% for day 1 and day 2 adults, far greater than the errors from standard ExCel (~1–3% for L1-L2 animals;~2–5% for L3-L4 animals,~3–6% for day 1 and day 2 adults). We hypothesized that most of this error was not due to local distortion of the kind of concern to microscopists, but rather the large changes in overall body posture mentioned above. To isolate body posture distortion from local distortion, we used a built-in image-straightening function from ImageJ to adjust the worm body posture into a straight shape, for both the pre-gelation and post-entire-protocol images (*Figure 14C*). The straightened images help isolate the local, tissue-level distortions across the entire animal. When we now apply non-rigid registration and RMS error quantification on these body-posture-normalized images, we observed that the local, tissue-level error over 0 to 100 microns was ~8–17% for L1-L2 animals,~12–22% for L3-L4 animals, and ~16–25% for day 1 and day 2 adults (*Figure 14D*). These error values are still greater than those of standard ExCel, suggesting that the replacement of the harsh but fully-homogenizing 2 day 37℃ Proteinase K treatment, by milder, epitope- or fluorescent-signal- preserving treatments, comes at a tradeoff of reduced micron-scale isotropy. However, such levels of error could be acceptable for many biological applications that are concerned with the relative organization of proteins, rather than precise length measurements.

## Super-resolution imaging of multiple endogenous proteins with epitope-preserving ExCel

We next explored whether epitope-preserving ExCel could enable multiplexed imaging of endogenous protein targets, at the sub-diffraction-limit resolutions (~90 nm at 2.8x expansion factor) granted by tissue expansion. Following the epitope-preserving ExCel protocol, we immunostained against three protein targets, including DAO-5 (a nuclear protein), DLG-1 (a marker of adherens junctions), and DYN-1 (a marker of clathrin-mediated endocytic sites), applied NHS-ester staining to visualize general anatomical contexts, and then imaged the sample after full expansion in deionized water, which yielded a 3.2x linear expansion factor (*Figure 15*). At this improved resolution, we more clearly observed the contours made by DLG-1, which shows a pattern of longitudinal lines spanning across the entire animal. In addition, we identified the six vulval precursor cells (i.e. P3.p to P8.p) from the DLG-1 staining, which shows six connected rings on the ventral midbody region of the L2 animal (*Figure 15D,E*), a pattern that has been previously characterized (*Pellegrino et al., 2011*). While we observed that DLG-1 staining clearly localizes to the interface between neighboring vulval precursor cells, which has been described before (*Shin and Reiner, 2018*), we additionally

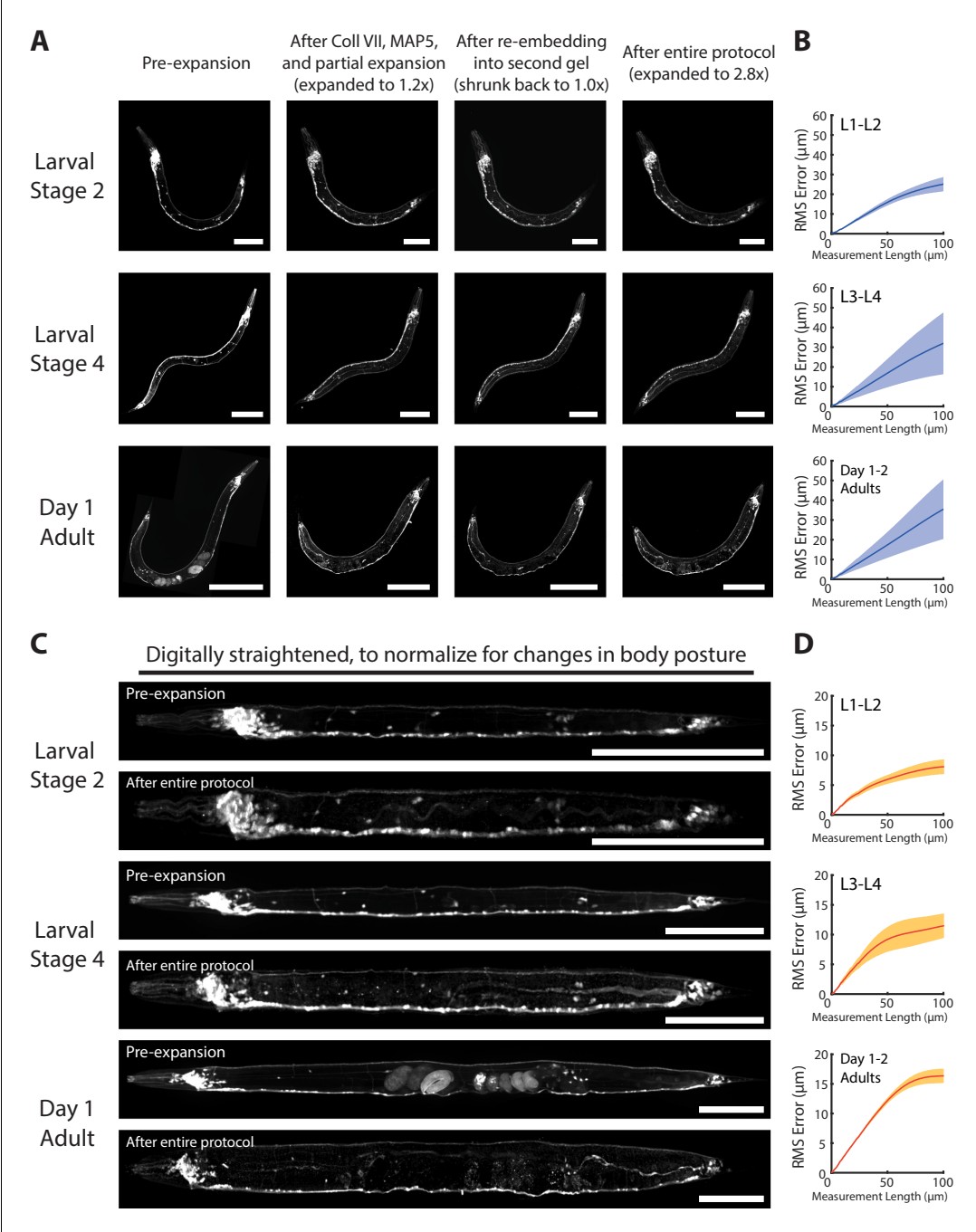

**Figure 14.** Isotropy of epitope-preserving ExCel. (**A**) Representative images of paraformaldehyde-fixed, β-mercaptoethanol-reduced, AcX-treated, and hydrogel-embedded (as in *Figure 1A–C, E–G*) hermaphrodite animals in the second larval stage ('Larval Stage 2'; L2), the fourth larval stage ('Larval Stage 4'; L4) and day 1 adulthood ('Day 1 Adult') at various stages of the epitope-preserving ExCel workflow. These stages include: right after hydrogel embedding ('Pre-expansion'; as in *Figure 13A*), after collagenase type VII digestion, denaturation, partial expansion and anti-GFP staining ('After Coll VII, MAP5, and partial expansion'; as in *Figure 13A–E*), after re-embedding into the second gel ('After re-embedding into second gel'; as in *Figure 13A–H*), or after the entire epitope-preserving ExCel protocol ('After entire protocol'; as in *Figure 13A–L*). Strain expressed *tag-168p::GFP*. Signals in the Pre-expansion images were from native GFP; signals in images from all later stages were from antibody staining against GFP. Images are max-intensity projections of confocal stacks acquired through the entire animal. Brightness and contrast settings: pre- and post-ExCel images (left and center), first individually set by the automatic adjustment function in Fiji, and then manually adjusted (raising the minimum-intensity threshold and lowering the maximum-intensity threshold) to improve contrast. Linear expansion factors (of the worm and of the surrounding hydrogel, shown before and after the slash sign): pre-expansion (first column), 1.0x/1.0x; after Coll VII, MAP5 and partial expansion (second column), 1.1–1.2x/2.2–2.3x; after re-embedding into second gel (third column), 0.9–1.1x/1.4–1.5x; after entire protocol (fourth column), 2.5–2.8x/7.4–7.5x. Scale bars: L2, 50 μm; L4, 100 μm;

*Figure 14 continued on next page*

*Figure 14 continued*

day 1 adult, 200 µm (in biological units, i.e. post-expansion lengths are divided by the expansion factor of the worm). (B) Root-mean-square length measurement error ('RMS Error') computed from pre-expansion (first column) and post-entire-protocol (fourth column) images, as acquired in A, for L1-L2 larvae (top), L3-L4 larvae (middle), and day 1 – day 2 adults (bottom). Blue line, mean; shaded area, standard deviation. n = 3 animals for each age group, from a single mixed-age population processed on a single hydrogel sample. Source data of the RMS length measurement errors are available in *Figure 14—source data 1*. (C) Animals shown in A, but digitally straightened, by performing the ImageJ function Straighten on manually selected spline control points along the body midline of each worm. Images show animals right after hydrogel embedding expansion (top; 'Pre-expansion', as in *Figure 13A*) or after the entire protocol (bottom; 'After entire protocol', as in *Figure 13A–L*). Scale bars: 100 µm. (D) Same analysis as in B, but performed on digitally straightened animals as in C. Orange line, mean; shaded area, standard deviation. n = 3 animals for each age group, from a single mixed-age population processed on a single hydrogel sample. Source data of the RMS length measurement errors are available in *Figure 14—source data 2*.

The online version of this article includes the following source data for figure 14:

**Source data 1.** Root-mean-square (RMS) length measurement errors plotted in *Figure 14B*.
**Source data 2.** Root-mean-square (RMS) length measurement errors plotted in *Figure 14D*.

---

detected that DYN-1 co-localizes with DLG-1 at these cell-cell contact sites (*Figure 15F*), which has not been directly reported in the past. Thus, epitope-preserving ExCel enables multiplexed readout of endogenous proteins at an effective resolution of ~90 nm, and demonstrates sufficient sensitivity to reveal a previously uncharacterized pattern of localization for a protein target.

## Super-resolution imaging of synaptic active and peri-active zone proteins with epitope-preserving ExCel

We next explored whether epitope-preserving ExCel could be used to study the spatial organization of endogenous proteins within the pre-synaptic side of a chemical synapse. The general structure of the pre-synaptic density has been described for multiple species (including vertebrates, *Drosophila*, and *C. elegans*), and consists of an active zone, which is the primary site of synaptic vesicle fusion and release of neurotransmitters, and the surrounding peri-active zone, which is functionally associated with endocytosis (to replenish the synaptic vesicle pool) and trans-synaptic cell-cell anchoring (*Südhof, 2012*; *Wahl et al., 2013*; *Gross and Von Gersdorff, 2016*). We selected two protein targets, UNC-10 and DYN-1, as putative markers for a subset of active and peri-active zones in *C. elegans*, respectively. UNC-10, which is a homolog of the vertebrate RIM protein that contributes to synaptic vesicle fusion, has been characterized as an active zone component in *C. elegans* and other organisms (*Wang et al., 1997*; *Stigloher et al., 2011*). DYN-1, a homolog of vertebrate dynamin, is a putative marker for peri-active zones in *C. elegans*, because dynamin has been characterized to localize to peri-active zones in neuromuscular junctions of fruit flies (*Marie et al., 2004*), and also because peri-active zones have been described as sites of synaptic vesicle endocytosis, a process mediated by dynamins (*Raimondi et al., 2011*).

To visualize the spatial organization of these putative active and peri-active zone proteins, we followed the epitope-preserving ExCel protocol to immuno-stain against DYN-1, UNC-10, and GFP (which is expressed pan-neuronally in the cytosol, under *tag-168p::GFP*). We then applied NHS-ester staining to visualize the general anatomy of the worm, and imaged the sample after full expansion in deionized water, which yielded a 3.5x linear expansion factor (*Figure 16A,B*).

We observed that both DYN-1 and UNC-10 densely localized to the nerve ring and the nerve cords, a pattern that is consistent with pre-re-embedded images (*Figure 12C*), but we additionally observed, at this post-expansion resolution, that DYN-1 and UNC-10 occupy spatially distinct sites (*Figure 16C*), with DYN-1 (the putative peri-active zone marker) showing a more diffuse localization than UNC-10 (the putative active zone marker), which forms sharper puncta but occupies less volume spatially. To better inspect the organization of these proteins at individual synapses, we imaged pharyngeal motor neuron projections along the outer surface of pharyngeal muscles (*Figure 16D*), and observed distinct UNC-10 puncta, which are inter-spaced from one another at similar distances. The UNC-10 puncta are spatially embedded amidst DYN-1 densities, which is reminiscent of observations made in other organisms, including lamprey, fruit fly and mouse, that the active zones are spatially surrounded by peri-active zones (*Marie et al., 2004*; *Brodin and Shupliakov, 2006*; *Wahl et al., 2013*). At this resolution, we also noticed that for synapses at the frontal end of the

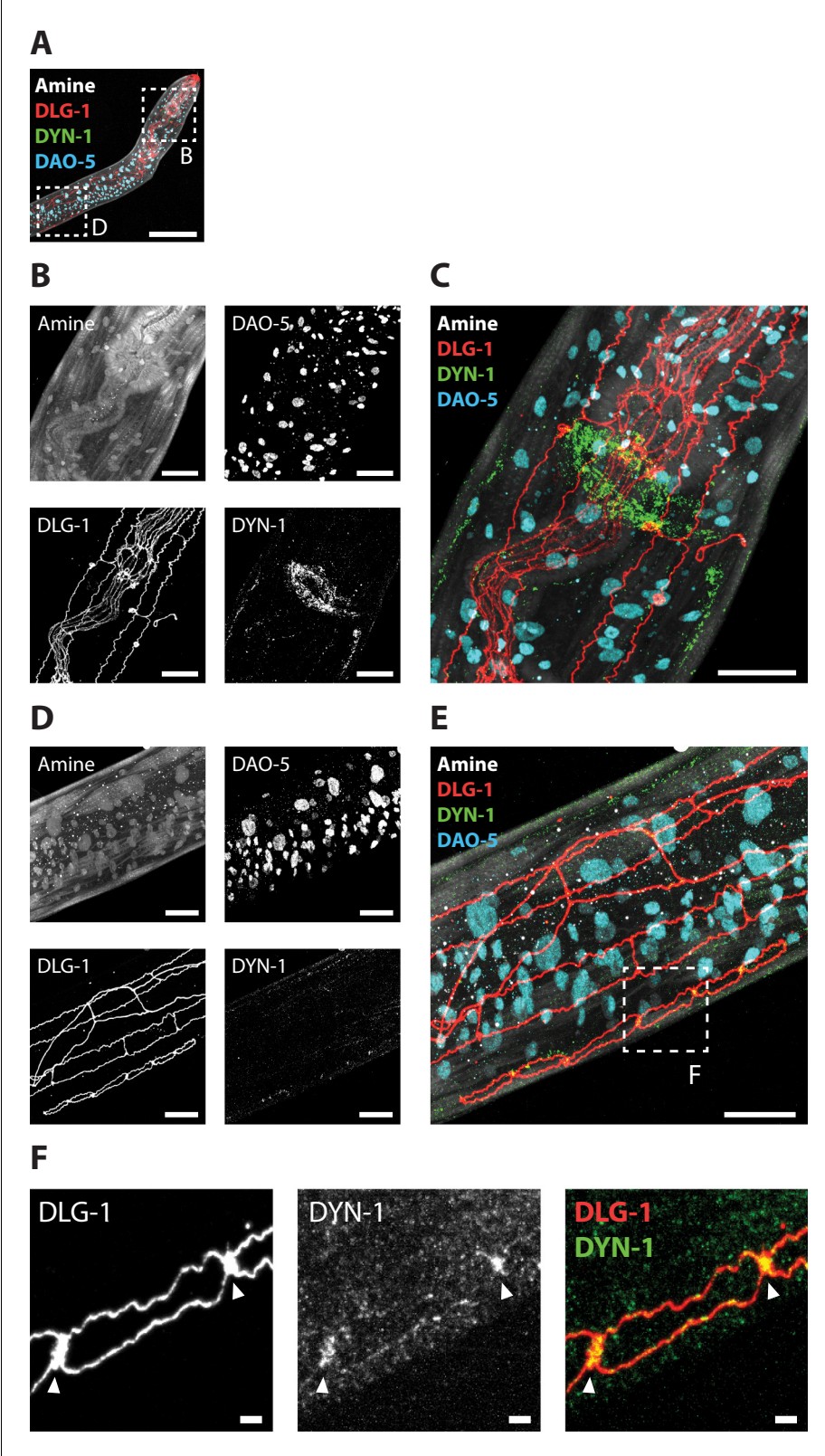

**Figure 15.** Epitope-preserving ExCel allows multiplexed imaging of endogenous proteins at nanoscale resolution. A representative epitope-preserving-ExCel-processed (formaldehyde-fixed, β-mercaptoethanol-reduced, AcX-treated, hydrogel-embedded, collagenase-digested, denaturation-processed, antibody-stained, re-embedded and fully expanded; as in *Figure 13A–L*) L2 hermaphrodite animal, stained with antibodies against DLG-1 (disc large; a scaffolding protein that localizes to adherens junctions), DYN-1 (dynamin; localizes to clathrin-mediated endocytic sites), DAO-5 (a nuclear protein) and

*Figure 15 continued on next page*

*Figure 15 continued*

an NHS ester of a fluorescent dye (Alexa 405 NHS ester; against amines; for anatomical features). (**A**) Merged composite image of the upper body, from the four staining modalities. Boxed regions mark the nerve ring and the developing vulva, and are shown in magnified views in Panels B and D, respectively. (**B**) Single-channel images of each staining modality, centered at the nerve ring (upper boxed region in A). (**C**) Merged composite image from combining images in B. (**D**) Single-channel images of each staining modality, centered nearby the developing vulva (lower boxed region in A). (**E**) Merged composite image from combining images in D. Boxed region marks one of the six vulval progenitor cells, as delineated by the adherens junction marker DLG-1, and is shown in magnified views in Panel F. (**F**) Magnified view of the boxed region in E, as single-channel images of DLG-1 (left) or DYN-1 (middle) staining, or merged composite image between these two channels (right). Arrows, sites of contact between vulval progenitor cells. Images are max-intensity projection of a confocal stack acquired through the entire animal (for Panels A-E), or only through the DLG-1 marked structure (for Panel F; to reduce the noise coming from planes outside of the structure of interest; i.e. the DLG-1 marked adherens junctions of the developing vulva). Brightness and contrast settings: each channel was first set by the automatic adjustment function in Fiji, and then manually adjusted (raising the minimum-intensity threshold and lowering the maximum-intensity threshold) to improve contrast. Linear expansion factor: worm, 3.2x; surrounding hydrogel, 7.9x. Scale bars: Panel A, 50 µm; Panels B-E, 10 µm; Panel E, 1 µm (in biological units, i.e. post-expansion lengths are divided by the expansion factor of the worm).

dorsal nerve cord, UNC-10 exclusively localizes to the ventral side (i.e. facing the internal tissue) of DYN-1, by distances of less than 0.5 micron (*Figure 16E*). Such an exclusive directional relationship is reminiscent of the spatial organization between body wall muscle arms and the dorsal cord neurons, in which the muscle arms, i.e. projections from the body wall muscles, exclusively attach to the ventral side of the neurons to receive synaptic input, as previously characterized by electron microscopy (*Altun and Hall, 2012*). These synapses could therefore correspond to the neuromuscular junctions by which the dorsal cord neurons innervate the body wall muscles. Thus, epitope-preserving ExCel achieves the detection of, and the analysis of nanometer-scale spatial relationships between, untagged endogenous proteins within a chemical pre-synapse.

## Design of the iterative ExCel protocol

The standard ExCel protocol achieves an effective resolution of ~65–75 nm with a 3.3–3.8x linear expansion factor, whereas the epitope-preserving ExCel protocol achieves an effective resolution of ~70–100 nm with a 2.5–3.5x linear expansion factor. These effective resolutions could be potentially improved by additional rounds of hydrogel embedding, cleavage of the previous-round gels, and expansion of the newest gels, as partially demonstrated by the two-gel scheme in the epitope-preserving ExCel protocol, and further demonstrated in by the three-gel scheme in the published iterative expansion (iExM) protocol (*Chang et al., 2017*), which achieved an effective resolution of ~25 nm with a ~20x linear expansion factor.

Thus, we explored whether the standard ExCel protocol could be improved in resolution through integration with the iExM protocol. We designed a protocol named iterative expansion of *C. elegans* (iExCel; schematized in *Figure 17*), which starts with identical sample preparation as the standard ExCel protocol, but followed by a modified protocol to enable a sequential round of sample expansion, based on procedures and strategies that were developed and validated in the original iExM protocol.

In the iExCel protocol, we first prepare hydrogel-embedded, Proteinase-K-digested, and partially expanded samples, following the same procedure as in the standard ExCel protocol (*Figure 1A–C, E–G*). Next, instead of fluorescent-dye-conjugated antibodies, we use DNA-oligo-conjugated antibodies to stain for fluorescent proteins in the sample (*Figure 17B*, recall that with the standard ExCel protocol, only protease-resistant proteins such as fluorescent proteins can be stained and detected, because of the Proteinase K treatment; this characteristic is inherited by this iExCel protocol). These antibodies were custom synthesized using commercially available reagents (see conjugation protocol in Appendix 1 – 'Protocol for synthesizing DNA-conjugated Secondary Antibody for iExCel', which is essentially the same as previously published protocols for synthesizing oligo-conjugated antibodies for ExM purposes [*Chen et al., 2015*]). They contain a 24-base DNA oligonucleotide that carries an unreacted acrydite group, which is a gel-anchorable moiety that later allows these DNA oligos to be covalently linked into the next hydrogel.

After immunostaining, we fully expand the sample to a 3.8x linear expansion factor (*Figure 17C*), and re-embed the sample into another non-expanding hydrogel (*Figure 17D*), which maintains the expanded state of the first hydrogel, even when the re-embedded sample is later immersed into a

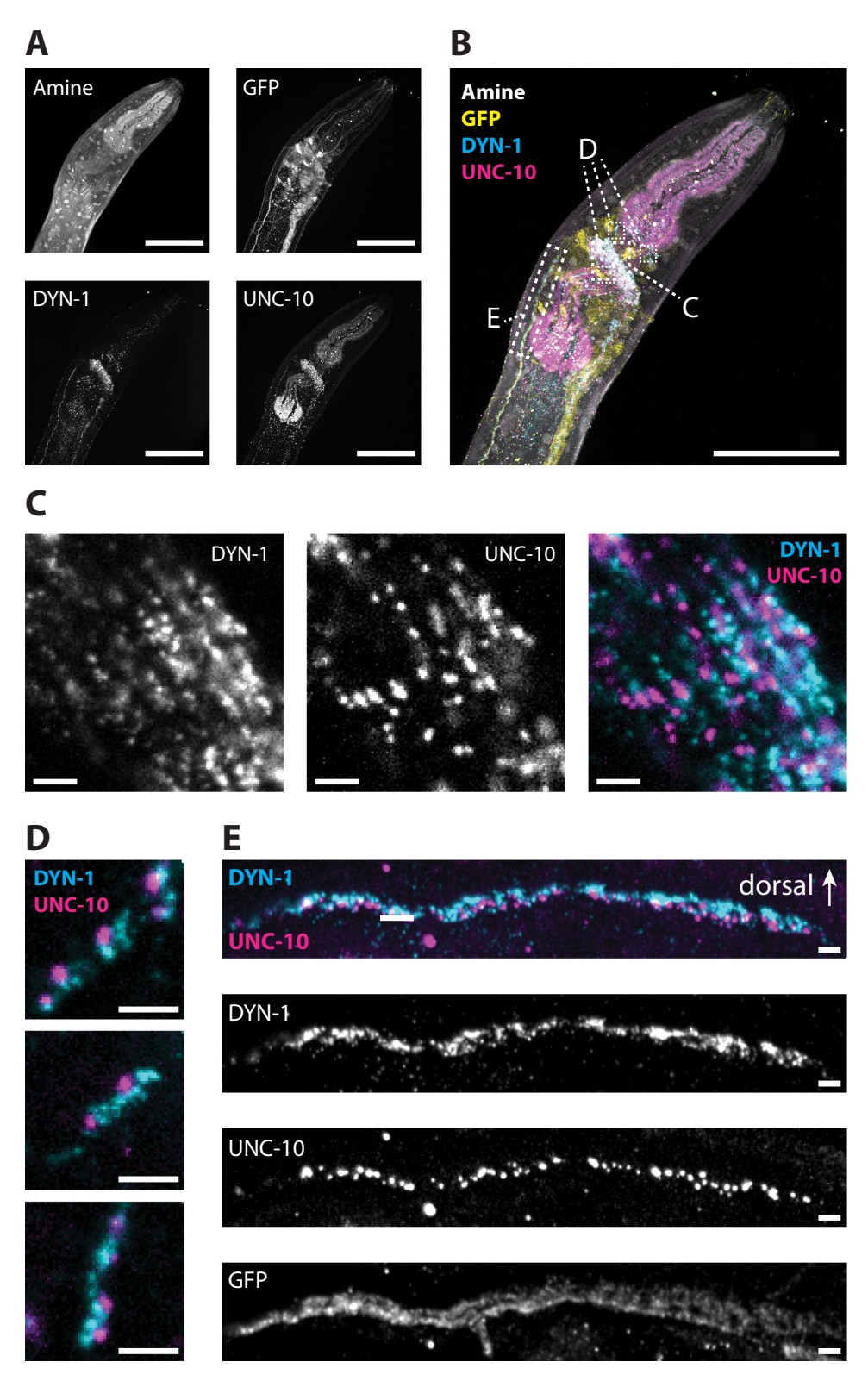

**Figure 16.** Super-resolution imaging of pre-synaptic active and peri-active zones with epitope-preserving ExCel. A representative epitope-preserving-ExCel-processed (formaldehyde-fixed, β-mercaptoethanol-reduced, AcX-treated, hydrogel-embedded, collagenase-digested, denaturation-processed, antibody-stained, re-embedded and fully expanded; as in *Figure 13A–L*) L2 hermaphrodite animal, stained with antibodies against GFP (the strain used had pan-neuronal cytosolic expression of GFP, under *tag168p::GFP*), DYN-1 (dynamin; localizes to clathrin-mediated endocytic sites, which mark

*Figure 16 continued on next page*

*Figure 16 continued*

peri-active zones of pre-synapses), UNC-10 (a homolog of the vertebrate Rim protein, which regulates synaptic vesicle release, and localizes to active zones of pre-synapses) and an NHS ester of a fluorescent dye (Alexa 405 NHS ester; against amines; for anatomical features). (**A**) Single-channel images of each staining modality, centered at the pharyngeal region of the animal. (**B**) Merged composite image from combining images in A. Boxed regions mark parts of the nerve ring, pharyngeal motor neurons, and the dorsal nerve cord, and are shown in magnified views in Panels C, D, and E, respectively. (**C**) Magnified view of a part of the nerve ring (as shown in the boxed region in B), as single-channel images of DYN-1 (left) or UNC-10 (middle) staining, or merged composite image between these two channels (right). (**D**) Magnified view of pharyngeal motor neuron projections (as shown in the boxed regions in B) located along the outer surface of the pharyngeal muscles, shown as merged composite images between the DYN-1 and the UNC-10 images. (**E**) Magnified view of a segment of the dorsal nerve cord (as shown in the rectangular boxed region in B), as single-channel images of DYN-1, UNC-10 and GFP (lower images), and as merged composite image between the DYN-1 and the UNC-10 image (top image). Arrow, the dorsal side of the worm. (For the dorsal nerve cord, the dorsal side is facing the cuticle, whereas the ventral side is facing the internal tissues of the worm.) Images are either max-intensity projections of a confocal stack acquired through the entire animal (for Panels A-B), or only through the thickness of the dorsal nerve cord (for Panel E; to reduce the noise coming from planes outside of the structure of interest), or confocal images at a single z-plane (Panels C-D). Brightness and contrast settings: each channel was first set by the automatic adjustment function in Fiji, and then manually adjusted (raising the minimum-intensity threshold and lowering the maximum-intensity threshold) to improve contrast. Linear expansion factor: worm, 3.5x; surrounding hydrogel, 7.5x. Scale bars: Panel A-B, 30 μm; Panels C-E, 1 μm (in biological units, i.e. post-expansion lengths are divided by the expansion factor of the worm).

high-salt environment downstream in the protocol. This re-embedding process is therefore functionally similar to the re-embedding process used in the standard ExCel protocol with ExFISH (*Figure 1M–N*). Because the re-embedding monomer solution contains some ions, the expansion factor slightly reduces during the re-embedding step, from 3.8x to 3.6x. (Recall that a drop in expansion factor is also observed for the re-embedding step in standard ExCel with ExFISH, from 3.8x to 3.3x, as shown in *Figure 1M–N*. The expansion factor drops by a smaller extent for the iExCel case, because the re-embedding monomer solution used here contains less ions, i.e. by 50 mM less Tris base, than the version used in standard ExCel; see Methods for full recipes). During this re-embedding step, the gel-anchorable acrydite groups on the DNA-conjugated antibodies are covalently anchored to the second hydrogel, thereby retaining their relative spatial positions in the hydrogel network, regardless of downstream treatments that could cause antibody dissociation from the currently-bound targets (e.g. downstream hybridization requires a buffer containing formamide, an organic solvent that could disrupt protein-protein interactions). Lastly, the crosslinker that we use for this second hydrogel is DATD, which is the same as the crosslinker of the initial hydrogel. Such design allows controllable disintegration (via periodate-mediated DATD cleavage, as utilized in epitope-preserving ExCel, i.e. *Figure 13I–J*) of both of these gels following re-embedding of the third (and the final) hydrogel, allowing the final expandable hydrogel to expand maximally.

After re-embedding into the second hydrogel, we hybridize a 100-base DNA oligo, which we refer to as the 'linker', to the hydrogel-anchored 24-base DNA oligo that was introduced via the oligo-conjugated antibodies (*Figure 17E*). This linker oligo contains three components: (1) a region that is complementary to the anchored 24-based oligo, which enables detection of the immunostained locations; (2) four identical regions that are complementary to a fluorescent-dye-conjugated 15-base locked-nucleic-acid (LNA) oligo, which will hybridize to the linker oligo downstream in the protocol, for the final readout; and (3) another unreacted gel-anchorable acrydite group, which allows this linker oligo to be covalently retained in the third hydrogel. During this hybridization step, the re-embedded sample swells by ~30% linearly, and results in a linear expansion factor of ~4.6x. The reason for this additional expansion is that although we refer to the re-embedding second gel as 'non-expanding' (by excluding sodium acrylate, the charged ingredient that confers hydrogel expandability in deionized water, from the recipe of this second gel), this acrylamide-only gel still swells by ~1.3x when immersed in an aqueous solution, a property that has been well-described for polyacrylamide gels (*Baselga et al., 1989*; *Patel et al., 1989*). For the standard ExCel protocol with ExFISH, we also perform re-embedding with an acrylamide-only hydrogel, but did not observe changes in the expansion factor afterwards (which stayed at ~3.3x for all subsequent procedures; *Figure 1N–Q*); this lack of change in the sample expansion factor could be potentially explained by a balance between two effects: the swelling effect of the polyacrylamide re-embedding gel, and the shrinking effect of the high-ionic-strength immersing solution used in ExFISH applications (5x SSC, whose ionic strength is ~1600 mOsm), the latter of which might have been reduced, but not eliminated, by the re-embedding procedure. On the other hand, for the iExCel case, the polyacrylamide-

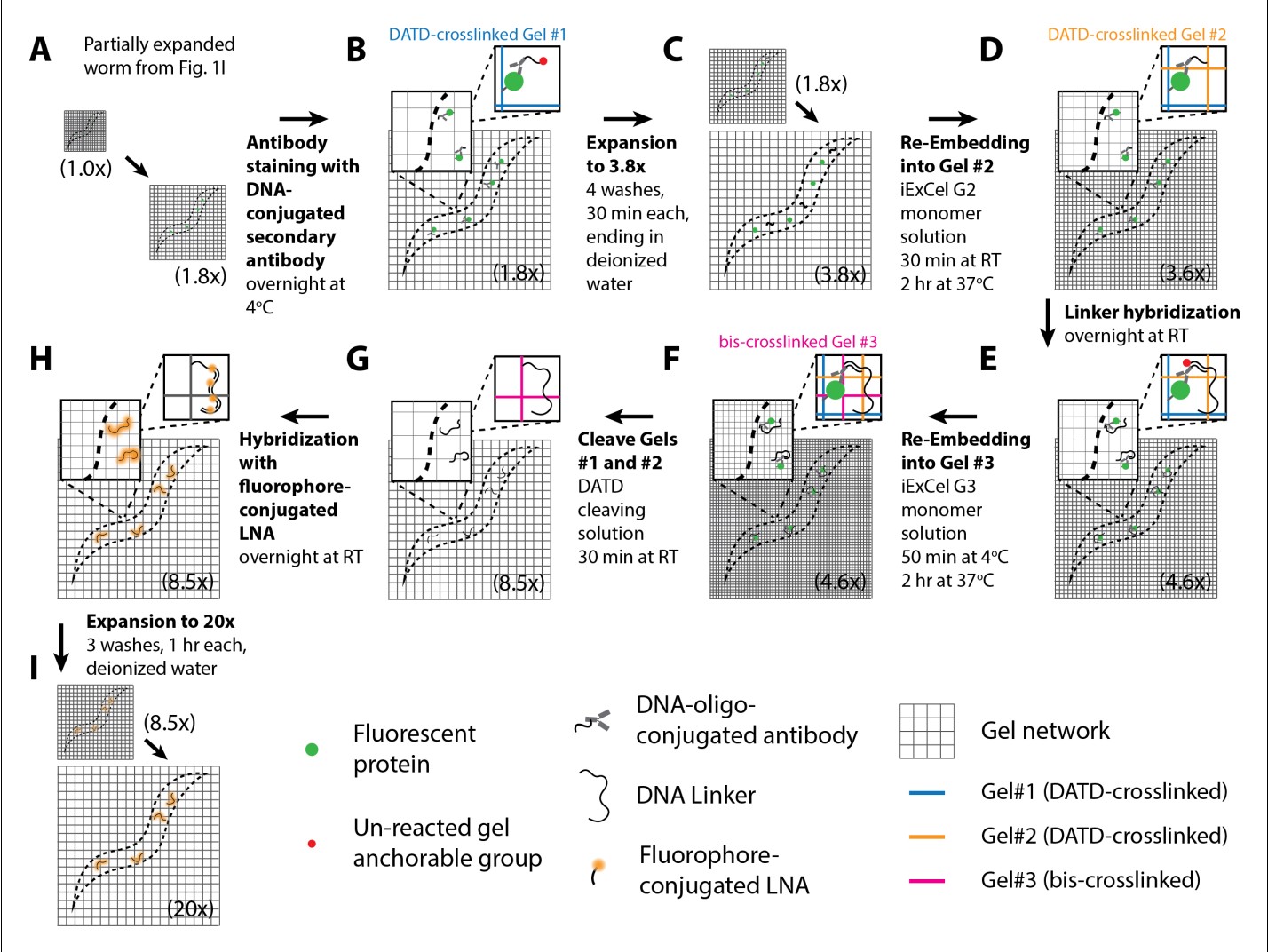

**Figure 17.** Workflow for iterative expansion of *C. elegans* (iExCel) sample processing. A method for iteratively expanding cuticle-enclosed intact *C. elegans*, for a final linear expansion factor of ~20x. Sample processing prior to Panel A is identical to the workflow for the standard ExCel protocol without ExFISH (as shown in blue arrows in *Figure 1*) until, and including, the post-Proteinase-K partial expansion step (*Figure 1A–C, E–G*). The linear expansion factor of the hydrogel-specimen composite is shown in parentheses. (A–I) Steps of the protocol, with the bold text indicating the title of the step. (A) Specimens are partially expanded from a linear expansion factor of 1.0x to 1.8x, with the same protocol as shown in *Figure 1I*. (B) Specimens are immunostained first with primary antibodies against fluorescent proteins in 5x SSCT overnight at 4°C, and then with secondary antibodies that have been conjugated to a 24-base DNA oligonucleotide, in DNA-conjugated Antibody Staining Buffer (2x SSC, 2% (w/v) dextran sulfate, 1 mg/mL yeast tRNA, 5%(v/v) normal donkey serum, 0.1% Triton X-100) overnight at 4°C. The DNA oligo is conjugated to the antibody at the 3' end, and contains a gel anchorable group at the 5' end. (C) Specimens are expanded from a linear expansion factor of 1.8x to 3.8x, with the same protocol as shown in *Figure 1M*. (D) Specimens are re-embedded into another non-expandable hydrogel ('Gel #2') to lock up its size at the expanded state, as shown in *Figure 1N*, except that the monomer solution is replaced by DATD-crosslinked re-embedding monomer solution (10% acrylamide, 1% N,N'-diallyl-tartardiamide (DATD), 0.05% TEMED, 0.05% APS), which results in a hydrogel that can be later disintegrated via crosslinker cleavage, to allow full expansion of the final expandable gel. The DATD-crosslinked re-embedding monomer solution contains a charged molecule APS. Therefore, the linear expansion factor slightly drops from 3.8x to 3.6x during this step. During hydrogel polymerization, the DNA oligo on the antibody, which contains a gel-anchorable group, is covalently anchored to the second hydrogel network (orange grids). (E) Specimens are incubated with a 100-base DNA oligonucleotide ('Linker'), which hybridizes to the 24-base DNA oligo on the secondary antibodies, and which contains a gel anchorable group on its 5' end, in iExCel hybridization buffer (4x SSC, 20% (v/v) formamide) overnight at RT. (F) Specimens are re-embedded into another expandable hydrogel ('Gel #3'), by incubating the specimens in activated Gel #3 monomer solution (1x PBS pH 7.4, 7.5% (w/w) sodium acrylate, 2.5% (w/w) acrylamide, 0.15% (w/w) N,N'-methylene-bis-acrylamide, 2M NaCl, 0.015% 4-hydroxy-TEMPO, 0.2% TEMED, 0.2% APS) for 50 min at 4°C, transferring the specimens into a gelation chamber, and incubating the chamber for 2 hr at 37°C. During polymerization, the linker DNA oligo, which contains a gel-anchorable group, is covalently anchored to the hydrogel network of the third hydrogel (magenta grids). (G) Specimens are treated with DATD cleaving solution (20 mM sodium meta-periodate in 1x PBS, pH 5.5) for 30 min at RT, to chemically disintegrate the first and the second hydrogels, which contain a periodate-

*Figure 17 continued on next page*

*Figure 17 continued*

cleavable crosslinker N,N′-diallyl-tartardiamide (DATD), while sparing the third hydrogel, which contains a periodate-resistant crosslinker N,N′-methylene-bis-acrylamide (bis). (**H**) Specimens are incubated with a fluorophore-conjugated 15-base locked nucleic acid (LNA) oligonucleotide, which hybridizes to the 100-base linker DNA oligo at four locations, in iExCel hybridization buffer (4x SSC, 20% (v/v) formamide) overnight at RT. (**I**) Specimens are expanded to a linear expansion factor of ~20x, with three washes in deionized water. After expansion, specimens are ready for imaging.

gel swelling effect could have overcome the ion-mediated shrinking effect, because the sample was immersed in 1x PBS (a lower-ionic-strength buffer, i.e.,~300 mOsm, than 5x SSC) prior to the measurement of expansion factor, and that the re-embedding hydrogel consists of a greater concentration of acrylamide (4% for standard ExCel; 10% for iExCel), which is expected to be mechanically stronger and more resistant to the shrinking effects of a ion-containing immersing solution. In summary, we observed that the post-re-embedding iExCel sample gains additional expansion (from 3.6x to 4.6x) during the hybridization step, a phenomenon that is not-observed for post-re-embedding samples from the standard ExCel protocol with ExFISH, likely because of differences in the ionic strengths of immersing buffers, and in the compositions of the re-embedding gels.

We then re-embed the linker-hybridized sample into a third (and final) hydrogel (*Figure 17F*). Similar to the first hydrogel, the third hydrogel is expandable, thus conferring two rounds of sample expansion. We designed a third-gel monomer solution that contains the periodate-resistant bis crosslinker, which permits this gel to survive through the periodate-mediated disintegration of the embedded hydrogels, as demonstrated in epitope-preserving ExCel (*Figure 13I–J*). During this re-embedding process, the gel-anchorable acrydite group on the linker oligo is covalently attached to the third hydrogel network.

Following the re-embedding of the third hydrogel, we chemically dis-integrate the first two hydrogels (both of which were constituted with the periodate-sensitive DATD crosslinker) with sodium periodate (*Figure 17G*), using the same protocol as used in epitope-preserving ExCel (*Figure 13I–J*). The sample is then incubated with fluorescent-dye-conjugated 15-base locked nucleic acid (LNA) strands, which hybridizes to the complementary regions on the linker oligos (*Figure 17H*; there are four regions per linker oligo, which allows signal amplification). Importantly, an LNA oligo was used instead of a normal DNA oligo, because of its ability to retain hybridization with the linker molecule even in deionized water, i.e. the immersing medium that is required for full sample expansion. Finally, we expand the sample with deionized water (*Figure 17I*), which is expected to yield a linear expansion factor of ~20x (based on results from the iExM publication), and then image the fluorescent signals from the LNA, which reports the spatial positions of the immunostained fluorescent proteins.

## Super-resolution imaging of sub-neuronal features with iExCel

We tested whether the iExCel protocol could indeed achieve ~20x linear expansion, which would yield an effective resolution of ~25 nm, in cuticle-enclosed entire animals of *C. elegans*. To validate improvements in resolution from the iterative expansion process, it would be informative to compare the same animal at three different stages of the protocol: prior to expansion, after 1 round of expansion, and after 2 rounds of expansion. To perform such a comparison, we first prepared hydrogel-embedded *tag-168p::GFP* animals (which express GFP pan-neuronally in the cytosol) with the standard ExCel protocol (*Figure 1A–C, E–G*), on which we acquired pre-expansion images right after hydrogel-embedding, and prior to Proteinase K digestion, similar to how we acquired pre-expansion images for other ExCel-related experiments (*Figures 3A*, *6A*, *7A* and *14A*). We then treated the sample with Proteinase K digestion and partial expansion (*Figure 1G–I*), and then proceeded to the iExCel protocol, by staining GFP with a primary antibody and a DNA-oligo-conjugated secondary antibody, performing full expansion of the first hydrogel to a 3.8x linear expansion factor, re-embedding into the second hydrogel, and linker hybridization (*Figure 17A–E*).

Normally, the iExCel protocol (*Figure 17*), similar to the published iExM protocol, does not include steps that enable readout at the post-1st-round-expansion stage (since the fluorescent dye, which reports the immunostained locations, is not introduced until right before the 2nd-round expansion), so we added the following steps into the iExCel protocol to enable post-1st-round-expansion readout, for this validation experiment. After linker hybridization, we incubate the sample with a fluorescent-dye-conjugated 15-base DNA oligo, which has the same sequence as the LNA

oligo that would be applied downstream for the post-2-round-expansion readout (i.e. at *Figure 17H*), except that it is composed completely of DNA, since the readout at this post-1st-round-expansion stage does not need to be performed in deionized water (it is re-embedded, so the hydrogel expansion factor is maintained at ~3–4x) and thus does not require LNA to maintain hybridization. These DNA oligos thus hybridize to the linker (which contains 4 repeats of the complementary sequences for the 15-base DNA oligos) and fluorescently report the stained positions, allowing readout at this post-1st-round-expansion stage, which had a linear expansion factor of 4.6x.

After acquisition of the post-1st-round-expansion image, we de-hybridize the sample in a buffer that contains a high concentration of formamide (80% formamide + 0.1% Triton X-100), which is an organic solvent that disrupts nucleic acid hybridization, at 37°C for 6 hr. Afterwards, we re-hybridize the sample with the linker. These de-hybridization and re-hybridization steps replace the linkers that are occupied by the 15-base fluorescent-dye-conjugated DNA oligos, with a set of new, unbound linkers, to ensure that downstream LNA hybridization would not be affected by this intermediate readout stage. After linker replacement, we proceed to the rest of the iExCel protocol (*Figure 17E–I*), by re-embedding into the third hydrogel, cleaving the first and the second gels, hybridizing with fluorescent-dye-conjugated LNA, and then fully expanding the sample, to yield an expansion factor of ~20.6x.

We then compared images of the same animal, acquired at the pre-expansion, post-1st-round-expansion, and post-2nd-round expansion stages (*Figure 18A*). We observed that visible distortion is fairly low throughout the iExCel process, perhaps because the devised protocol uses the same, highly homogenizing 2 day 37°C Proteinase K treatment as in the standard ExCel protocol. To quantify the distortion associated with the entire iExCel protocol, we applied non-rigid registration between the pre-expansion and post-2nd-round-expansion images, and calculated the root-mean-square (RMS) error of feature measurements (*Figure 18—figure supplement 1*), as performed in *Figure 3B* and *Figure 14B*. We observed that the error over 0 to 100 microns was ~1.5–4.5% for animals across developmental stages from L1 to day 1 adulthood, which are within a similar range to that from standard ExCel (~1–6%), a single-round expansion protocol. These observations are consistent with results from earlier ExM studies, which suggested that the micron-scale isotropy of iteratively expanded samples was essentially equivalent to those of single-round-expanded samples (*Chen et al., 2015*; *Chang et al., 2017*). Next, we observed improved sharpness of cell boundaries and neuronal processes with progressive rounds of expansion, over the length scale of the entire pharyngeal region of the animal (*Figure 18B*, *Video 5*). The improvement from the unexpanded to the 1st-round-expanded stage seems more obvious than the improvement from the 1st-round- to the 2nd-round-expanded stage, perhaps because the former (a change of resolution from ~250 nm to ~250 nm / 4.6x = ~55 nm) is more easily appreciated at this tissue-level length scale, compared to the later (a change of resolution from ~55 nm to ~250 nm / 20.6x = ~12 nm). Consequently, when we examined regions of the animal over the length scales of single neurons (*Figure 18C*) or over length scales of sub-neuron compartments (*Figure 18D*; showing a neuronal projection and a portion of a soma), we observed that the 2-round-expanded image clearly resolves individual fluorescent puncta (which likely arose from individual GFP molecules; in the iExCel protocol, the target protein (GFP, in this case) could be equipped by antibody binding and oligo-nucleotide hybridization events, to become a complex of ~37 nm in diameter in the pre-expansion scale (i.e.,~750 nm in the post-expansion scale), and would appear as fluorescent puncta of ~50 nm in diameter in the pre-expansion scale (i.e.,~1040 nm in post-expansion scale) after the diffraction effect, on a confocal microscope), even in regions where the puncta are positioned so densely such that optical diffraction made them appear as continuous in the 1-round-expanded samples. We note that the highly punctate appearance of the cytosol-filling GFP, as observed from the 2nd-round-expanded images, is what we expected, because fluorescent signals are composed of individual molecules (GFP, in this case) that can appear as continuous under the diffraction effect if densely packed (i.e. if the intermolecule distance falls below the diffraction-limited distance); however, as we approach spatial resolutions around the scale of inter-molecule distances, as achieved by the ~20 nm resolution from iExM, we would expect to observe, and indeed observed, a collection of individually separable fluorescent puncta, much like other localization-based super-resolution techniques such as STORM. Thus, iExCel enables visualization of fluorescent proteins following iterative expansion of intact animals, for ~20x linear expansion factor and ~25 nm effective resolution, which is demonstrated here to be finer than the inter-molecule distance for an over-expressed protein target in the cytosol.

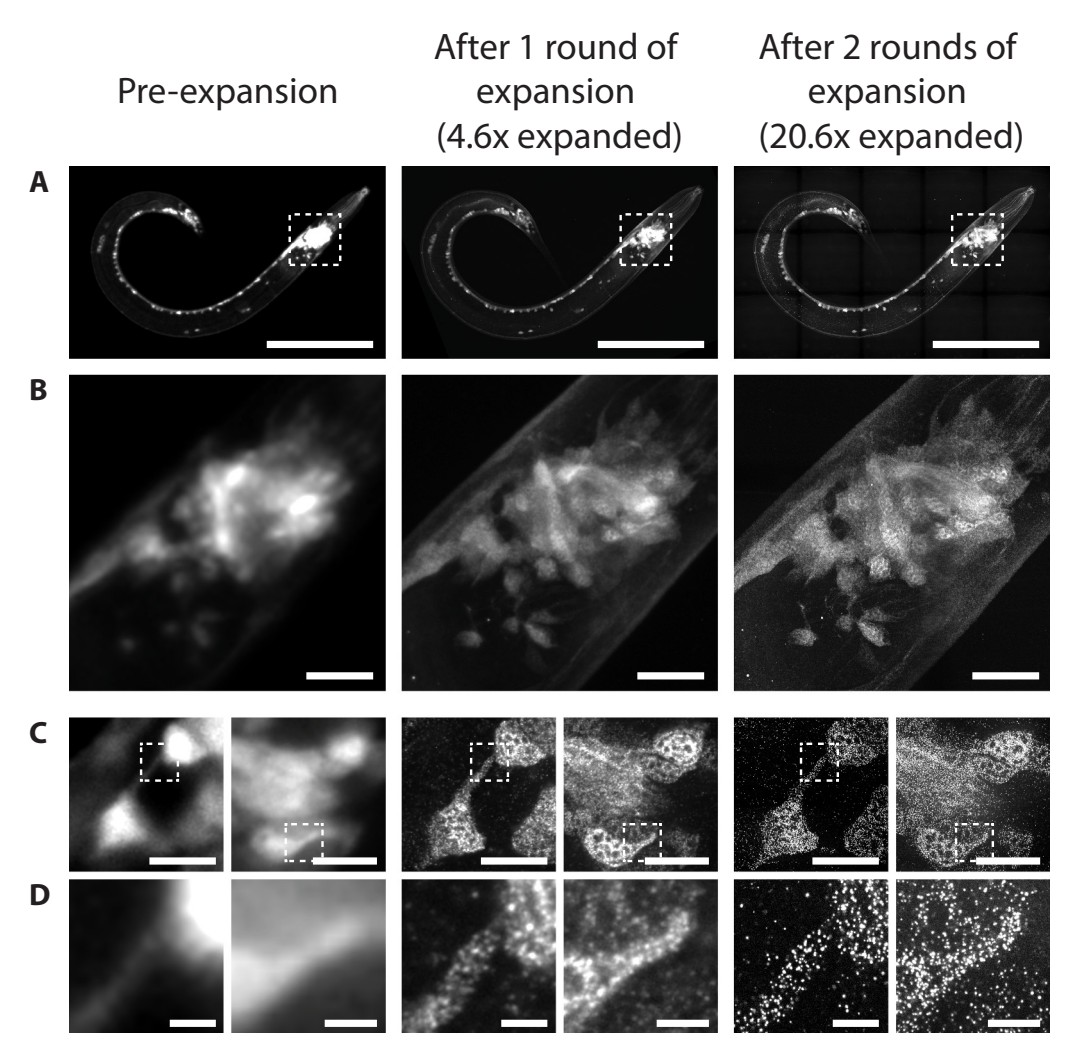

**Figure 18.** iExCel-expanded whole *C. elegans*. A representative iExCel-processed (formaldehyde-fixed, β-mercaptoethanol-reduced, AcX-treated, hydrogel-embedded, Proteinase-K digested, antibody-stained, second-gel-re-embedded, linker-hybridized, third-gel-re-embedded, LNA-hybridized, as in *Figure 17A–I*) L3 hermaphrodite animal at various stages along the iExCel protocol. These stages include: (left column) right after first hydrogel embedding, (middle column) after re-embedding into the second hydrogel, and (right column) after full expansion with the third hydrogel. The strain used had pan-neuronal cytosolic expression of GFP, under *tag-168p::GFP*. Pre-expansion images were acquired from native GFP fluorescence. Post-1-round-expansion images were acquired after linker hybridization and before re-embedding into the third gel (as in *Figure 17E*), accompanied by the following additional steps (not included in the standard protocol shown in *Figure 17*, because we performed this intermediate readout only for the purpose of method validation): specimens were incubated with a fluorophore-conjugated 15-base DNA oligo that hybridizes to the 100-base linker, imaged, incubated in de-hybridization buffer (80% formamide, 0.1% Triton X-100) at 37°C for 6 hr to remove the fluorophore-conjugated-DNA-bound linker, and re-hybridized with a fresh set of linker, using the same hybridization protocol shown in *Figure 17E*. This linker refreshment ensures that the linkers have completely unoccupied binding sites for the downstream LNA hybridization. Post-2-round-expansion images were acquired after full iExCel protocol (as in *Figure 17I*). (A–D) The animal at various optical and digital magnifications. (A) Entire worm. White dotted box marks the pharyngeal region of the worm, which is shown in greater magnification in B. (B) Pharyngeal region of the worm, as marked by the white dotted box in A. (C) Two regions within the pharyngeal region of the worm, as shown in B. Corresponding regions were not marked in B, because the single-confocal-plane images shown in C do not clearly register to regions in B, which is a maximum-intensity projection acquired through the entire thickness of the animal. White dotted box marks subcellular features that are shown in greater magnification in D. (D) Subcellular features of neurons, such as a neuronal process (right) and a portion of the neuronal soma (left), as marked by the white dotted box in C. Objective used: (A–B) 10x, NA 0.50; (C–D) 40x, NA 1.15. Image depth: (A–B) max-intensity projections of confocal stacks acquired through the entire thickness of the animal. (C–D) single z-position confocal images, except for post-2-round-expansion images (right column), which are max-intensity projections of 2 consecutive images within the confocal stack, because the expansion-mediated improvement in the axial resolution causes each z-plane image to capture a reduced tissue thickness. Thus, a combination of features captured across two consecutive z-planes was required to register to all perceptible, z-distributed features in the pre-expansion and the post-1-round-expansion images. Brightness and contrast settings: each panel was first set by the automatic adjustment function in

*Figure 18 continued on next page*

*Figure 18 continued*

Fiji, and then manually adjusted (raising the minimum-intensity threshold and lowering the maximum-intensity threshold) to improve contrast. Linear expansion factor: post-1-round expansion, 4.6x; post-2-round expansion, 20.6x. Scale bars: (**A**) 100 μm; (**B**) 10 μm; (**C**) 5 μm; (**D**) 1 μm.

The online version of this article includes the following source data and figure supplement(s) for figure 18:

**Source data 1.** Root-mean-square (RMS) length measurement errors plotted in *Figure 18—figure supplement 1*.
**Figure supplement 1.** Isotropy of iExCel.

## Discussion

We present three alternative protocols for ExCel, all of which enable physical expansion of intact, cuticle-enclosed animals of *C. elegans*. The standard ExCel protocol is a an ExM variant that permeates the tough *C. elegans* cuticle and permits antibody staining against fluorescent proteins, RNA fluorescent in situ hybridization, and morphological staining, all in the context of expansion microscopy. We show that ExCel can be used for multiplexed imaging of multiple molecular types, for synapse mapping, and for gene expression analysis in multiple individual neurons of the same animal. Because the standard ExCel protocol cannot detect general epitopes, and requires attachment of fluorescent reporters to target proteins, we developed epitope-preserving ExCel, which enables antibody staining against general untagged endogenous proteins. We show that epitope-preserving ExCel can detect previously unreported localization of a protein at the cell junctions between developing cells, and resolve peri-active and active zone proteins in chemical pre-synapses. Finally, we developed iterative ExCel, which enables two consecutive rounds of hydrogel-mediated expansion of entire worms, for ~20x linear expansion and a ~25 nm resolution, which is sufficient to resolve distances between individual cytosolic GFP molecules expressed in neurons. Compared to earlier super-resolution methods, all three variants of ExCel offer the ability to image at arbitrary depths in the worm, and on existing common microscopes available to many groups already.

The current standard of *C. elegans* in situ RNA detection, single-molecule FISH (smFISH), is diffraction-limited and requires wide-field illumination (*Ji and van Oudenaarden, 2012*), yielding single transcript sizes of ~200–500 nm laterally, and ~700–800 nm axially. Given the small neuronal size of *C. elegans*, smFISH can practically detect up to ~10–20 transcripts in an average-sized neuronal soma before losing single-transcript resolution. On the other hand, standard-ExCel-based ExFISH-HCR can detect at least 300 individually resolvable transcripts at ~3.3x linear expansion (which reduces the effective size of HCR amplified transcripts to ~150 nm laterally, and ~250 nm axially), thereby allowing precise digital quantification even for highly abundant mRNA targets, while also improving the attributability of single transcripts into sub-cellular compartments and identified single cells, due to nanoscale resolution in all other channels that provide spatial context (GFP, DAPI, NHS-ester-stained anatomical features). We note that out of the three variants of ExCel, the standard ExCel is the only one that currently supports RNA readout. The epitope-preserving ExCel uses a heat-mediated denaturation treatment at pH 9.0, which could induce alkaline hydrolysis of RNA molecules and disrupt downstream detection. In principle, RNA readout could be compatible with iterative ExCel, using a serial hybridization strategy similar to how the current iExCel protocol passes positional signals across hydrogels in different rounds of expansion, but we have not explored such a strategy in the present study. Therefore, the current incompatibility with RNA

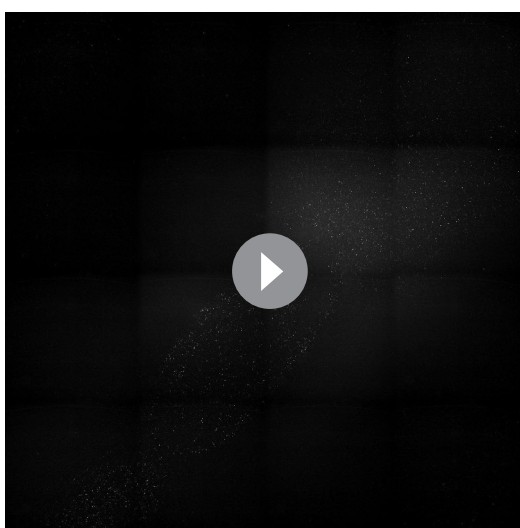

**Video 5.** iExCel-expanded nerve ring. Full confocal stack of the pharyngeal region of the 2nd-round-expanded L3 hermaphrodite shown in the right panel of *Figure 18B*. Scale bar: 10 μm.

https://elifesciences.org/articles/46249#video5

readout for these ExCel variants provide grounds for future improvements.

The standard ExCel protocol has good isotropy, with exceptions in the gonad and mouth areas, which suggests future improvements may be possible. Given that eggshells of *C. elegans* embryos (inside adult gonads) and the pharyngeal lumen walls (inside the mouth region) both contain chitin (*Heustis et al., 2012*; *Stein and Golden, 2018*), and are the only structures reported to contain chitin in *C. elegans* (*Zhang et al., 2005*), it is possible that future versions of ExCel that incorporate chitin disruption may be able to alleviate these distortions. Epitope-preserving ExCel has a lower expansion isotropy compared to standard ExCel, because its epitope-friendly treatment, which combines collagenase and a heat-mediated denaturation, is not as effective in mechanically homogenizing the cuticle and/or the internal worm tissue, as the standard, epitope-disruptive, Proteinase K treatment. On the other hand, iterative ExCel has a high isotropy that is on par with standard ExCel, because it inherits the Proteinase K treatment from standard ExCel; for that same reason, it can only detect fluorescent proteins, but not general proteins, in its current state. Future improvements on the ExCel toolbox could aim at enhancing the isotropy of epitope-preserving ExCel, and increasing the molecular types that can be imaged with iterative ExCel. iExCel achieves ~20x linear expansion, and a theoretical limit of resolution down to ~25 nm. Using additional hydrogel crosslinkers that are compatible with DATD and bis, it would be feasible to perform three or more rounds of expansion, as previously demonstrated in iExM (*Chang et al., 2017*), which could further improve theoretical limits of resolution. Similar to any other super-resolution microscopy, such as electron microscopy, the ability to visualize fine structural details is dependent on not only the magnification, but also a stain that can densely label biological structures, to make it possible for finer spatial sampling of the structure. In this regard, future developments of iExM and iExCel should consider strategies to maximize the density of labeling molecules on biological structures (e.g. use of smaller-size affinity probes, instead of large molecules that can have steric effects and reduced packing densities, e. g. ~150 kDa IgG antibodies), hydrogel chemistries that can accommodate increased densities of labeling molecules, and strategies to ensure positional signals are thoroughly transferred across hydrogels in multiple rounds of expansion.

Expansion microscopy protocols are continuously being extended, to include greater multiplexing capability for both RNA imaging (*Chen et al., 2016*; *Wang et al., 2018*) and protein imaging (*Ku et al., 2016*), physical magnification (*Chang et al., 2017*; *Truckenbrodt et al., 2018*), imaging of lipid membranes (*Karagiannis, 2019*), hydrogels that yield greater nanoscale isotropy (*Gao, 2019*), and other features. These augmentations could in principle be applied to ExCel as well. For example, if combined with a high-density membrane stain and antibodies against synaptic proteins, ExCel and its variant protocols, such as iterative ExCel with three or more rounds of hydrogel expansion, might facilitate the reconstruction of a molecularly annotated connectome (i.e. with molecular information at each synapse) of a single entire animal, potentially acquired and segmented at a higher speed than with electron microscopy by utilizing unique advantages offered by light microscopy, such as optical sectioning, spectral multiplexing, and the related wealth of information that might facilitate semi- or fully automated connectomic analysis.

# Materials and methods

**Key resources table**

| Reagent type (species) or resource | Designation | Source or reference | Identifiers | Additional information |
|---|---|---|---|---|
| Strain, strain background (*Caenor-habditis elegans*) | N2 | *Caenorhabditis Genetics* Center | RRID: WB-STRAIN:WB Strain00000001 | Genotype: wild-type |
| Strain, strain background (*C. elegans*) | CX16682 | This paper | | Genotype: *kyIs 700 [tag-168p::GFP; tag-168p::rpl-22-3xHA] ?* ('?" denotes *that the chromosome where the construct got integrated is unknown*) |

*Continued on next page*

*Continued*

| Reagent type (species) or resource | Designation | Source or reference | Identifiers | Additional information |
|---|---|---|---|---|
| Strain, strain background (*C. elegans*) | CZ1632 | *Caenorhabditis Genetics* Center | RRID: WB-STRAIN:WBStrain00005366 | Genotype: *juIs76 [unc-25p::GFP + lin-15(+)] II* |
| Strain, strain background (*C. elegans*) | NM2415 | *Caenorhabditis Genetics* Center | RRID: WB-STRAIN:WBStrain00029065 | Genotype: *jsIs682 [rab-3p::GFP::rab-3 + lin-15(+)] III* |
| Strain, strain background (*C. elegans*) | KP1148 | *Caenorhabditis Genetics* Center | RRID: WB-STRAIN:WBStrain00023626 | Genotype: *nuIs25 [glr-1p::glr-1::GFP + lin-15(+)] ?* |
| Strain, strain background (*C. elegans*) | CZ333 | *Caenorhabditis Genetics* Center | RRID: WB-STRAIN:WBStrain00005345 | Genotype: *juIs1 [unc-25p::snb-1::GFP + lin-15(+)] IV* |
| Strain, strain background (*C. elegans*) | CF702 | *Caenorhabditis Genetics* Center | RRID: WB-STRAIN:WBStrain00004831 | Genotype: *muIs32 [mec-7p::GFP + lin-15(+)] II* |
| Strain, strain background (*C. elegans*) | NQ570 | *Caenorhabditis Genetics* Center | RRID: WB-STRAIN:WBStrain00029098 | Genotype: *qnIs303 [hsp-16.2p::flp-13 + hsp-16.2p::GFP + rab-3p::mCherry] IV* |
| Strain, strain background (*C. elegans*) | AML32 | *Caenorhabditis Genetics* Center | RRID: WB-STRAIN:WBStrain00000192 | Genotype: *wtfIs5 [rab-3p::NLS::GCaMP6s + rab-3p::NLS::TagRFP] ?* |
| Strain, strain background (*C. elegans*) | OH16372 | This paper | | Genotype: *che-7(ot866[che-7::TagRFP)]* |
| Antibody | anti-GFP (chicken polyclonal) | Abcam | Cat#: ab13970; RRID:AB_300798 | IHC (1:100) |
| Antibody | anti-GFP (rabbit polyclonal) | Thermo Fisher Scientific | Cat#: A11122; RRID:AB_221569 | IHC (1:200) |
| Antibody | anti-mCherry (rabbit polyclonal) | Kerafast | Cat#: EMU106 | IHC (1:500 – 1:2000) |
| Antibody | anti-TagRFP (guinea pig polyclonal) | Kerafast | Cat#: EMU108 | IHC (1:500 –1:2000) |
| Antibody | anti-RFP (rabbit polyclonal) | Thermo Fisher Scientific | Cat#: R10367; RRID:AB_2315269 | IHC (1:100) |
| Antibody | anti-DLG-1 (mouse monoclonal) | Developmental Studies Hybridoma Bank | Cat#: DLG1; RRID:AB_2314321 | IHC (5 µg/mL) |
| Antibody | anti-UNC-10 (mouse monoclonal) | Developmental Studies Hybridoma Bank | Cat#: RIM; RRID:AB_579790 | IHC (5 µg/mL) |
| Antibody | anti-UNC-10 (mouse monoclonal) | Developmental Studies Hybridoma Bank | Cat#: RIM2; RRID:AB_10570332 | IHC (5 µg/mL) |
| Antibody | anti-myotactin (mouse monoclonal) | Developmental Studies Hybridoma Bank | Cat#: MH46; RRID:AB_528387 | IHC (5 µg/mL) |
| Antibody | anti-SAX-7 (mouse monoclonal) | Developmental Studies Hybridoma Bank | Cat#: CeSAX7; RRID:AB_10567266 | IHC (5 µg/mL) |

*Continued on next page*

*Continued*

| Reagent type (species) or resource | Designation | Source or reference | Identifiers | Additional information |
|---|---|---|---|---|
| Antibody | anti-SNB-1 (mouse monoclonal) | Developmental Studies Hybridoma Bank | Cat#: SB1; RRID:AB_579792 | IHC (5 µg/mL) |
| Antibody | anti-DYN-1 (mouse monoclonal) | Developmental Studies Hybridoma Bank | Cat#: DYN1; RRID:AB_10572297 | IHC (5 µg/mL) |
| Antibody | anti-acetylated-tubulin (mouse monoclonal) | Millipore Sigma | Cat#: T7451; RRID:AB_609894 | IHC (5 µg/mL) |
| Antibody | anti-DAO-5 (mouse monoclonal) | Developmental Studies Hybridoma Bank | Cat#: DAO5; RRID:AB_10573805 | IHC (5 µg/mL) |
| Antibody | anti-LMP-1 (mouse monoclonal) | Developmental Studies Hybridoma Bank | Cat#: LMP1; RRID:AB_2161795 | IHC (5 µg/mL) |
| Antibody | anti-LMN-1 (mouse monoclonal) | Developmental Studies Hybridoma Bank | Cat#: LMN1; RRID:AB_10573809 | IHC (5 µg/mL) |
| Antibody | anti-HCP-4 (mouse monoclonal) | Developmental Studies Hybridoma Bank | Cat#: HCP4; RRID:AB_2078913 | IHC (5 µg/mL) |
| Antibody | anti-CYP-33E1 (mouse monoclonal) | Developmental Studies Hybridoma Bank | Cat#: CYP33E1; RRID:AB_10571938 | IHC (5 µg/mL) |
| Antibody | anti-HMR-1 (mouse monoclonal) | Developmental Studies Hybridoma Bank | Cat#: HMR1; RRID:AB_2153752 | IHC (5 µg/mL) |
| Antibody | anti-LET-413 (mouse monoclonal) | Developmental Studies Hybridoma Bank | Cat#: LET413; RRID:AB_10571452 | IHC (5 µg/mL) |
| Antibody | anti-chicken-IgY (donkey polyclonal) | Jackson Immuno-Research | Cat#: 703-005-155; RRID:AB_2340346 | IHC (10 µg/mL) |

## *C. elegans* strains and maintenance

All strains were maintained at 20℃ under standard conditions (*Stiernagle, 2006*). All experiments presented in this study were based on non-synchronized, non-starved populations. Developmental stages of individual worms were estimated based on previously reported length measurements at various developmental time points (*Byerly et al., 1976*). The following strains were used in this study (For all integrated transgenic strains, the chromosome which the transgene is integrated into is specified after the construct notation; we use '?" to denote cases in which the chromosome is unknown.): N2, CX16682 *kyIs700 [tag-168p::GFP; tag-168p::rpl-22-3xHA] ?*, CZ1632 *juIs76 [unc-25p::GFP + lin-15(+)] II*, NM2415 *jsIs682 [rab-3p::GFP::rab-3 + lin-15(+)] III*, KP1148 *nuIs25 [glr-1p::glr-1::GFP + lin-15(+)] ?*, CZ333 *juIs1 [unc-25p::snb-1::GFP + lin-15(+)] IV*, CF702 *muIs32 [mec-7p::GFP + lin-15(+)] II*, NQ570 *qnIs303 [hsp-16.2p::flp-13 + hsp-16.2p::GFP + rab-3p::mCherry] IV*, AML32 *wtfIs5 [rab-3p::NLS::GCaMP6s + rab-3p::NLS::TagRFP] ?*, OH16372 *che-7(ot866[che-7::TagRFP])*. Strain CX16682 was a gift from Steven W. Flavell. Strain OH16372 was generated by inserting the TagRFP-sequence in the che-7 locus right before the stop-codon using CRISPR/Cas9-mediated homologous recombination based on a previously described co-CRISPR method (*Kim et al., 2014*); the C-terminal fusion allele tags all isoforms of the che-7 loci. All other stains were obtained from Caenorhabditis Genetics Center (CGC).

## Fixation and cuticle reduction

A detailed step-by-step protocol for ExCel is available in Appendix 1. The most updated protocol can also be found on expansionmicroscopy.org. Unless otherwise specified, all procedures were

performed at room temperature (RT) and all centrifugation was performed at 400 g for 2 min. Animals were collected from agar plates into a 15 mL conical tube with M9 buffer. Animals were washed three times in fresh M9 buffer, distributed into 1.5 mL Eppendorf tubes, pelleted by centrifugation, chilled on ice for 5 min, and fixed with ice-cold paraformaldehyde fixative (1x PBS + 4% paraformaldehyde) on a tube rotator for 30 min at RT, and then incubated for 4 hr at 4°C. Fixed animals were washed three times in cuticle reduction buffer (25 mM borate buffer, 0.5% Triton X-100, 2% β-mercaptoethanol, pH 8.5), and incubated in cuticle reduction buffer overnight at 4°C. Cuticle-reduced animals were washed twice in BT (25 mM borate buffer, pH 8.5, 0.5% Triton X-100, pH 8.5), twice in PBST-0.5% (1x PBS, 0.5% Triton X-100), and stored in 1x PBS at 4°C for up to 2 weeks.

## LabelX and AcX treatments

For the standard ExCel protocol, the LabelX and AcX treatments were performed using the following procedure. LabelX reagent was prepared as previously described (*Chen et al., 2016*). Briefly, Label-IT Amine Modifying Reagent (Mirus Bio, LLC) was re-suspended in the provided Reconstitution Solution at 1 mg/mL. LabelX stock solution was prepared by reacting 100 µL of Label-IT Amine Modifying Reagent stock solution (at 1 mg/mL) to 10 µL of AcX stock solution (6-((acryloyl)amino)hexanoic acid, succinimidyl ester; also known as Acryloyl-X, SE; here abbreviated AcX; Thermo Fisher Scientific; re-suspended in DMSO at 10 mg/mL) overnight at RT. LabelX stock solution was stored at −20°C. To add gel-anchorable groups onto nucleic acids for ExFISH readout, fixed animals were treated with LabelX reagent at 0.02 mg/mL in nucleic-acid-anchoring buffer (20 mM MOPS, 0.1% Triton X-100, pH 7.7), overnight at 37°C with gentle shaking. We note that although LabelX was applied at 0.02 mg/mL for all data presented in this study, we observed that a lower concentration of 0.01 mg/mL resulted in no discernable difference in the quantity of detected RNA transcripts, but significantly improved the extent by which the animals expand (by ~10% of the expansion factor). Therefore, we recommend the use of 0.01 mg/mL for future applications. After the overnight incubation, samples were washed with PBST-0.1% (1x PBS, 0.1% Triton X-100) for three times, 30 min each at RT on a tube rotator. To add gel-anchorable groups onto fluorescent proteins for post-ExCel antibody staining, fixed animals were treated with AcX at 0.1 mg/mL in protein-anchoring buffer (also referred as MBST pH 6.0; 100 mM MES, 150 mM NaCl, pH 6.0), overnight at RT on a tube rotator. After the overnight incubation, samples were washed with MOPST-0.1% (100 mM MOPS pH 7.0, 150 mM NaCl, 0.1% Triton X-100) for three times, 30 min each at RT on a tube rotator. For samples treated with both LabelX and AcX, the LabelX treatment was performed prior to the AcX treatment.

## Embedding worms into the first hydrogel

Fixed animals were incubated in non-activated monomer solution (50 mM MOPS pH 7.0, 2 M NaCl, 7.5% (w/w) sodium acrylate, 2.5% (w/w) acrylamide, 0.5% (w/w) N,N′-diallyl-tartardiamide) overnight at 4°C. Fixed animals were incubated in activated monomer solution (0.015% (w/w) 4-hydroxy-TEMPO, 0.2% (w/w) TEMED, 0.2% (w/w) APS, in addition to the monomer solution) for 1 hr at 4°C. Gelation chambers were constructed on a rectangular glass slide based on a similar geometry as previously reported (*Chen et al., 2015*). Briefly, a rectangular 22 mm * 50 mm #1.5 coverslip was attached on top of a plain rectangular glass slide, using a small amount (1–5 µL) of deionized water to adhere the glass surfaces. Next, two square 22 mm * 22 mm #1.5 coverslips were placed on top of the rectangular coverslip, one at each end, as spacers. Animals were then transferred to the center of the gelation chamber, and another 22 mm * 50 mm #1.5 rectangular cover glass was placed on top of the droplet to enclose the chamber. Gelation chambers were incubated in a humidified 37°C incubator for 2 hr. Gelation chambers were gently opened by removing the top cover glass, and excessive gel was trimmed away with a razor blade.

## Standard ExCel: Proteinase K digestion

Proteinase K (New England Biolabs, 800 U/mL stock) was diluted at 1:100 into non-expanding digestion buffer (50 mM Tris pH 8.0, 500 mM NaCl, 40 mM CaCl₂, 0.1% Triton X-100). Gelled samples were incubated in this solution at 37°C over 2 days (40–48 hr), with an exchange of freshly prepared Proteinase K solution (prepared in the same way) after the first day (16–24 hr). Digested samples were step-wise expanded with serial washes of reducing salt concentration. Briefly, samples were washed once in TNT buffer (50 mM Tris pH 8.0, 1 M NaCl, 0.1% Triton X-100) with 20 mM CaCl₂,

once in TNT buffer with 10 mM CaCl$_2$, twice in TNT buffer with no CaCl$_2$, once in 5x saline-sodium citrate buffer (SSC) + 500 mM NaCl, and once with 5x SSC. Washing steps were performed for 30 min each at RT.

### Standard ExCel: Immunohistochemistry

Samples were stained with standard primary and secondary antibodies at 10 µg/mL in 5x SSCT (5x SSC + 0.1% Tween 20), overnight at 4°C. Following primary and secondary staining, samples were washed in 5x SSCT, 3 times for 1 hr each at RT.

### Standard ExCel: NHS-ester staining and DAPI staining

For NHS-ester staining, stock solution of the NHS ester of a fluorescent dye (Atto 647N NHS ester) was prepared at 10 mg/mL in anhydrous DMSO, and stored at −20°C. Samples were incubated with Atto 647N NHS ester at 2 µM in NHS-ester staining buffer (5x SSCT, pH 6.0) overnight at RT. Samples were washed in 5x SSCT, 3 times for 30 min each at RT. For DAPI staining, samples were incubated with DAPI at 5 µg/mL in 5x SSCT for 30 min at RT. Samples were washed in 5x SSCT, 3 times for 10 min each at RT.

### Standard ExCel: Sample expansion

For 3.3x linear expansion factor, samples were serially expanded in 2.5x SSC, 0.5x SSC and 0.05x SSC, for 30 min each at RT. For 3.8x linear expansion factor, samples were additionally washed once in deionized water, for 30 min at RT.

### Standard ExCel: Imaging

All liquid is removed from around the gelled sample to minimize sample movement during imaging. Samples were imaged in glass-bottom 6-well plates, on a Nikon Eclipse Ti-E inverted microscope with a CSU-W1 spinning disk confocal module, with a long-working-distance water-immersion 40x objective (1.15 NA). All scale bars represent lengths before expansion (in biological units, e.g. post-expansion lengths are divided by the expansion factor, for all expanded specimens) unless otherwise noted.

### Standard ExCel (with ExFISH-HCR): Hydrogel re-embedding

For applications involving ExFISH-HCR in the standard ExCel protocol, hydrogels were re-embedded into a 4% acrylamide gel following a previously published protocol (*Chen et al., 2016*). Briefly, digested samples were step-wise expanded with serial washes of reducing salt concentration (2.5x SSC, 0.5x SSC, 0.05x SSC, deionized water), for 30 min each at RT. Expanded samples were incubated in re-embedding monomer solution (4% acrylamide, 0.2% N,N'-methylenebisacrylamide, 5 mM Tris base, 0.05% TEMED, 0.05% APS) for 30 min at RT, and enclosed in a gelation chamber constructed in the same way as described in the 'Embedding worms into the first hydrogel' section, but with three times the spacer height (stack of three #1.5 cover glasses) to accommodate the expanded height of the gel. The gelation chamber was incubated in a humidified 37°C incubator for 2 hr. Gelation chambers were gently opened by removing the top cover glass, and excess gels were trimmed away with a razor blade. Re-embedded samples were washed 3 times in 5x SSC, 10 min each at RT.

### Standard ExCel (with ExFISH-HCR): Probe design for FISH-HCR

Custom HCR v3.0 probe sets against *egfp*, mouse parvalbumin, *unc-25* (isoform C), *cho-1* (the only isoform), *tph-1* (isoform B), *eat-4* (isoform B), *cat-2* (isoform B) and *rab-3* (isoform B) were obtained from Molecular Technologies, using mRNA sequences of the target transcript found on the NCBI Reference Sequence (RefSeq) Database. Probe sets were synthesized at a probe set size of 20 split-initiator pairs.

### Standard ExCel (with ExFISH-HCR): Probe hybridization and HCR amplification for RNA readout

ExFISH-HCR was performed with HCR v3.0 kits from Molecular Technologies, which contains split-initiator probes against the specified RNA target, fluorophore-tagged HCR hairpins, and associated buffers. To perform probe hybridization, samples were pre-incubated in the provided 30% probe

hybridization buffer for 1 hr at 37°C. Probe solution was prepared by adding HCR-initiator-attached probes to 30% probe hybridization buffer at 4 nM (1:500 dilution). Samples were incubated in the probe solution at 37°C overnight. Samples were washed with the provided 30% probe wash buffer four times, for 30 min each at 37°C, and then washed with 5x SSCT (5x SSC, 0.1% Tween 20) three times, for 30 min each at RT. To perform HCR amplification, samples were pre-incubated in the provided amplification buffer for 30 min at RT. Fluorophore-tagged HCR hairpin stock was snap-cooled at 95°C for 90 s, and left at RT for 30 min in the dark. HCR hairpin solution was prepared by adding each snap-cooled hairpin stock to the amplification buffer at 15 nM (1:200 dilution). Samples were incubated in HCR hairpin solution overnight at RT, and washed with 5x SSCT two times, for 1 hr each at RT.

## Epitope-preserving ExCel: Collagenase VII digestion

Hydrogel-embedded worm samples, as prepared in section 'Embedding worms into the first hydrogel', were digested with collagenase type VII at 0.5 kU/mL in a calcium-containing buffer (100 mM Tris pH 8.0, 500 mM NaCl, 40 mM CaCl$_2$). Afterwards, samples were washed three times in TNC-40020 Buffer (50 mM Tris pH 8.0, 400 mM NaCl, 20 mM CaCl$_2$) for 1 hr each at RT.

## Epitope-preserving ExCel: Denaturation

Collagenase VII digested samples were denatured in Protein Denaturation Buffer (50 mM Tris pH 9.0, 200 mM sodium dodecyl sulfate, 400 mM NaCl, 20 mM CaCl$_2$) for 18–24 hr at 37°C, and then for 18–24 hr at 70°C, and finally for 2 hr at 95°C. Afterwards, samples were washed three times in TNC-40020 Buffer for 1 hr each at 37°C, and then overnight. Next, samples were washed once in TNT Buffer (50 mM Tris pH 8.0, 1M NaCl, 0.1% Triton X-100) + 10 mM CaCl$_2$, and then twice in TNT Buffer without CaCl$_2$, for 1 hr each at RT. Then, samples were incubated in TNT Buffer overnight at RT. Afterwards, samples were washed once in PNT-500 (1x PBS, 0.1% Triton X-100, 500 mM NaCl) for 30 min at RT, and then washed twice with PBST-0.1% (1x PBS, 0.1% Triton X-100) for 30 min each at RT.

## Epitope-preserving ExCel: Immunostaining and AcX-mediated antibody anchoring

Samples were stained with standard primary and secondary antibodies at 10 µg/mL in PBST-0.1% (1x PBS, 0.1% Triton X-100), overnight at RT. Following primary and secondary staining, samples were washed in PBST-0.1%, 3 times for 1 hr each at RT. Afterwards, samples were treated with AcX at 0.1 mg/mL in PBST-0.1% overnight at RT, and then washed in PBST-0.1% for three times, 1 hr each at RT.

## Epitope-preserving ExCel: Re-embedding into an expandable second gel

Samples were re-embedded into an expandable second gel, using the same protocol as described in 'Embedding worms into the first hydrogel', except for three changes; (1) the monomer solution was replaced by the epitope-preserving ExCel G2 Monomer Solution (50 mM MOPS pH 7.0, 0.15 M NaCl, 40 mM CaCl$_2$, 7.5% (w/w) sodium acrylate, 2.5% (w/w) acrylamide, 0.15% (w/w) N,N'-methylenebisacrylamide); (2) the pre-gelation 4°C incubation time was 45 min instead of 1 hr; (3) the gelation chamber spacer consisted of a stack of one #0 and one #1 cover glasses (for a total height of ~250 µm), instead of one #1.5 cover glass (which is ~180 µm).

## Epitope-preserving ExCel: Proteinase K digestion

Proteinase K (New England Biolabs, 800 U/mL stock) was diluted at 1:100 into non-expanding digestion buffer (50 mM Tris pH 8.0, 500 mM NaCl, 40 mM CaCl$_2$, 0.1% Triton X-100). Re-embedded samples were incubated in this solution at RT overnight (18–24 hr). Digested samples were step-wise expanded with serial washes of reducing salt concentration. Briefly, samples were washed once in TNT buffer (50 mM Tris pH 8.0, 1 M NaCl, 0.1% Triton X-100) with 20 mM CaCl$_2$, once in TNT buffer with 10 mM CaCl$_2$, three times in TNT buffer with no CaCl$_2$, once in PNT-500 (1x PBS + 0.1% Triton X-100 + 500 mM NaCl), and once with 1x PBS. Washing steps were performed for 30 min each at RT.

### Epitope-preserving ExCel: Cleavage of DATD-crosslinked first hydrogel

Samples were incubated in DATD-cleaving solution (1x PBS pH 5.5, 200 mM sodium periodate) for 30 min at RT with gentle shaking. Samples were then washed with 1x PBS for six times, 20 min each, at RT with gentle shaking. Then, samples were incubated in 1x PBS overnight at RT with gentle shaking.

### Epitope-preserving ExCel: NHS-ester staining

Stock solution of the NHS ester of a fluorescent dye was prepared by the same procedure as described in section 'Standard ExCel: NHS-ester staining and DAPI staining'. Samples were incubated with Alexa 405 NHS ester at 5 µM in PBST (1x PBS + 0.1% Triton X-100) overnight at RT. Samples were washed in PBST, 3 times for 30 min each at RT.

### Epitope-preserving ExCel: Expansion and imaging

Samples were serially expanded in once in 0.1x PBS, and then twice in deionized water, for 30 min each at RT. Afterwards, samples were imaged via the same procedure as described in 'Standard ExCel: Imaging'.

### iExCel: Immunostaining

Proteinase K digested samples, prepared as in section 'Standard ExCel: Proteinase K digestion', were stained with standard primary antibodies at 10 µg/mL in 5x SSCT (5x SSC + 0.1% Tween 20), overnight at 4°C. Samples were washed in 5x SSCT, 3 times for 1 hr each at RT. Samples were then stained with secondary antibodies that are conjugated to a 24-base DNA oligo with a 5' acrydite modification (see Appendix 1 – 'Oligo-nucleotides used in the iExCel Protocol' for oligo sequence and detailed conjugation procedure; essentially the same as previously described in the original ExM publication [*Chen et al., 2015*]), except that the S-HyNic reaction is performed at 3 times the original concentration, i.e. 6:100 dilution, except for 2:100 dilution from the original protocol), at 10 µg/mL in DNA-conjugated Antibody Staining Buffer (2% dextran sulfate, 2x SSC, 1 mg/mL Baker's yeast tRNA, 5% normal donkey serum, 0.1% Triton X-100) overnight at 4°C. Samples were washed in 5x SSCT, 3 times for 1 hr each at RT.

### iExCel: Re-embedding into a second, non-expanding hydrogel

Samples were re-embedded into the iExCel second hydrogel, using the same protocol as described in 'Standard ExCel (with ExFISH-HCR): Hydrogel re-embedding', except that the re-embedding monomer solution (4% acrylamide, 0.2% N,N'-methylenebisacrylamide, 5 mM Tris base, 0.05% TEMED, 0.05% APS) was replaced by iExCel G2 Monomer Solution (10% acrylamide, 0.5% DATD, 0.05% TEMED, 0.05% APS).

### iExCel: Linker hybridization

Samples were incubated with the 100-base DNA oligo linker (see Appendix 1 – 'Oligo-nucleotides used in the iExCel Protocol' for oligo sequence), at 100 nM, in iExCel hybridization buffer (4x SSC + 20% formamide) overnight at RT with gentle shaking. Then, samples were washed with iExCel hybridization buffer for four times, for 1 hr each for the first three washes and overnight for the final wash, at RT with gentle shaking. Samples were washed with 1x PBS for three times, 1 hr each at RT.

### iExCel: Re-embedding into a third, expanding hydrogel

Samples were incubated in activated iExCel G3 Monomer solution (1x PBS, 2 M NaCl, 7.5% (w/w) sodium acrylate, 2.5% (w/w) acrylamide, 0.15% (w/w) N,N'-methylenebisacrylamide, 0.015% (w/w) 4-hydroxy-TEMPO, 0.2% (w/w) TEMED, 0.2% (w/w) APS) for 50 min at RT. Afterwards, samples were enclosed in a gelation chamber, as described in the 'Embedding worms into the first hydrogel' section, but with three times the spacer height (stack of three #1.5 cover glasses) to accommodate the expanded height of the gel. The gelation chamber was incubated in a humidified 37°C incubator for 2 hr. Gelation chambers were gently opened by removing the top cover glass, and excessive gels were trimmed away with a razor blade. Re-embedded samples were washed 3 times in 5x SSC, 10 min each at RT.

### iExCel: Cleavage of DATD-crosslinked first and second hydrogels

Samples were incubated in DATD-cleaving solution (1x PBS pH 5.5, 200 mM sodium periodate) for 30 min at RT with gentle shaking. Samples were then washed with 1x PBS for eight times, 15 min each, at RT with gentle shaking. Then, samples were incubated in 1x PBS overnight at RT with gentle shaking.

### iExCel: LNA hybridization

Samples were incubated with the fluorescent-dye-conjugated 15-base LNA oligo (see Appendix 1 – 'Oligo-nucleotides used in the iExCel Protocol' for oligo sequence), at 100 nM, in iExCel hybridization buffer (4x SSC + 20% formamide) overnight at RT with gentle shaking. Then, samples were washed with iExCel hybridization buffer for four times, for 1 hr each for the first three washes and overnight for the final wash, at RT with gentle shaking. Samples were washed with 5x SSC for three times, 1 hr each at RT.

### iExCel: Expansion and imaging

Samples were serially expanded three times in deionized water, for 1 hr each at RT. Afterwards, samples were imaged via the same procedure as described in 'Standard ExCel: Imaging'.

### Analysis of signal-to-background ratio of immunohistochemical methods

Strains CX16682, NQ570, AML32 were used for this analysis. Animals were fixed and immunostained with one of the following immunohistochemistry protocols: Tube Fixation, Bouin's Tube Fixation, Peroxide Fixation (all as described previously [*Duerr, 2006*]), or the standard ExCel protocol (as described above). Primary antibodies used were chicken anti-GFP (Abcam ab13970), rabbit anti-mCherry (Kerafast EMU106) and guinea pig anti-TagRFP (Kerafast EMU108). Secondary antibodies used were conjugated to dyes that are spectrally separate from the fluorescent proteins (Alexa Fluor 546 was used for GFP; Alexa Fluor 647 was used for mCherry and TagRFP). Confocal stacks were acquired throughout entire animals. Optical intensities of the laser power were computed as the ratio of the laser power to the beam area. The laser powers at wavelengths of 561 nm and 647 nm were measured with an optical power sensor (Thorlabs S170C), and the beam area was computed from the beam diameter, which was measured with a caliper. For the signal-to-background analysis, max intensity projection of an animal was first cropped into 50 μm * 50 μm images centered around the nerve ring and the upper body (between the nerve ring and the vulva) regions (as in *Figure 2B*). Mask were generated by a semi-automated algorithm (MATLAB) to capture regions of the images corresponding to neurons around the nerve ring (where FPs were strongly expressed by the pan-neuronal promoters *tag-168p* or *rab-3p*) and to the tissue background (i.e. non-neuronal regions, with non-detectable FP expression) based on the percentile-ranking of intensity values of each pixel relative to the rest of the cropped image. Signal-to-background ratio was defined as the ratio of average intensity values between pixels inside the two masks. Analysis of the signal-to-background ratio was performed as schematized in *Figure 2B*. Representative images shown in *Figure 2C and E* were animals with the median signal-to-background ratio among all animals within each condition.

### Isotropy analysis of whole-nematode expansion

Strain CX16682 was used for this analysis. For the isotropy analysis of the standard ExCel protocol, animals of various developmental stages (L1 to day 2 adult) were imaged before and after ExCel under the same settings (spinning disk confocal microscopy with a 40 × 1.15 NA water-immersion objective). The pre-ExCel image was acquired after hydrogel embedding (*Figure 1G*) to fixate the orientation of the animals to enable registration with the post-ExCel image. The post-ExCel image was acquired after the additional steps of Proteinase K digestion, partial expansion to 1.8x, antibody staining, and expansion to 3.3 or 3.8x (*Figure 1H–J,L*). For *Figure 3B*, non-rigid registration was performed on max-intensity projections of pre- and post- ExCel images, using a custom MATLAB algorithm as previously described (*Chen et al., 2015*). Measurement error was defined as the absolute difference between the pre- and post- ExCel length measurements.

For the isotropy analysis of the epitope-preserving ExCel protocol, the same procedure as described above was applied, except that the animals were imaged at four different stages in the protocol (as shown in *Figure 14A*), including (1) the pre-ExCel stage (acquired after hydrogel

embedding, as in *Figure 13A*), (2) the stage after collagenase type VII digestion, denaturation, partial expansion and anti-GFP staining (as in *Figure 13E*), (3) the stage after re-embedding into the second gel (as in *Figure 13H*), and (4) the stage after the entire epitope-preserving ExCel protocol (as in *Figure 13L*). The measurement error in *Figure 14B* was quantified from the non-rigid registration between images of the pre-ExCel and the post-entire-protocol stages, using the same algorithm as described above.

For the additional analysis of straightened worms for epitope-preserving ExCel (*Figure 14C,D*), which normalizes body postures and removes the error contribution from changes in body posture during the epitope-preserving ExCel procedure, we applied the ImageJ built-in function 'Straighten' on images acquired as in *Figure 14A*, based on manually selected spline control points along the body midline of each worm. Afterwards, the measurement error in *Figure 14D* was quantified from the non-rigid registration between the straightened images of the pre-ExCel and the post-entire-protocol stages, using the same algorithm as described above.

For the isotropy analysis of the iExCel protocol, the same procedure as described above (for the standard ExCel protocol) was applied, except that the animals were imaged, using a $10 \times 0.50$ NA objective, at three different stages in the protocol (as shown in *Figure 18A*), including (1) the pre-ExCel stage (acquired after hydrogel embedding, as in *Figure 1G*), (2) the stage after second-gel re-embedding, linker hybridization, and hybridization of fluorophore-conjugated 15-base DNA oligo (as described in the main text, and in the caption of *Figure 18*), and (3) the stage after the entire iExCel protocol (as in *Figure 17I*). The measurement error in *Figure 18—figure supplement 1* was quantified from the non-rigid registration between images of the pre-ExCel and the post-entire-protocol stages, using the same algorithm as described above.

## Analysis of local distortion in mouth and adult gonad

Strain CX16682 was used for this analysis. Confocal images of before and after ExCel were acquired as described in 'Isotropy analysis of whole-nematode expansion'. Pre- and post-ExCel images were manually inspected and assigned with a score to quantify the extent of distortion. Images with distortion scores ranked at the 5th, 25th, 50th, 75th and 95th percentile were selected to represent the distribution of local distortion in ExCel-expanded animals in *Figure 3—figure supplements 1* and *2*.

## Quantification of RAB-3::GFP puncta

Strain NM2415 was used for this analysis. Confocal images of before and after ExCel were acquired as described in 'Isotropy analysis of whole-nematode expansion'. Max-intensity projections of the pre- and post- ExCel images were rigidly registered by scaled rotation. From each animal, line intensity profiles were generated over multiple segments of the ventral nerve cord or the SAB axonal processes. Line intensity profiles were linearly normalized to an intensity range between 0 and 1, and averaged with a moving window of 3 pixels to reduce noise. Local maxima from the line profiles were detected. The topological prominence of each local maxima was computed using the MATLAB built-in function *findpeak*. Local maxima with topographic prominence <0.01, which corresponds to local maxima whose heights are insignificant compared to the neighboring intensity level, were rejected as noise. The number of local maxima with topographic prominence >0.01 was defined as the peak count. We used deionized water to fully expand animals to ~3.8x linear expansion factor in *Figure 6A*. We used 0.05x SSC to expand animals to ~3.3x linear expansion factor in *Figure 6B* and *Figure 6C*, due to the higher stability of antibody staining in slightly salted environments, compared to deionized water.

## Quantification of FISH-HCR spots

Strain CX16682 was used for this analysis. Animals were fixed, sequentially treated with LabelX and AcX treatments, gel-embedded, digested with Proteinase K, re-embedded, processed with FISH-HCR and immunohistochemistry against GFP as described in sections above (*Figure 1A–I,M–Q*). Specimens were imaged under spinning disk confocal microscopy with a $40 \times 1.15$ NA water-immersion objective. Single-neuron RNA quantification was performed by a custom workflow developed in MATLAB. Briefly, max-intensity-projections for the x-, y- and z- directions were generated for both the GFP and the FISH-HCR channels. From the projection images, individual neurons were identified by their stereotypical location and neuronal morphology provided by the cytosolic GFP signal, as

well as prior knowledge of neuron-specific expression from the ExFISH signal. A 3D-bounding box enclosing each identified neuron was manually selected with an ROI-selection tool developed in MATLAB. Based on the coordinates of the selected 3D-bounding box, 3D stacks containing the identified neurons were cropped out from raw image stacks. For each cropped stack, FISH-HCR spots were detected using a 3D spot-finding algorithm developed by the Raj Lab (*Raj, 2017*) (source code and instructions can be found at https://bitbucket.org/arjunrajlaboratory/rajlabimagetools/wiki/Home). For each stack, a manually selected intensity threshold was applied to remove detected spots with very weak signals (ones close to the background noise).

## Expandability and stainability assays for the screen of post-gelation treatments

Strain CX16682 was used for these assays. For the expandability assay shown in *Figure 11A and C*, transillumination images were acquired on paraformaldehyde-fixed, β-mercaptoethanol-reduced, AcX-treated, and hydrogel-embedded hermaphrodite animals (as processed by *Figure 1A–C,E–G*) at three stages: (1) right after hydrogel embedding and prior to any hydrogel expansion, (2) after a candidate treatment and 1.9x-2.1x hydrogel expansion, by incubating the post-treatment gelled sample in 1x PBS, or (3) after a candidate treatment and 3.3–3.7x hydrogel expansion, by sequentially washing the post-treatment gelled sample with 0.5x PBS, 0.1x PBS, and 0.01x PBS. From images of the first and the third stages (i.e. pre-expansion and post-3.3–3.7x-expansion, respectively), expansion factors of the worm and the surrounding hydrogel were computed. The normalized expansion factor of the worm was computed via dividing the worm expansion factor by the hydrogel expansion factor.

For the stainability assay shown in *Figure 11B and C*, post-treatment samples were stained by a panel of 5 primary antibodies: anti-GFP, anti-LMN1, anti-myotactin, anti-DLG1 and anti-acetylated tubulin. The primary antibodies are then detected by secondary antibodies with the following fluorescent dyes, respectively: DyLight 405, Alexa 488, Alexa 546, Alexa 546 and Alexa 647. An IHC score from 0 to 1 was manually assigned to each channel, based on the estimated signal-to-noise ratio of the expected pattern of staining (estimated by manual inspection of the relative intensity values of the on-target pixels (signal; pixels identified by prior knowledge about the expected localization of staining) versus the off-target pixels (noise) in the fluorescence images). The overall stainability score of a treatment was the average IHC score across the four spectral channels.

## iExCel post-1-round-expansion readout

For readout of the post-1st-round-expansion stage, as shown in *Figure 18*, the following additional steps were added to the iExCel protocol, right after the linker hybridization step (described in section 'iExCel: Linker hybridization') and before the step of re-embedding into the third gel (described in section 'iExCel: Re-embedding into a third, expanding hydrogel'). Samples were incubated with a fluorescent-dye-conjugated 15-base DNA oligo (see Appendix 1 – 'Oligo-nucleotides used in the iExCel Protocol' for oligo sequence), at 100 nM, in iExCel hybridization buffer (4x SSC + 20% formamide) overnight at RT with gentle shaking. Then, samples were washed with iExCel hybridization buffer for four times, for 1 hr each for the first three washes and overnight for the final wash, at RT with gentle shaking. Samples were washed with 1x PBS for three times, 1 hr each at RT. Afterwards, samples were imaged via the same procedure as described in 'Standard ExCel: Imaging', to collect the post-1st-round-expansion images.

To remove the linkers that are occupied by the fluorescent-dye-conjugated 15-based DNA oligo, samples are incubated in de-hybridization buffer (80% formamide, 0.1% Triton X-100) at 37°C for 6 hr. Afterwards, the samples are again hybridized with a fresh, un-occupied set of linkers, using the same procedure as described in section 'iExCel: Linker hybridization'. Samples then proceed to re-embedding into the third hydrogel.

## Selection of representative images

For *Figure 2C and E*, and *Figure 3—figure supplements 1* and *2*, representative images were selected based on quantitative ranking of a particular attribute about the image, such as the signal-to-background ratio (for *Figure 2C and E*) or the distortion score (for *Figure 3—figure supplements*

*1* and *2*). Unless otherwise noted, the representative image is the image whose scored attribute is at the median, within the set of all acquired images.

For all other images (*Figures 2A, B*, *3A*, *4*, *5*, *6A*, *7*, *8A, B*, *9*, *10A*, *11A, B*, *12B, C*, *14A*, *15*, *16* and *18*, *Figure 13—figure supplement 1*), we did not perform quantitative ranking of a particular attribute about an image, often because images could contain multiple attributes (e.g. each channel within a multi-channel image could be ranked differently) without a particular order of priority. Instead, we manually selected images in which all the perceived attributes (such as the intensity and localization of each spectral channel, and the level of isotropy) appear typical among the set of all acquired images, based on author inspection.

## Statistical methods and data handling

As noted in *Dell et al. (2002)*, 'in experiments based on the success or failure of a desired goal, the number of animals required is difficult to estimate.' As was also noted in this paper, "the number of animals required is usually estimated by experience instead of by any formal statistical calculation, although the procedures will be terminated [when the goal is achieved]." Since the goal of the current paper was to demonstrate a technology, rather than test a hypothesis, we did not pre-determine any sample sizes for this study. The measurements were collected simply for providing a quantitative characterization of the underlying population. For this purpose, we employed statistical heuristics, and aimed to collect sufficient data to ensure that all reported quantities have at least three biological replicates. For the statistical test performed in *Figure 10D*, we used the two-sided Wilcoxon rank sum test, since we cannot assume normality of the underlying populations. For all experiments performed in this study, fixed animals were randomly allocated into experimental groups, by thoroughly mixing the tube of fixed animals (after the steps described in the 'fixation and cuticle reduction' section) and then aliquoting constant quantities into separate groups. Masking of experimental conditions (blind experiments) was not performed during data collection and analysis.

## Acknowledgements

The authors thank Steven W Flavell for sharing the transgenic strain CX16682. The authors thank Steven W Flavell, Piali Sengupta, Zainab Tanvir, Kiryl Piatkevich, Grace Huynh, Adam Marblestone, Anna Kazatskaya and Inna Nechipurenko for helpful discussions. C-CY acknowledges the McGovern Institute for Brain Research at MIT for the Friends of the McGovern Fellowship. GH acknowledges the New Jersey Institute of Technology for the Faculty Seed Grant. ESB acknowledges, for funding, Lisa Yang, John Doerr, the Open Philanthropy Project, NIH 1R01EB024261, NSF Grant 1734870, the HHMI-Simons Faculty Scholars Program, US Army Research Laboratory and the US Army Research Office under contract/grant number W911NF1510548, NIH 1R01MH103910, NIH 1R01MH114031, NIH 1R01MH110932, IARPA D16PC00008, NIH 2R01DA029639, and NIH 1RM1HG008525.

## Additional information

### Competing interests

Oliver Hobert: Reviewing editor, *eLife*. Chih-Chieh (Jay) Yu, Asmamaw T Wassie, Anubhav Sinha, Fei Chen: Is a co-inventor on patents related to expansion microscopy: WO2015127183A2, US20170089811A1, US20160305856A1, US20170067096A1, US20160304952A1, WO2017147435A1, US20190256633A1, and WO2020013833A1. Edward S Boyden: Is a co-inventor on patents related to expansion microscopy: WO2015127183A2, US20170089811A1, US20160305856A1, US20170067096A1, US20160304952A1, WO2017147435A1, US20190256633A1, and WO2020013833A1 and a co-founder of a company seeking to commercialize medical applications of expansion microscopy. The other authors declare that no competing interests exist.

## Funding

| Funder | Grant reference number | Author |
|---|---|---|
| McGovern Institute for Brain Research at MIT | Friends of the McGovern Fellowship | Chih-Chieh (Jay) Yu |
| New Jersey Institute of Technology | Faculty Seed Grant | Gal Haspel |
| John Doerr | | Edward S Boyden |
| The Open Philanthropy Project | | Edward S Boyden |
| National Institutes of Health | 1R01EB024261 | Edward S Boyden |
| National Science Foundation | 1734870 | Edward S Boyden |
| Howard Hughes Medical Institute | HHMI-Simons Faculty Scholars Program | Edward S Boyden |
| U.S. Army Research Laboratory | W911NF1510548 | Edward S Boyden |
| National Institutes of Health | 1R01MH103910 | Edward S Boyden |
| National Institutes of Health | 1R01MH114031 | Edward S Boyden |
| National Institutes of Health | 1R01MH110932 | Edward S Boyden |
| Intelligence Advanced Research Projects Activity | D16PC00008 | Edward S Boyden |
| National Institutes of Health | 2R01DA029639 | Edward S Boyden |
| National Institutes of Health | 1RM1HG008525 | Edward S Boyden |
| Lisa Yang | | Edward S Boyden |
| Army Research Office | W911NF1510548 | Edward S Boyden |

The funders had no role in study design, data collection and interpretation, or the decision to submit the work for publication.

## Author contributions

Chih-Chieh (Jay) Yu, Conceptualization, Resources, Data curation, Software, Formal analysis, Validation, Investigation, Visualization, Methodology, Writing - original draft, Project administration, Writing - review and editing; Nicholas C Barry, Investigation, Methodology; Asmamaw T Wassie, Anubhav Sinha, Methodology, Writing - review and editing; Abhishek Bhattacharya, Resources, Investigation, Generation of a transgenic animal strain that was crucial for drawing a major conclusion in this study; Shoh Asano, Investigation, Writing - review and editing; Chi Zhang, Resources, Investigation; Fei Chen, Conceptualization, Methodology; Oliver Hobert, Resources, Supervision, Writing - review and editing; Miriam B Goodman, Gal Haspel, Supervision, Writing - review and editing; Edward S Boyden, Supervision, Funding acquisition, Writing - original draft, Project administration, Writing - review and editing

## Author ORCIDs

Chih-Chieh (Jay) Yu (iD) https://orcid.org/0000-0001-8713-0446
Oliver Hobert (iD) http://orcid.org/0000-0002-7634-2854
Miriam B Goodman (iD) http://orcid.org/0000-0002-5810-1272
Gal Haspel (iD) http://orcid.org/0000-0001-6701-697X
Edward S Boyden (iD) https://orcid.org/0000-0002-0419-3351

## Decision letter and Author response

Decision letter https://doi.org/10.7554/eLife.46249.sa1
Author response https://doi.org/10.7554/eLife.46249.sa2

## Additional files

### Supplementary files

• Transparent reporting form

### Data availability

Source data files have been provided for all figures that contain quantitative analyses, including Fig. 2D, 2F, 3B, 6B-C, 10D-F, 11C, 14B, 14D, and Fig. 18 fig. supp. 1.

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

**Appendix 1**

# Expansion of *C. elegans* (ExCel) Step-by-Step Protocols

The most updated version of these protocols can be found at www.expansionmicroscopy.org.
   Protocols included in this section:

A.  Standard Expansion of *C. elegans* (ExCel)

- 3.3–3.8x linear expansion factor
- Supports readout of fluorescent proteins, RNAs, DNA, anatomical structures
- High isotropy (1–6% error over length scales between 0 to 100 μm)
- Protocol duration, from fixation to imaging:
  - For readout of fluorescent proteins only: 7 days
  - For readout of fluorescent proteins and RNAs: 10 days

B.  Epitope-Preserving Expansion of *C. elegans*

- 2.5–3.5x linear expansion factor
- Supports readout of a majority of endogenous and exogenous epitopes (~70% of epitopes that can be detected by non-IgM-class antibodies)
- Moderate isotropy (8–25% error over length scales between 0–100 μm)
- Protocol duration, from fixation to imaging: 14 days

C.  Iterative Expansion of *C. elegans* (iExCel)

- ~20.0x linear expansion factor
- Supports readout of fluorescent proteins; requires oligo-conjugated secondary antibodies (Conjugation protocol in 'Protocol for synthesizing DNA-conjugated Secondary Antibody for iExCel'; protocol duration: ~3 days)
- Similar level of isotropy as ExCel (1.5–4.5% error over length scales between 0–100 μm)
- Protocol duration, from fixation to imaging: 12 days

   The 'Materials' section, included at the end of these three protocols, contains recipes for all solutions used, and a list of vendors for all the reagents used.
   Supplementary notes in this section:

Notes on sodium acrylate quality
Oligo-nucleotides used in the iExCel protocol
Protocol for synthesizing DNA-conjugated secondary antibody for iExCel
Methods to track the surface of the hydrogel on which the worms are located

# Standard Expansion of *C. elegans* (ExCel) Protocol

### Fixation and Cuticle Reduction:

The default centrifugation setting is 400 g for 2 min, unless stated otherwise. Washes are performed at room temperature (RT), unless stated otherwise.

1.  Prepare PFA Fixative and chill to 4℃.
2.  Wash worms from plates into a 15 mL conical tube with M9. Fill the tube with M9 to 15 mL.
3.  Spin down and replace the supernatant with 15 mL of fresh M9. Repeat the M9 wash until bacteria is gone and solution is clear. (It is typically sufficient to perform two washes total.)
4.  Transfer worms to 1.5 mL tubes. For fixing large amounts of worms, split worms into multiple tubes such that the pellet size in each tube is no more than ~50 μL/tube. Spin down and remove as much supernatant as possible without disturbing the worm pellet.
5.  Place worm pellets and PFA Fixative on ice for 5 min.
6.  Add 1 mL of PFA Fixative to the worm pellet. Place sample on a tube rotator to mix vigorously at RT for 30 min.
7.  Incubate sample at 4℃ for 4 hr.
8.  Wash (spin down, remove supernatant, add buffer and mix thoroughly) sample with 1 mL of BTB for 3 times.
9.  After the 3 quick washes with BTB, incubate sample in 1 mL of BTB at 4℃ overnight.
10. Wash with 1 mL of BT for 2 times.

11. Wash with 1 mL of PBST-0.5% for 2 times.
12. Wash with 1 mL of PBS. The fixed sample can now be stored at 4°C for up to 2 weeks.

Before proceeding to the expansion protocol, determine the number of gel samples needed, allocate the corresponding number of worms to a separate 1.5 mL tube, and only process those worms instead of the entire tube of fixed worms. We recommend 30–50 worms per gel; higher densities will raise the difficulty of sample handling and imaging.

Reaction volumes in the following sections are specified for a single 1.5 mL tube of allocated worms. Each tube should contain a pellet size no more than ~30 µL / tube. Split into multiple tubes if necessary.

## RNA Anchoring (optional, for RNA readout)

1. Pre-incubate worms with 1 mL of RNA Anchoring Buffer at RT on a tube rotator for 1 hr.
2. Incubate worms with LabelX at 0.01 mg/mL (1:100 dilution of LabelX stock) in 1 mL of RNA Anchoring Buffer, overnight at RT on a tube rotator.
3. Wash worms with 1 mL of PBST-0.1% for 3 times, 30 min each at RT on a tube rotator.

## Protein Anchoring (for fluorescent protein readout)

1. Pre-incubate worms with 1 mL of Protein Anchoring Buffer at RT on a tube rotator for 1 hr.
2. Incubate worms with AcX at 0.1 mg/mL (1:100 dilution of AcX stock) in 1 mL of Protein Anchoring Buffer, overnight at RT on a tube rotator.
3. Wash worms with 1 mL of MOPST-0.1% for 3 times, 30 min each at RT on a tube rotator.

## Gelation

1. Incubate worms in 1 mL of Non-activated Monomer Solution overnight at 4°C.
2. Prepare 1 mL of Activated Monomer Solution with all reagents except for APS (Monomer Solution Stock + Triton X-100 + 4-HT + TEMED; do not add the activator APS yet).
3. Spin down worms. Remove supernatant.
4. Activate the monomer solution from Step 2 by adding 20 µL of 10% APS.
5. Add 1 mL of Activated Monomer Solution to worms. Incubate sample at 4°C for 60 min.
6. During incubation, construct gelling chambers besides the top coverslip (refer to **Appendix 1—figure 1**). Use #1.5 square cover glass (e.g. 22 * 22 mm) for the spacer.
7. Immediately after the 60 min incubation, spin down worms. Without disturbing the pellet, transfer ~30 µL of the supernatant as a droplet on the glass slide. Remove the rest of the supernatant and leave only ~30 µL along with the worm pellet.
8. Carefully re-suspend the pellet in the leftover volume with a pipette, without creating bubbles. Load all volume in the tube into a pipette. Push out any air in the tip of the pipette. Insert the pipette tip to the center of the droplet on the glass slide. Slowly push out the content so that the worms are relatively concentrated at the center of the droplet (refer to the top row in **Appendix 1—figure 1**). It is desirable to avoid worms from overlapping with one another (which can increase the difficulty with imaging), so it can be useful to gently spread out the worms if too concentrated.
9. Close the chamber by carefully and slowly placing a rectangular cover glass (e.g. 22 * 50 mm) on top of the spacers, minimizing bubble formation. One way to achieve this is to use a tweezer to clip on one end of a rectangular cover glass, hold the cover glass right above the chamber by ~5 mm (in parallel to the glass slide; no contact yet), press down on the un-clipped end of the rectangular cover glass until it contacts the spacer, release the clipping force so that the cover glass just rests on the lower branch of the tweezer, and then slowly lower the tweezer until the cover glass contacts the other spacer.
10. Incubate the gelling chamber at 37°C for 2 hr, in a humidified chamber.
11. After the 2 hr incubation and before proceeding to Step 12 (chamber opening), prepare the digestion solution (see Step 1 of the next section). The digestion procedure should be applied right after Step 13 (gel trimming) to avoid the gel drying out.
12. With a tweezer or a razor blade, remove the top cover glass and spacer and leave the gel on the glass slide. The gel can occasionally come off with the top cover glass. In that case, leave the gel on the cover glass and dispose the rest of the chamber.
13. With a razor blade, trim away excessive gel. The size of the trimmed gel should be such that, when fully expanded, the gel can fit flatly into the imaging container (a good start is ~7 mm * 7 mm, which can fit into a 6-well plate when expanded). If the worms were

successfully kept concentrated at the center of the gel, it should be relatively easy to retain most worms in the trimmed gel. (refer to the 4th row in *Appendix 1—figure 1*)

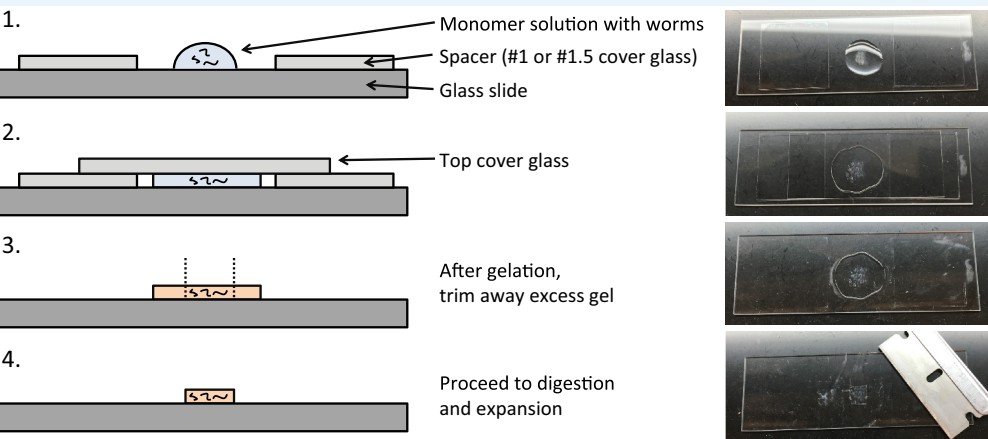

**Appendix 1—figure 1.** Construction of the gelling chamber.

## Digestion

Based on our experience, the extent of digestion seems to be quite sensitive to the quality of Proteinase K. Use Proteinase K ordered within 2 months. Reaction volumes in the following sections are specified for a single hydrogel. Scale up if you are processing multiple gels.

1. Prepare 1 mL of digestion solution by adding Proteinase K to Non-expanding Digestion Buffer to a final concentration of 8 U/mL (1:100 dilution).
2. Add ~50 µL of digestion solution on top of the trimmed gel. Let it sit for 5 min at RT.
3. Add ~950 µL of digestion solution to a well in a 24-well plate. With a flat-tip paint brush (see recommended brush in Materials; Utrecht 09013–1006), slowly insert the brush under the gel until most of the brush are inserted, vertically lift the gel with the brush, and transfer the gel into the well. Incubate sample overnight at 37˚C.
4. Replace solution with 1 mL of freshly prepared digestion solution. Incubate sample overnight at 37˚C.
5. Wash (now that the worms have been embedded, we use this word to mean simply exchange the immersing solution, without the centrifuging) once with 1 mL of TNT-20 Buffer (TNT Buffer + 20 mM CaCl$_2$) for 30 min at RT.
6. Wash once with 1 mL of TNT-10 Buffer (TNT Buffer + 10 mM CaCl$_2$) for 30 min at RT.
7. Wash twice with 1 mL of TNT Buffer for 30 min each at RT.
8. Wash once with 1 mL of 5x SSC-NaCl Buffer (5x SSC + 500 mM NaCl) for 30 min at RT.
9. Wash once with 1 mL of 5x SSC for 30 min at RT.

## ExFISH-HCR (for RNA readout)

### Part A: Maintain expansion factor of the gelled sample with re-embedding

ExFISH-HCR requires hybridization between DNA probes and RNA targets, which has reduced affinity in low ionic strength environment (e.g. de-ionized water, which is necessary to fully expand the gelled sample). This trade-off can be resolved by first casting the expanded sample into a non-expanding acrylamide gel, which locks up the expansion factor of the first gel. The re-embedded gel can then be processed and imaged in higher-salt environments that favor hybridization and maximize HCR amplicon stability.

1. Place a rectangular glass slide into a 4-well rectangular dish. Transfer sample onto the glass slide. Expand sample serially in 2.5x SSC, 0.5x SSC, 0.05x SSC and ddH2O, for 30 min each at RT. Use 3 mL for each washing step.

2. Incubate the sample in 3 mL of re-embedding monomer solution for 30 min at RT with gentle shaking. The amount of re-embedding monomer solution should be just sufficient to barely reach the top surface of the gel (80–100% of gel height), but not much more than that. Oxygen in the ambient environment slows down the gelation reaction to allow complete diffusion of the gel monomers over the 30 min incubation; too much monomer solution can cause premature gelation at the bottom of the well, due to reduced access to ambient oxygen.

3. After 30 min of incubation, transfer all monomer solution from the well to a separate 50 mL tube. During the solution transfer, position the expanded gel to the center of the glass slide, by tilting the 4-well plate and/or gently moving the gel with a flow of monomer solutions (using a pipette). The gel should not be close to, or hanging off from, the edge of the glass slide. Carefully transfer the glass slide from the 4-well plate onto a sheet of Kimwipe, to remove droplets on the bottom of the glass slide. Trim away excessive gel if necessary.

4. Place spacers of 3 times the height of initial gel (e.g. if #1.5 cover glass was used, use a stack of 3 * #1.5 cover glass) on both ends of the glass slide, with the gel in the center. Add a small amount (~50 µL) of the saved monomer solution onto the gel, and slowly place a rectangular cover slip on the gel, to enclose the gelling chamber in the same style as performed in the initial gelation. Avoid bubbles. Use a pipette to completely fill the chamber with extra monomer solution, such that the gel is fully enclosed by monomer solution, with minimal contact to ambient oxygen.

5. Incubate the gelling chamber at 37°C for 2 hr, in a humidified chamber (e.g. a 0.75-quart storage container, such as Amazon.com #B01M2BTKYB, with the gelling chamber sitting on top of a stack of two lids of the 6-, 24- or 96- well plate, and with a small amount (10–20 mL) of deionized water at the bottom of the sealed container).

6. Remove top cover glass and trim away excessive gels. We recommend splitting large gels into smaller ones with no more than 1.5 cm per dimension, which simplifies downstream processing.

7. Transfer trimmed gels into a 24-well plate. Wash 3 times with 5x SSC, 10 min each.

## Part B: Probe hybridization and HCR amplification

ExFISH-HCR readout is performed with HCR v3.0 kits from Molecular Technologies (https://www.moleculartechnologies.org/).

1. Pre-incubate sample in 500 µL of the provided 30% probe hybridization buffer for 1 hr at 37°C.

2. Prepare 500 µL of probe solution by adding HCR-initiator-attached DNA probes to 30% probe hybridization buffer at 4 nM for each probe mixture (odd and even; 1:500 dilution of the 2 µM stock).

3. Incubate sample with 500 µL of probe solution overnight at 37°C.

4. Wash sample with 1 mL of the provided 30% probe wash buffer for 4 times, 30 min each at 37°C.

5. Wash sample with 1 mL of 5x SSCT for 3 times, 30 min each at RT.

6. Pre-incubate sample in 500 µL of amplification buffer for 30 min at RT.

7. Snap cool 2.5 µL of each fluorophore-tagged HCR hairpin stock (provided at 3 µM) at 95°C for 90 s. Let snap-cooled stock solution sit in a dark environment at RT for 30 min.

8. Prepare HCR hairpin solution by adding 2.5 µL of each snap-cooled hairpin stock to 500 µL of amplification buffer.

9. Incubate sample in 500 µL of HCR hairpin solution overnight at RT.

10. Wash sample with 1 mL of 5x SSCT for 2 times, 1 hr each at RT.

## Antibody Staining (for fluorescent protein readout)

1. Pre-incubate sample in 1 mL of 5x SSCT for 30 min at RT.

2. Incubate sample with primary antibody against fluorescent protein, at desired concentration (a good starting point is 10 µg/mL) in 500 µL of 5x SSCT overnight at 4°C.

3. Wash sample with 1 mL of 5x SSCT for 3 times, 1 hr each at RT.

4. Incubate sample with secondary antibody with a fluorophore, at desired concentration (a good starting point is 10 µg/mL) in 500 µL of 5x SSCT overnight at 4°C.

5. Wash sample with 1 mL of 5x SSCT for 3 times, 1 hr each at RT.

## NHS-ester Staining (optional, for morphology readout)

1. Incubate sample with 2 µM of the NHS-ester of a fluorescent dye (e.g. Alexa Fluor 546 NHS Ester, or Alexa Fluor 647 NHS Ester) in 1 mL of NHS-ester staining buffer (5x SSCT, pH 6.0), overnight at RT.
2. Wash sample with 1 mL of 5x SSCT for 3 times, 30 min each at RT.

### DAPI Staining (optional, for DNA readout)

1. Incubate sample with 5 µg/mL DAPI in 1 mL of 5x SSCT for 30 min at RT.
2. Wash sample with 1 mL of 5x SSCT for three times, 10 min each at RT.

### Expansion and Imaging

1. Transfer sample to a container suitable for imaging, e.g. a glass-bottom 6-well plate.
2. For samples that have not been re-embedded, expand sample by serial washes with reducing salt concentration:
3. Wash sample with 2 mL of 2.5x SSC for 30 min at RT.
4. Wash sample with 2 mL of 0.5x SSC for 30 min at RT.
5. Wash sample with 2 mL of 0.05x SSC for 30 min at RT.
6. For all samples:
7. Remove as much liquid around the expanded gel as possible.
8. Perform imaging.
9. To store the sample, re-hydrate the sample in 2 mL of 5x SSCT at 4°C after imaging. HCR amplicons and IHC staining are stable for up to a week.

### Agar Immobilization (Optional)

If too much sample vibration or drifting is observed under a high-resolution objective (e. g. >= 40x) that affects image quality, immobilize expanded gel by adding droplets of lukewarm 2%(w/w) low-melt agarose solution to the edges of the expanded gel (after water is completely removed from the well, leaving only the gel at the center of the well). After ~2 min, slowly add more low-melt agarose solution to cover the entire gel. Wait for ~2 more minutes for the agarose to solidify, and proceed to imaging.

## Epitope-preserving Expansion of *C. elegans* Protocol

### Fixation and Cuticle Reduction

Same as in the standard ExCel protocol.

### Protein Anchoring

Same as in the standard ExCel protocol.

### Gelation

Same as in the standard ExCel protocol, but in Step 11, prepare Collagenase VII Solution instead of digestion solution. Also, to ensure that the final-state expanded worm could be captured within the working distance of a high-NA lens, perform one of the two methods for tracking hydrogel orientation (see below).

Since the hydrogel sample will become ~7.4x larger in the z-dimension at the end of this protocol, if it is desired to image with a short-working-distance objective (e.g. <1 mm; most high NA and magnification objectives are in this category) at the fully expanded state, it is necessary to ensure that the gel is flipped back to the same side as the Gel#1 casted state, at two time points: (1) when the gel is re-embedded into Gel #2, and (2) before the expansion process prior to the final imaging. This way, the worms (which settle to the lower ~50 µm during Gel #1 gelation, out of the full ~180 µm thickness of the gel) will remain at the bottom surface of the gel at the end of this protocol, ensuring maximal coverage of the worm tissue by the working distance of the objective during the final imaging. In order to provide a way to check the orientation (side) of the gel throughout the protocol, at least one of the two additional procedures is necessary at the end of the Gelation step. See 'Methods to track the surface of the hydrogel on which the worms are located' for a protocol of these procedures.

Reaction volumes in the following sections are specified for a single hydrogel. Scale up if you are processing multiple gels.

## Collagenase VII-mediated Cuticle Digestion:

1. Prepare 1 mL of Collagenase VII Solution by adding 500 µL of Collagenase VII Stock Solution (at 1 kU/mL, see Materials) to 500 µL of Collagenase VII Dilution Buffer.
2. Add ~50 µL of Collagenase VII Solution on top of the trimmed gel. Let it sit for 5 min at RT.
3. Add ~950 µL of Collagenase VII Solution to a 2 mL Eppendorf tube (which has a flatter bottom compared to the 1.5 mL tubes; if such is not available, 1.5 mL tube can work too). With a flat-tip paint brush (see recommended brush in Materials; Utrecht 09013–1006), slowly insert the brush under the gel until most of the brush are inserted, vertically lift the gel with the brush, and transfer the gel into the well. Incubate sample overnight at 37˚C.
4. Wash sample (i.e., replace the incubating solution) 3 times with 1 mL of TNC-40020 Buffer for 1 hr each at RT.

## Protein Denaturation

1. Incubate sample with 1 mL of Protein Denaturation Buffer (stored at 37˚C; see Materials) overnight at 37˚C.
2. Replace solution with another 1 mL of Protein Denaturation Buffer (at 37˚C). Incubate overnight at 70˚C.
3. Pre-warm 1 mL of Protein Denaturation Buffer to 70˚C.
4. Replace solution with 1 mL of Protein Denaturation Buffer (pre-warmed to 70˚C in Step 3). Incubate for 2 hr at 95˚C.
5. After the 95˚C incubation, incubate samples at 37˚C for 30 min. At this time, pre-warm at least 4 mL of TNC-40020 Buffer to 37˚C.
6. Wash sample 3 times with 1 mL of TNC-40020 (pre-warmed to 37˚C in Step 5) for 1 hr each at 37˚C. Then, incubate sample in TNC-40020 overnight at RT.
7. Wash sample once with TNT-10 Buffer (TNT Buffer + 10 mM CaCl$_2$) for 30 min at RT.
8. Wash sample twice with TNT Buffer for 30 min each at RT. Then, incubate sample in TNT Buffer overnight, or at least 6 hr, at RT. The rationale here is to completely wash out remaining calcium ions from the gel, so when a phosphate-based buffer (PBS) is added in the next step to prepare for immunostaining, phosphate ions will not precipitate with calcium ions to form a nearly insoluble product, which can cause a cloudy appearance in the gel and affect image quality.
9. Transfer gel sample to a 24-well plate by directing pouring the content in the 2 mL tube into a well. Alternatively, use a flat-tip paint brush to transfer the gel.
10. Wash sample once with PNT-500 (1x PBS + 0.1% Triton X-100 + 500 mM NaCl) for 30 min at RT.
11. Wash sample twice with PBST-0.1% (1x PBS + 0.1% Triton X-100) for 30 min each at RT.

## Antibody Staining

1. Incubate sample with primary antibody, at desired concentration (a good starting point is 10 µg/mL) in 500 µL of PBST-0.1% overnight at RT.
2. Wash sample with 1 mL of PBST-0.1% for 3 times, 1 hr each at RT.
3. Incubate sample with secondary antibody with a fluorophore, at desired concentration (a good starting point is 10 µg/mL) in 500 µL of PBST-0.1% overnight at 4˚C.
4. Wash sample with 1 mL of PBST-0.1% for 3 times, 1 hr each at RT.
5. Imaging can be optionally performed at this point to confirm expected patterns of staining. Linear expansion factor of the worm is ~1.2x at this stage. (Hydrogel expansion factor is ~2.0x, so worms will detach from the surrounding hydrogel, but stay inside the hydrogel mold, as shown in *Figure 12*. With reasonably gentle washes, retention of worms inside the hydrogel sample is typically >95% despite of this detachment due to the expansion factor mismatch between the worm and the gel.)

## Antibody Anchoring

1. Incubate sample with AcX at 0.1 mg/mL (1:100 dilution of AcX stock) in 1 mL of PBST-0.1%, overnight at RT.
2. Wash sample with 1 mL of PBST-0.1% for 3 times, 1 hr each at RT.

## Re-Embedding into an Expandable Second Gel

1. Flip each gel to the same side as when they were casted during the initial gelation. This ensures that the worms remain on the bottom surface of the gels, so the working distance of the objective can maximally cover the depth of the worm at the final imaging stage.
2. Incubate sample in 1 mL of Non-activated G2 Monomer Solution overnight at 4°C.
3. Prepare 1 mL of Activated G2 Monomer Solution with all reagents except for APS (Monomer Solution Stock + Triton X-100 + 4-HT + TEMED; do not add the activator APS yet).
4. Remove liquids from the sample.
5. Activate the monomer solution from Step 3 by adding 20 µL of 10% APS.
6. Add 1 mL of Activated Monomer Solution to the sample. Incubate sample at 4°C for 45 min.
7. During incubation, construct gelling chambers besides the top coverslip (based on the same architecture as shown in the Gelation section of the standard ExCel protocol above). Use a stack of one #0 cover glass and one #1 cover glass (total height ~250 µm) for the spacer.
8. Immediately after the 45 min incubation, transfer 65 µL of the monomer solution from the well that contains the gel sample to the center of the gelation chamber, as a single droplet.
9. With a flat-tip paint brush, transfer the gel sample into the droplet of monomer solution. If the gel orientation has been confirmed in Step 1, ensure that the hydrogel is placed in the correct orientation (i.e. with worms located at the bottom surface of the gel). Then, use a pipettor to temporarily remove the monomer solution from the gel, to ensure that the gel is completely flat and not folded. If the gel is folded, use a paint brush to unfold the gel. Then, add the monomer solution back on top of the gel to immerse the gel.
10. Close the chamber by carefully and slowly placing a rectangular cover glass (e.g. 22 * 50 mm) on top of the spacers, minimizing bubble formation. One way to achieve this is to use a tweezer to clip on one end of a rectangular cover glass, hold the cover glass right above the chamber by ~5 mm (in parallel to the glass slide; no contact yet), press down on the un-clipped end of the rectangular cover glass until it contacts the spacer, release the clipping force so that the cover glass just rests on the lower branch of the tweezer, and then slowly lower the tweezer until the cover glass contacts the other spacer.
11. Incubate the gelling chamber at 37°C for 2 hr, in a humidified chamber (e.g. a 0.75-quart storage container, such as Amazon.com #B01M2BTKYB, with the gelling chamber sitting on top of a stack of two lids of the 6-, 24- or 96- well plate, and with a small amount (10–20 mL) of deionized water at the bottom of the sealed container).
12. After the 2 hr incubation and before proceeding to Step 13 (chamber opening), prepare the digestion solution (see Step 1 of the next section).
13. With a tweezer or a razor blade, remove the top cover glass and spacer and leave the gel on the glass slide. The gel can occasionally come off with the top cover glass. In that case, leave the gel on the cover glass and dispose the rest of the chamber.
14. At this point, the boundary of the re-embedded gel sample should be easily visible under sufficient room light. With a razor blade, trim away the excessive gel outside of the original gel sample.

## Proteinase K Digestion

Based on our experience, the extent of digestion seems to be quite sensitive to the quality of Proteinase K. Use Proteinase K ordered within 2 months.

1. Prepare 1 mL of digestion solution by adding Proteinase K to Non-expanding Digestion Buffer to a final concentration of 8 U/mL (1:100 dilution).
2. Add ~50 µL of digestion solution on top of the trimmed gel. Let it sit for 2 min at RT.
3. Add ~950 µL of digestion solution to a well in a 24-well plate. With a flat-tip paint brush to transfer the gel into the well. Incubate sample overnight (18–24 hr) at **room temperature**. (Bold text underscores the difference from the digestion procedure in the standard ExCel protocol.)
4. Wash once with 1 mL of TNT-20 Buffer (TNT Buffer + 20 mM CaCl$_2$) for 30 min at RT.
5. Wash once with 1 mL of TNT-10 Buffer (TNT Buffer + 10 mM CaCl$_2$) for 30 min at RT.
6. Wash **3 times** with 1 mL of TNT Buffer for **1 hr** each at RT.
7. Wash sample once with **PNT-500** (1x PBS + 0.1% Triton X-100 + 500 mM NaCl) for 30 min at RT.
8. Wash sample once with **1x PBS** for 30 min at RT.

### Cleave DATD-crosslinked Gels #1

1. Transfer sample into a 6-well plate with a flat-tip paint brush.
2. Make DATD-cleaving solution from freshly prepared sodium periodate stock solution, as described in the Materials section.
3. Incubate sample in 2 mL of DATD-cleaving solution (or volumes sufficient to cover the entire gel) at RT for exactly 30 min with gentle shaking. Longer incubation period can degrade the structural integrity of the bis-crosslinked Gel #2, which we aim to preserve here.
4. Wash sample with 5 mL of 1x PBS for 20 min at RT with gentle shaking.
5. Repeat Step 4 for 5 times (6 washes total), and then leave in 5 mL of 1x PBS at RT overnight with gentle shaking. It is crucial to completely wash away sodium periodate from the gels, since residual amounts of periodate ion can disintegrate the bis-crosslinked Gel #2 over a time scale of ~2–24 hr.

### NHS-ester Staining (optional, for morphology readout)

1. Incubate sample with 5 µM of the NHS-ester of a fluorescent dye in 1 mL of PBST, overnight at RT with gentle shaking.
2. Wash sample with 1 mL of PBST for three times, 30 min each at RT.

### Expansion and Imaging

1. Transfer sample to a container suitable for imaging, e.g. a glass-bottom 6-well plate. For high-magnification objectives with limited working distance, ensure that the gel is flipped to the same orientation as when it was casted during the initial gelation, so that the animals will be located at the bottom surface of the gel.
2. Wash sample with 2 mL of 0.1x PBS for 30 min at RT.
3. Wash sample with 2 mL of deionized water for 30 min at RT.
4. Repeat 10 for one more time (two washes with deionized water total).
5. Remove as much liquid around the expanded gel as possible.
6. Perform imaging.
7. To store the sample, re-hydrate the sample in 2 mL of 1x PBS and incubate at 4°C after imaging. Due to the covalent linkage between the antibody and the hydrogel, performed in the 'Antibody Anchoring' section above, fluorescent signal in the sample is stable for at least 1 week.

### Agar Immobilization (Optional)

If too much sample vibration or drifting is observed under a high-resolution objective (e. g. >= 40x) that affects image quality, immobilize expanded gel by adding droplets of lukewarm 2%(w/w) low-melt agarose solution to the edges of the expanded gel (after water is completely removed from the well, leaving only the gel at the center of the well). After ~2 min, slowly add more low-melt agarose solution to cover the entire gel. Wait for ~2 more minutes for the agarose to solidify, and proceed to imaging.

## Iterative Expansion of *C. elegans* (iExCel) Protocol

The iterative expansion protocol described here only supports readout of fluorescent proteins, i.e. same as the standard ExCel protocol shown above. This protocol is not currently not compatible with RNA readout. The sensitivity to read out target signal is lower than the standard ExCel protocol (due to a greater volumetric dilution associated to the expansion process), and therefore it works best on fluorescent proteins that are strongly expressed (ideally, with clearly observable signal at the pre-expansion state, when imaged under a high NA (>~0.8) objective under typical illumination settings on a confocal microscope).

A number of oligo-nucleotides are used in this protocol. They serve several purposes: (1) acting as spatial anchors for the immunostained locations, (2) signal transfer between gels, and (3) signal amplification and fluorescent readout. Their sequences and ordering information can be found in 'Oligo-nucleotides in iExCel protocol'.

## Synthesis of DNA-conjugated secondary antibody

Perform the protocol shown in 'Protocol for synthesizing DNA-conjugated secondary antibody for iExCel'.

## Fixation and Cuticle Reduction

Same as in the standard ExCel protocol.

## Protein Anchoring

Same as in the standard ExCel protocol.

## Gelation (Gel #1)

Same as in the standard ExCel protocol. In addition, to ensure that the final-state expanded worm could be captured within the working distance of a high-NA lens, perform one of the two methods for tracking hydrogel orientation (see below).

Since the hydrogel sample will become ~20x larger in the z-dimension at the end of this protocol, if it is desired to image with a short-working-distance objective (e.g. <1 mm; most high NA and magnification objectives are in this category) at the fully expanded state, it is necessary to ensure that the gel is flipped back to the same side as the Gel#1 casted state, at two time points: (1) when the gel is re-embedded into Gel #2, and (2) before the expansion process prior to the final imaging. This way, the worms (which settle to the lower ~50 µm during Gel #1 gelation, out of the full ~180 µm thickness of the gel) will remain at the bottom surface of the gel at the end of this protocol, ensuring maximal coverage of the worm tissue by the working distance of the objective during the final imaging. In order to provide a way to check the orientation (side) of the gel throughout the protocol, at least one of the two additional procedures is necessary at the end of the Gelation step. See 'Methods to track the surface of the hydrogel on which the worms are located' for a protocol of these procedures.

## Digestion

Same as in the standard ExCel protocol.

## Antibody Staining (for fluorescent protein readout)

As a reference, we have successfully used the following antibodies and obtained results with good signal-to-noise ratios:

  a. Anti-GFP: Abcam ab13970, a chicken polyclonal, at 10 µg/mL
  b. Anti-mCherry: Kerafast EMU106, a rabbit polyclonal, at 1:500 dilution
  c. Anti-tagRFP: Thermo R10367, a rabbit polyclonal, at 10 µg/mL

1. Pre-incubate sample in 1 mL of 5x SSCT for 30 min at RT.
2. Incubate sample with primary antibody against fluorescent protein, at desired concentration (a good starting point is 10 µg/mL) in 500 µL of 5x SSCT overnight at 4°C.
3. Wash sample with 1 mL of 5x SSCT for 3 times, 1 hr each at RT.
4. Incubate sample with secondary antibody conjugated to an acrydite-bearing DNA oligo (synthesis instruction is shown in the earlier section 'Synthesis of DNA-conjugated secondary antibody'), at desired concentration (a good starting point is 10 µg/mL) in 500 µL of DNA-conjugated Antibody Staining Buffer overnight at 4°C.
5. Wash sample with 1 mL of 5x SSCT for 3 times, 1 hr each at RT.

## Expansion and Re-embedding into Non-Expanding Gel #2

Same as in the standard ExCel protocol, in sub-section '*Part A: Maintain expansion factor of the gelled sample with re-embedding*' of the section 'ExFISH-HCR', except for two points:

a. Prior to the re-embedding protocol, flip the gels to the same side as when it was casted in Gel#1 gelation, to ensure that the worms are at the bottom of the gel.
b. For re-embedding, replace the Re-embedding monomer solution with iExCel G2 monomer solution.

After Gel#2 gelation, it is recommended to trim the gels, to avoid samples too large to handle and image at the fully expanded final state. Trimming can be facilitated by observing where worms are located inside the gel, with naked eye, or on an upright microscope, if it is desired to not cut apart certain animals. Ideally, gels are trimmed into a dimension less than ~6 mm on each side, and have at least several worms (>=3) or regions of interest in the trimmed gel. Also, not all trimmed gels need to proceed with the protocol (which might be laborious to handle), and can be saved at this point, in 5x SSC at 4°C for at least 2 months. Pick at most ~6 trimmed gels to proceed.

## Linker Hybridization (to amplify and transfer signals from Gel#1 to Gel#3)

a. Transfer trimmed gels into a 24-well plate, if not already.
b. Pre-incubate each trimmed gel in 1 mL of iExCel hybridization buffer for 30 min at RT.
c. Incubate gel with Post-Gel#2 acrydite-bearing linkers (refer to the second entry of the read-out systems shown in 'Oligo-nucleotides used in the iExCel Protocol') corresponding to the oligo sequence on the secondary antibody applied in the section 'Antibody Staining', at the concentration of 100 nM, in 1 mL of iExCel hybridization buffer overnight at RT with gentle shaking.
d. Wash gel with 1 mL of iExCel hybridization buffer for four times, for 1 hr each for the first three washes and overnight for the final wash, at RT with gentle shaking.
e. Wash gel with 1 mL of 1x PBS for three times, 1 hr each at RT.

## Re-embedding into Expanding Gel #3

1. Flip each trimmed gel to the same side as when they were casted during Gel#1 and #2 gelation. This ensures that the worms remain on the bottom surface of the gels, so the working distance of the objective can maximally cover the depth of the worm at the final imaging stage.
2. Incubate each trimmed gel in 1 mL of Activated iExCel G3 Monomer Solution for 50 min at 4°C.
3. During the incubation, construct a gelling chamber with the same architecture as in Gel#1 (Step 6 in Section 'Gelation') but with 3x thicker spacer (i.e. the same thickness as used in Gel #2; e.g. if a #1.5 coverslip was used for Gel #1, use a stack of 3 here for each spacer). Do not close the chamber with the top coverslip yet.
4. Immediately after 50 min of incubation, transfer 105 µL of the monomer solution from the well to the center of the gelation chamber. Remove bubbles if there is any. Then, transfer the gel from the 24-well plate into the monomer solution droplet on the gelling chamber, using a flat-tip paint brush (see recommended brush in Materials; Utrecht 09013–1006). Be sure that the gel is on the side where the worms are at the bottom surface of the gel, as confirmed in Step 1. (i.e. If the gel is already correctly sided in the 24-well plate, do not flip the gel during this transfer.) Slowly place a rectangular top cover slip on the gel, to enclose the gelling chamber in the same style as perform in previous rounds of gelation, avoiding any bubbles. If the distance between the gel to the border of the gelation solution is less than ~2 mm in any direction, add more monomer solution to the chamber until solution border is sufficiently away from the gel. Alternatively, fill the entire chamber with monomer solution. (As explained previously, ambient oxygen inhibits gelation, and so the gel right at the solution-air interface is not well-formed.)
5. Incubate the gelling chamber at 37°C for 2 hr, in a humidified chamber.

6. Remove top cover glass. There should be a visible interface around the re-embedded gel. Trim away the excessive gels around the re-embedded gel.
7. Transfer trimmed gels into a 6-well plate. Wash 3 times with 5x SSC, 10 min each.

### Cleave DATD-crosslinked Gels #1 and #2

1. Make DATD-cleaving solution from the sodium periodate powder, described in the Materials section.
2. Incubate trimmed gels in 3 mL of DATD-cleaving solution (or volumes sufficient to cover the entire gel) at RT for 30 min with gentle shaking.
3. Wash gels with 5 mL of 1x PBS at RT for 15 min with gentle shaking.
4. Repeat Step 3 for seven times (eight washes total), and then leave in 5 mL of 1x PBS at RT overnight with gentle shaking. It is crucial to completely wash away sodium periodate from the gels, since residual amounts of periodate ion will also disintegrate Gel #3 over a time scale of ~2–24 hr.

### LNA Hybridization (to attach fluorophores to the stained locations for final readout)

1. Pre-incubate each trimmed gel in 1 mL of iExCel hybridization buffer for 30 min at RT.
2. Incubate gel with fluorophore-conjugated LNA oligos corresponding to the DNA linker applied in the section 'Linker Hybridization', at the concentration of 100 nM, in 1 mL of iExCel hybridization buffer overnight at RT with gentle shaking.
3. Wash gel with 1 mL of iExCel hybridization buffer for four times, for 1 hr each for the first three washes and overnight for the final wash, at RT with gentle shaking.
4. Wash gel with 1 mL of 5x SSC for 3 times, 1 hr each at RT.

### Expansion and Imaging

1. Transfer sample to a container suitable for imaging, e.g. a glass-bottom 6-well plate. For high-magnification objectives with limited working distance, ensure that the gel is flipped to the same side as when it was first casted in Gel #1, so that the animals will be at the bottom of the gel.
2. Wash sample with 5 mL of deionized water three times, for 1 hr each at RT.
3. Remove as much liquid around the expanded gel as possible.
4. Perform imaging.

### Agar Immobilization (Optional)

If too much sample vibration or drifting is observed under a high-resolution objective (e. g. >= 40x) that affects image quality, immobilize expanded gel by adding droplets of lukewarm 2%(w/w) low-melt agarose solution to the edges of the expanded gel (after water is completely removed from the well, leaving only the gel at the center of the well). After ~2 min, slowly add more low-melt agarose solution to cover the entire gel. Wait for ~2 more minutes for the agarose to solidify, and proceed to imaging.

### Materials

#### Fixation

- PFA Fixative (10 mL)
    - 1 mL 10x PBS (final 1x)
    - 2.5 mL 16% paraformaldehyde (final 4%)
    - Add water to 10 mL
- 40x Borate Buffer Stock (50 mL)
    - 3.1 g Boric acid (final 1 M)
    - 1.0 g NaOH (final 0.5 M)
    - Add water to 50 mL
- BT (40 mL)
    - 1 mL 40x Borate Buffer Stock (final 1x)
    - 1 mL 20% (v/v) Triton X-100 (final 0.5%)
    - 38 mL ddH2O
- BTB (40 mL; a.k.a. cuticle reduction buffer)

- ○ 1 mL 40x Borate Buffer Stock (final 1x)
- ○ 1 mL 20% (v/v) Triton X-100 (final 0.5%)
- ○ 0.8 mL 2-mercaptoethanol (final 2%)
- ○ 37.2 mL ddH2O
- PBST-0.5%
  - ○ 1x PBS
  - ○ 0.5% (v/v) Triton X-100

## RNA and Protein Anchoring

- AcX Stock (as prepared in ProExM paper)
  - ○ 10 mg/mL AcX (Acryloyl-X, SE; re-suspended in anhydrous DMSO)
  - ○ Store dessicated at −20°C until needed
- LabelX Stock (as prepared in ExFISH paper)
  - ○ 100 μL of 1 mg/mL Label-IT Amine Modifying Reagent (re-suspended in provided Mirus Reconstitution Solution)
  - ○ 10 μL of AcX Stock
  - ○ Incubate overnight at RT; then store at −20°C until needed
- RNA Anchoring Buffer
  - ○ 20 mM MOPS pH 7.7
  - ○ 0.1% (v/v) Triton X-100
- Protein Anchoring Buffer (also referred as MBST pH 6.0)
  - ○ 100 mM MES pH 6.0
  - ○ 150 mM NaCl
  - ○ 0.1% (v/v) Triton X-100
- PBST-0.1%
  - ○ 1x PBS
  - ○ 0.1% (v/v) Triton X-100
- MOPST-0.1%
  - ○ 100 mM MOPS pH 7.0
  - ○ 150 mM NaCl
  - ○ 0.1% (v/v) Triton X-100

## Gelation and Expansion

- Monomer Solution Stock (47 mL; stored at −20°C for up to 6 months; recommended to split into 1 mL single-use aliquots to avoid repeated freeze-thaw cycles)
  - ○ 11.36 mL 33% (w/w) sodium acrylate
  - ○ 2.5 mL 50% (w/w) acrylamide
  - ○ 5 mL 5% (w/w) N,N'-diallyl-tartardiamide (DATD crosslinker)
  - ○ 20 mL 5M NaCl
  - ○ 2.5 mL 1M MOPS pH 7.0
  - ○ Add ddH2O to 47 mL
- Non-activated Monomer Solution (1 mL)
  - ○ 925 μL Monomer Solution Stock
  - ○ 5 μL 20% (v/v) Triton X-100
  - ○ 70 μL ddH2O
- Activated Monomer Solution (1 mL)
  - ○ 925 μL Monomer Solution Stock
  - ○ 5 μL 20% (v/v) Triton X-100
  - ○ 30 μL 0.5% (w/w) 4-Hydroxy-TEMPO (4-HT)
  - ○ 20 μL 10% (w/w) N,N,N',N'-Tetramethylethylenediamine (TEMED)
  - ○ 20 μL 10% (w/w) Ammonium persulfate (APS; activator; add last)
- Non-expanding Digestion Buffer (add Proteinase K right before use)
  - ○ 50 mM Tris pH 8.0
  - ○ 0.5 M NaCl
  - ○ 40 mM CaCl$_2$
  - ○ 0.1% (v/v) Triton X-100

- TNT Buffer
  - 50 mM Tris pH 8.0
  - 1 M NaCl
  - 0.1% (v/v) Triton X-100

## ExFISH-HCR and Antibody Staining

- Re-embedding Monomer Solution (10 mL)

  - 1 mL 40% Acrylamide/Bis 19:1 40% (w/v) solution
  - 50 µL 1M Tris base
  - 8.85 mL ddH2O
  - 50 µL 10% (w/w) N,N,N',N'-Tetramethylethylenediamine (TEMED)
  - 50 µL 10% (w/w) Ammonium persulfate (APS; activator; add last)
- ExFISH Wash Buffer
  - 2x SSC
  - 20% (v/v) formamide
- Hybridization Buffer
  - 2x SSC
  - 20% (v/v) formamide
  - 10% (w/v) dextran sulfate
- Amplification Buffer
  - 5x SSC
  - 0.1% (v/v) Tween 20
  - 10% (w/v) dextran sulfate
- 5x SSCT

  - 5x SSC
  - 0.1% (v/v) Tween 20

## Epitope-Preserving Expansion of *C. elegans*

- Collagenase VII Stock Buffer (50 mL)

  - 2.5 mL 1M Tris pH 8.0
  - 1.5 mL 5M NaCl
  - 50 µL 1M $CaCl_2$
  - Add ddH2O to 50 mL
- Collagenase VII Dilution Buffer (50 mL)

  - 7.5 mL 1M Tris pH 8.0
  - 8.5 mL 5M NaCl
  - 4 mL 1M $CaCl_2$
  - Add ddH2O to 50 mL
- Collagenase VII Stock Solution
  - Collagenase VII from Millipore Sigma (C0773; we typically get >= 15 kU)
  - Add Collagenase VII Stock Buffer to a final concentration of 1 kU/mL
  - Aliquot into 500 µL portions, and store at −20˚C. Each aliquot is intended for a single use. Do not re-freeze after thawing.
- TNC-40020 Buffer (50 mL)

  - 2.5 mL 1M Tris pH 8.0
  - 4 mL 5M NaCl
  - 1 mL 1M $CaCl_2$
  - 50 mL
  - Protein Denaturation Buffer (50 mL; add reagents in the following order, to prevent SDS precipitation)
  - 25 mL ddH2O
  - 2.5 mL 1M Tris base
  - 4 mL 5M NaCl
  - 1 mL 1M $CaCl_2$
  - Adjust pH to 9.0 with 5M HCl
  - Add ddH2O to 35.7 mL
  - Vortex the solution to mix, before SDS addition
  - 14.3 mL 20% (w/w) sodium dodecyl sulfate (SDS)

- ○ Vortex to mix, and store solution at 37℃. Prolonged storage (>1 hr) at RT causes SDS to precipitate, which can be reversed by 37℃ incubation.
- Epitope-preserving ExCel G2 Monomer Solution Stock (47 mL; stored at −20℃ for up to 6 months; recommended to split into 1 mL single-use aliquots to avoid repeated freeze-thaw cycles)

  - ○ 11.36 mL 33% (w/w) sodium acrylate
  - ○ 2.5 mL 50% (w/w) acrylamide
  - ○ 3.75 mL 2% (w/w) N,N'-Methylenebisacrylamide (Bis crosslinker)
  - ○ 2.5 mL 1M MOPS pH 7.0
  - ○ 1.5 mL 5M NaCl
  - ○ 2 mL 1M CaCl$_2$
  - ○ Add ddH2O to 47 mL
- Non-activated G2 Monomer Solution (1 mL)

  - ○ 930 μL Epitope-preserving ExCel G2 Monomer Solution Stock
  - ○ 70 μL ddH2O
- Activated G2 Monomer Solution (1 mL)

  - ○ 930 μL Epitope-preserving ExCel G2 Monomer Solution Stock
  - ○ 30 μL 0.5% (w/w) 4-Hydroxy-TEMPO (4-HT)
  - ○ 20 μL 10% (w/w) N,N,N',N'-Tetramethylethylenediamine (TEMED)
  - ○ 20 μL 10% (w/w) Ammonium persulfate (APS; activator; add last)
- DATD-cleaving solution ( 5 mL; make the 500 mM sodium meta-periodate solution right before use; do not use if powder has been dissolved for more than 3 hr)

  - ○ 200 μL of 500 mM sodium meta-periodate
    - ▪ We note that recent batches of sodium meta-periodate from Thermo Fisher do not have consistent solubility, and this 500 mM stock solution could either look completely clear or moderately cloudy. Even in the case of cloudy stock solution, similar outcome was obtained as long as the stock solution is well mixed (via vortexing) before its addition into the final cleaving solution, as it eventually dissolves at a final concentration of 20 mM.
  - ○ 500 μL of 10x PBS
  - ○ 4.5 mL
  - ○ Adjust pH to 5.5 with 5M HCl
  - ○ 5.0 mL

## Iterative Expansion of *C. elegans* (iExCel)

- ○ DNA-conjugated Antibody Staining Buffer (20 mL; stable for up to 2 months if stored at 4℃)

  - ▪ 4 g of 10% (w/v) Dextran sulfate
  - ▪ 2 mL of 20x SSC
  - ▪ 1 mL of 20 mg/mL Baker's yeast tRNA
  - ▪ 1 mL of 100% Normal donkey serum
  - ▪ 19.9 mL
  - ▪ 100 μL of 20% Triton X-100
- ○ iExCel G2 Monomer Solution (10 mL; the components without TEMED and APS can be pre-mixed and is stable for up to 3 months if stored at 4℃)

  - ▪ 2 mL of 50% (w/w) acrylamide
  - ▪ 1 mL of 5% (w/w) N,N'-diallyl-tartardiamide (DATD crosslinker)
  - ▪ 6.9 mL of ddH2O
  - ▪ 50 μL 10% (w/w) N,N,N',N'-Tetramethylethylenediamine (TEMED)
  - ▪ 50 μL 10% (w/w) Ammonium persulfate (APS; activator; add last)
- ○ iExCel hybridization buffer
  - ▪ 4x SSC
  - ▪ 20% (v/v) formamide
- ○ iExCel G3 Monomer Solution Stock (47 mL; stable for up to 6 months if stored at −20℃; recommended to split into 1 mL single-use aliquots to avoid repeated freeze-thaw cycles)

  - ▪ 11.36 mL 33% (w/w) sodium acrylate

- 2.5 mL 50% (w/w) acrylamide
- 3.75 mL 2% (w/w) N,N'-Methylenebisacrylamide (Bis crosslinker)
- 20 mL 5M NaCl
- 5 mL 10x PBS
- Add ddH2O to 47 mL
  - Activated iExCel G3 Monomer Solution (1 mL)

    - 930 µL Bis-crosslinked Expanding G3 Monomer Solution
    - 30 µL 0.5% (w/w) 4-Hydroxy-TEMPO (4-HT)
    - 20 µL 10% (w/w) N,N,N',N'-Tetramethylethylenediamine (TEMED)
    - 20 µL 10% (w/w) Ammonium persulfate (APS; activator; add last)
  - DATD-cleaving solution (5 mL; make the 500 mM sodium meta-periodate solution right before use; do not use if powder has been dissolved for more than 3 hr)

    - 200 µL of 500 mM sodium meta-periodate
    - We note that recent batches of sodium meta-periodate from Thermo Fisher do not have consistent solubility, and this 500 mM stock solution could either look completely clear or moderately cloudy. Even in the case of cloudy stock solution, similar outcome was obtained as long as the stock solution is well mixed (via vortexing) before its addition into the final cleaving solution, as it eventually dissolves at a final concentration of 20 mM.
    - 500 µL of 10x PBS
    - Add ddH2O to 4.5 mL
    - Adjust pH to 5.5 with 5M HCl
    - Add ddH2O to 5.0 mL
  - Buffer A

    - 150 mM NaCl
    - 100 mM Na2HPO4
    - pH to 7.4
  - Buffer C

    - 150 mM NaCl
    - 100 mM Na2HPO4
    - pH to 6.0

| Vendor | Product Name | Catalog Number |
| --- | --- | --- |
| Various (see 'Notes on sodium acrylate quality') | Sodium acrylate | See 'Notes on sodium acrylate quality' |
| Electron Microscopy Sciences | Formaldehyde, 37% | 15686 |
| Millipore Sigma | 2-Mercaptoethanol | M6250 |
| | Triton X-100 | X100 |
| | MES sodium salt | M3058-25G |
| | Acrylamide | A9099 |
| | N,N'-Methylenebisacrylamide (Bis crosslinker) | M7279 |
| | Ammonium Persulfate | A3678 |
| | N,N,N',N'-Tetramethylethylenediamine | T7024 |
| | 4-Hydroxy-TEMPO | 176141 |
| | Atto 647N NHS ester | 18373 |
| | tRNA from bakers yeast | 10109495001 |
| | 2-Hydrazinopyridine dihydrochloride | H17104-10G |
| Thermo Fisher Scientific | Tris, pH 8 | AM9855 |
| | 20X SSC | 15557–044 |
| | AcX (Acryloyl-X, SE) | A20770 |
| | Sodium meta-periodate | 20504 |
| | Acrylamide/Bis 19:1, 40% (w/v) solution | AM9022 |

*continued*

| Vendor | Product Name | Catalog Number |
|---|---|---|
| | Formamide | AM9342 |
| | Alexa Fluor 546 NHS Ester (Succinimidyl Ester) | A20002 |
| | Alexa Fluor 647 NHS Ester (Succinimidyl Ester) | A37573 |
| | Zeba Spin Desalting Columns | 87766 |
| | DMSO, anhydrous | D12345 |
| Millipore | Dextran sulfate | S4030 |
| New England Biolabs | Proteinase K | P8107S |
| Mirus Bio LLC | Label IT Nucleic Acid Modifying Reagent | MIR 3900 |
| Thermo Scientific | Thermo Scientific Nunc Rectangular Dishes | 267061 |
| Utrecht | Blick Masterstroke Golden Taklon Brushes, Shader, Size 6 | 09013–1006 |
| Alfa Aesar | N,N'-Diallyl-L-tartardiamide (DATD crosslinker) | A12195 |
| Jackson Immuno Research | Normal donkey serum | 017-000-121 |
| Solulink | 10x Turbolink Catalyst Buffer | S-2006–105 |
| | Sulfo S-4FB Crosslinker | S-1008–105 |
| | S-HyNic Crosslinker | S-1002–105 |
| Amicon | Ultra-0.5mL Centrifugal Filters | UFC510024 |
| Vivaproducts | Vivaspin 500 Centrifugal Concentrator | VS0101 |

## Notes on sodium acrylate quality

During the execution of experiments in this study, we switched the vendor of sodium acrylate, a key ingredient of the monomer solutions, from Millipore Sigma (Product number 408220) to Santa Cruz Biotechnologies (Product number sc-236893B), because we noticed that the quality of the more recent lots from Millipore Sigma (roughly from 2019/06, to the time of publication) differed significantly from that of the previous lots (roughly before 2019/06), in several aspects: the powder texture (which differed distinctly in the fluffiness), solubility (e.g. powder no longer dissolves at the stock concentration of 33%, and instead left insoluble, string-like precipitates, which might indicate premature polymerization within the source), and color of solution (high quality, near-complete clearness vs. poor quality, moderately yellow). Since we worried that the inability to fully dissolve the sodium acrylate would affect experimental results, we switched to another vendor, Santa Cruz Biotechnologies, whose sodium acrylate lots yielded completely dissolved stock solutions, at the time of our experiments (ordered at around 2019/06). However, we later received reports that some of the more recent lots from Santa Cruz Biotechnologies also had quality issues, similar to the ones associated with the Millipore Sigma lots.

Thus, we screened through sodium acrylates from various vendors, and compiled our results below:

| Vendor for sodium acrylate | Catalog Number | Reports of insoluble lots |
|---|---|---|
| Combi-Blocks | QC-1489 | None at the time of publication |
| AK Scientific | R624 | None at the time of publication |
| BLDpharm | BD151354 | None at the time of publication |
| Santa Cruz Biotechnology | sc-236893B | one report after ~2019/06 |
| Fisher Scientific | 50-750-9773 | one report after ~2019/11 |

*continued*

| Vendor for sodium acrylate | Catalog Number | Reports of insoluble lots |
|---|---|---|
| Millipore Sigma | 408220 | >5 reports after ~2019/06 |

We recommend the following practice for ExCel users:

1. Order sodium acrylate from multiple vendors listed above. Dissolve ~5 g of each powder to make a 33% (w/w) stock solution. Reject powders that do not fully dissolve after ~1–2 min of vortexing. Among the stock solutions in which the sodium acrylate fully dissolves, select the one that appears the least yellow (and the most colorless).
2. Immediately re-order multiple bottles of sodium acrylate from the same vendor, with the same lot number. Store the quality-screened lots of sodium acrylate powder at −20℃ in anhydrous conditions, which should remain stable for at least 6-12 months.
3. Never store 33% (w/w) stock solutions of sodium acrylate. Always use freshly made stock to produce a large batch of the Monomer Solution Stock (we typically make ~50 mL per batch), which can be in turn stored at -20℃ as 1 mL aliquots for 6 months, as the reduced final concentration of sodium acrylate (7.5%(w/w)) is more stable for storage.

We have not investigated how the low-quality sodium acrylate affects the outcomes of expansion microscopy (e.g. isotropy, signal retention, etc.), but we would advise against using them (especially ones that do not fully dissolve into the 33% (w/w) stock solution), because the effective concentration might not meet the specified concentration, which could affect the expansion factor, among other properties of the hydrogel.

## Oligo-nucleotides used in the iExCel protocol

For readout of a single fluorescent protein target, use only the B1-B2 system. For readout of two targets, additionally use the A2-A1 system.

Order DNA oligos from IDT (https://www.idtdna.com/site/order/oligoentry). Order LNA oligos (at least 250 nmol synthesis scale) from Qiagen (https://www.qiagen.com/us/shop/genes-and-pathways/custom-products/custom-oligo-nucleotide-designer-new/). Based on our experience, estimated delivery will be ~1 week and ~3 weeks for these two companies, respectively.

| DNA oligo name | Purpose | Sequence | Required modification | Purification | Recommended synthesis scale |
|---|---|---|---|---|---|
| B1-B2 system (For 1-color readout) | | | | | |
| 5'Ac-AA-B1-AA-3'Amine | Conjugation to secondary antibody | /5ACryd/AAG TTC GGA TTC TTA GGG CG T AAA/ 3AmMO/ | 5'acrydite 3'amine | Standard desalting | one $\mu$mole |
| 5'Ac-B1'-4xB2' | Post-Gel#2 Linker to transfer signal location between gels; amplifies signal by 4x via branched DNA scheme | /5ACryd/AT ACG CCC TAA GAA TCC GAA ATA GCA TTA CAG TCC TCA TAA TAG CAT TAC AGT CCT CAT AAT AGC ATT ACA GTC CTC ATA ATA GCA TTA CAG TCC TCA TA | 5'acrydite | PAGE | one $\mu$mole |

*continued*

| DNA oligo name | Purpose | Sequence | Required modification | Purification | Recommended synthesis scale |
|---|---|---|---|---|---|
| LNA_B2-Atto647N | Post-Gel#3 final readout | T+GA +G+G +G C+T+G +TA+A +T+GC/3AT-TO647NN/ (+ denotes that the next base is a locked nucleic acid) | 3' Atto 647N | HPLC | one $\mu$mole |
| A2-A1 system (For 2-color readout) | | | | | |
| 5'Ac-AA-A2-AA-3'Amine | Conjugation to secondary antibody | /5ACryd/AAA GAT TGA GAT GCC TGT CAC CAA/ 3AmMO/ | 5'acrydite 3'amine | Standard desalting | one $\mu$mole |
| 5'Ac-A2'-4xA1' | Post-Gel#2 Linker to transfer signal location between gels; amplifies signal by 4x via branched DNA scheme | /5ACryd/GGT GAC AGG CA T CTC AA TCT A TT ACA AAG CAT CAA CGA TTA CAA AGC ATC AAC GAT TAC AAA GCA TCA ACG ATT ACA AAG CAT CAA CG | 5'acrydite | PAGE | one $\mu$mole |
| LNA_A1-Atto647N | Post-Gel#3 final readout | C+GT +T+GA +TG+C +T+T +T G+T+A/3AT-TO565N/ (+ denotes that the next base is a locked nucleic acid) | 3' Atto 565 | HPLC | one $\mu$mole |

## Protocol for synthesizing DNA-conjugated secondary antibody for iExCel

This protocol is essentially identical to the conjugation protocol in the original ExM manuscript (available on www.expansionmicroscopy.org), except for one modification. – For the reaction between unconjugated secondary antibody and S-HyNic, 3 times of the S-HyNic concentration is used in this protocol compared to the original one. (6:100 dilution of the S-HyNic stock solution, instead of 2:100 dilution in the original protocol.)

## Part A: Prepare the DNA for conjugation

In this part, the process can be paused after any of these steps: Step 10, 12, 15, 19, 29. If pausing is desired, store the reagent at 4°C, after the specified steps are complete.

1. Order from IDT the DNA oligo '5'Ac-AA-B1-AA-3'Amine' (see 'Oligo-nucleotides used in the iExCel Protocol' for sequence) at a synthesis scale of 1 umole. If two-color readout is desired, additionally order the oligo '5'Ac-AA-A2-AA-3'Amine'. Be sure to add the specified modifications at the 5' and 3' ends. Choose standard desalting as the purification

method.

The instruction below applies to a single tube of DNA oligo from IDT. If two-color readout is desired, perform process to both tubes of oligos.

2. Add 100 µL of deionized water to the DNA oligo (shipped dry). Vortex for 1 min to dissolve. Spin with a tabletop centrifuge for 5 s to collect solution. Transfer the solution to a separate 1.5 mL tube. Keep the original tube from IDT, as its label contains DNA dry weight information that is useful later.

3. Add 100 µL of chloroform to the tube in a chemical hood. (Chlorform is volitale; handle only in the hood.) Vortex the tube to mix, and then spin down with a tabletop centrifuge to separate the aqueous and chloroform fractions from each other. (A clear line separating the two fractions should be visible.) Use a pipette to take the top fraction (i.e. the aqueous fraction, where DNA is dissolved in) and transfer to a separate 1.5 mL tube. This process removes the impurities from the shipped oligo, as they dissolve in the chloroform.

4. Repeat Step 3 for two more times (three chloroform washes total).

5. Add 10 µL of 3M NaCl to the DNA solution. Vortex and spin down, for 5 s each.

6. Add 250 µL of ice cold 100% ethanol. White precipitates (DNA) should form immediately. Do not mix the content. Directly incubate the tube at −20°C for 30 min.

7. Centrifuge the tube in a refrigerated centrifuge at 4°C, for 30 min, at 18000 g or the maximum speed of the centrifuge. A firm white pallet (the consists of the DNA) should form after this step.

8. Remove supernatant (ethanol) from pellet. Gently add 1 mL of ice cold 70% ethanol to the inner wall of the tube, without disturbing the pallet, and then remove as much supernatant as possible.

9. Uncap the tube, and let it sit at RT for ~1 hr, to briefly dry pallet.

10. After 1 hr, add Buffer A to make a DNA solution with 25 µg/µL. The amount of Buffer A needed = (DNA weight / (25 µg/µL)). The DNA weight is available on IDT data sheet, and also on the label of the shipped tube. Use a combination of vortexing and pipetting up and down to mix the content, until the pallet completely dissolves and is no longer visible.

11. To measure the DNA concentration in the tube (cannot use the information on the shipped tube, since there are losses during the purification steps), mix 1 µL of the DNA oligo stock solution with 99 µL of Buffer A in a separate 1.5 mL tube.

12. Measure the absorbance at 260 nm for the 100x diluted DNA oligo solution with Nanodrop (or any equivalent spectrometer) using Buffer A as the blank solution. Use the extinction coefficient of the DNA (shown on the IDT datasheet) to convert the A260 to DNA concentration of the diluted sample. Multiply the value by 100 to obtain the concentration for the stock tube. Multiply the stock concentration by the volume inside the stock tube, to obtain the number of nanomoles of DNA in the stock tube.

13. Calculate the amount of Sulfo-S4FB solution needed:
uL of Sulfo-S4FB to use) = (nmol of DNA) * 349 * 40 * 15/10⁶ Explanation: 349 g/mol is the the molecular weight of Sulfo-S4FB; (1 mg / 40 µL) is the concentration of Sulfo-S4FB solution. 15 is the target molar ratio between Sulfo-S4FB and DNA. 10⁶ adjust the unit to the correct order of magnitude.

14. Add 40 µL of anhydrous DMSO (Thermo D12345) to 1 mg of Sulfo-S4FB (TriLink S-1008). Vortex for at least 1 min, spin down, and check if the entire dried pallet has dissolved. If more than 40 µL of Sulfo-S4FB solution is needed, do this to multiple tubes and collect fully-dissolved solutions into a single tube.

15. Add the calculated amount of Sulfo-S4FB to the DNA oligo stock tube. Vortex to mix, and spin down to collect the liquid. Incubate the tube at RT overnight.

16. Purify DNA from un-reacted S4FB with Vivaspin 500 centrifugal filter with 5 kDa molecular weight cutoff (Vivaproducts VS0101). To do this, add Buffer C to the tube until the total volume is 1000 µL. Transfer the 500 µL each to two centrifugal filters. Centrifuge the tube at 13000 g for 10 min. Discard the flow-through (the portion that comes out from the bottom of the filter).

17. Add Buffer C into each filter until the volume is ~500 µL. Thoroughly mix the content, by pipetting up and down for ~5 times. Centrifuge the tube at 13000 g for 10 min. Discard the flow-through (the portion that comes out from the bottom of the filter).

18. Repeat Step 17 for three more times (five spins total, from Steps 16–18).

19. Transfer the concentrate from the filter to a separate 1.5 mL tube. Calculate the amount of Buffer C to add, in order to bring the total volume to 150 µL. Add this amount of Buffer C to the centrifugal filter and pipette up and down to mix (to collect residual DNA oligo

solution from the filter), and then transfer the solution to the tube. The 4FB-DNA stock solution can now be stored at 4°C for at least 2 years.

20. Weigh out 5–10 mg of 2-hydrazinopyridine·2HCl (abbreviated as 2-HP from this point on, Millipore Sigma H17104). Dissolve the powder in ultrapure $H_2O$, to a concentration of 50 mg/mL.

21. Add 91 μL of the 2-HP solution to a tube containing 50 mL of 100 mM MES Buffer, pH 5.0. This 2-HP working solution remains stable for up to 30 days at 4°C.

22. Prepare 2-HP blank solution by adding 2 μL of water to 18 μL of 2-HP working solution.

23. Prepare 4FB-DNA MSR solution by adding 2 μL of the 4FB-DNA stock (mix stock well by vortex, before use) to 18 μL of 2-HP working solution.

24. Vortex and spin down solutions made in Steps 22 and 23. Incubate solutions at 37°C for 60 min.

25. After incubation, spin down the tubes for 15 s to collect condensation to the bottom of the tube. Vortex for 5 s, and spin down once more for 5 s.

26. Measure the absorbance at 360 nm and 260 nm of the 4FB-DNA MSR solution on Nano-drop (or equivalent spectrometer) using the solution made in Step 22 as the blank solution, under the UV-Vis function of the spectrometer. Read absorbance values from the 1 mm pathlength. However, if the A260 reading is much greater than 1, read from the 0.1 mm pathlength to get an accurate reading. Make sure to use A360 and A260 that came from the same pathlength setting (either use values that both come from the 1 mm setting, or both from the 0.1 mm setting).

27. Calculate the molar substitution ratio (MSR) with the following formula:
MSR = A360/A260 * (DNA extinction coefficient/24500) The MSR should be close to 1.00 (0.90–1.20 is the acceptable range). If the MSR is greater than 1.20, repeat Steps 16–19 to further purify un-reacted S4FB from the 4FB-DNA stock solution, using new centrifugal filters, until the MSR is below 1.20.

28. Calculate the amount of 4FB-DNA stock solution needed for 100 μg of antibody with the following formula:
(uL of 4FB-DNA for100 μg of antibody)=7.5 * 100/150000 * 1000 / (MSR * [4FB-DNA stock concentration in mM]
Explanations: 7.5 is the molar ratio between 4FB-DNA and antibody. 100 μg is the antibody mass that we are calculating for. 150000 g/mol is the molecular weight of an IgG antibody. 1000 adjusts to the correct order of magnitude. 4FB-DNA concentration is calculated from the A260 reading from UV-Vis and the extinction coefficient of the DNA oligo – be sure to account for the 1:10 dilution performed in Step 23, and the pathlength setting use. For example, if the A260 is measured from the 4FB-DNA MSR solution at the 0.1 mm wavelength, multiple the A260 first by a factor of 10 (accounts for 1:10 dilution), and then by a factor of 100 (accounts for the 0.1 mm pathlength used, as the IDT extinction coefficient is given in the unit of $(M * cm)^{-1}$., where 1 cm / 0.1 mm=100. Divide this post-adjusted A260 by the extinction factor, to get the 4FB-DNA stock concentration in M. Finally, multiple by 1000 to get the quantity in mM.

29. Record the numbers from Steps 27 and 28 for each 4FB-DNA stock solution synthesized. The analyzed 4FB-DNA stock solution is ready for conjugation with antibodies, and can be stored at 4°C for at least 2 years.

## Part B: Prepare the antibody for conjugation

For this part, it is recommended to execute the entire protocol within a session (takes ~3–4 hr) without pausing.

1. Order unconjugated whole IgG secondary antibody from Jackson Immuno Research (https://www.jacksonimmuno.com/), to match the host species of the primary antibody against the fluorescent protein. As a reference, we have successfully used the following primary antibodies against fluorescent proteins. Hence, order secondary antibodies against chicken and/or rabbit, if you plan to also use these antibodies.

   a. Anti-GFP: Abcam ab13970, a chicken polyclonal
   b. Anti-mCherry: Kerafast EMU106, a rabbit polyclonal
   c. Anti-tagRFP: Thermo R10367, a rabbit polyclonal

   The following protocol specifies quantities needed for a single conjugation between

DNA-oligo and antibody. If multiple combinations of DNA and antibody are desired, such as for the 2-color readout (e.g. anti-chicken conjugated to B1, anti-rabbit conjugated to A2), scale up by processing multiple batches in parallel.

2. Buffer exchange 100 µL of the secondary antibody into Buffer A using a Zeba spin columns with 40 kDa molecular weight cutoff (Thermo 87766). To do this, use the follow protocol (essentially the same as the manufacturer's instructions):

   a. Remove the bottom closure of the spin column, and loosen the cap (do not remove cap).
   b. Place column into a 1.5 mL tube, and centrifuge at 1500 g for 1 min.
   c. Discard the flow-through and put the column back to the 1.5 mL tube.
   d. Use a marker to place a mark on the side of the column where the compacted resin is slanted upward.
   e. Add 300 µL of the equilibration buffer (i.e. the buffer to exchange the antibody into; in this case, use Buffer A) to the column. Place the cap back to the column (again, do not cap tightly). Centrifuge at 1500 g for 1 min, with the mark facing outward from the center of the centrifuge. (This preserves the shape of the compacted resin bed across rounds of centrifugation, which yields better protein retention.).
   f. Repeat Step e for two more times (for a total of 3 Buffer A washes). For the third (final) wash, centrifuge at 1500 g for 2 min, to completely remove buffer from the resin bed.
   g. Move the column to a new 1.5 mL tube. Discard the original one.
   h. Apply the antibody solution to the column. Centrifuge at 1500 g for 2 min, again with the mark facing outward from the center of the centrifuge.
   i. Discard the column, and save the flow-through, which is the antibody now buffer-exchanged into Buffer A.

3. Check antibody concentration in mg/mL with Nanodrop (or equivalent spectrometer), using Buffer A as the blank solution. Be sure to select the setting for IgG antibody (otherwise, the reading will be off by ~30%, because the mass extinction coefficient, i.e. A280 of a 10% (w/w) solution, is 13.7 for IgG, but 10.0 for general proteins). Then, dilute the antibody to the concentration of ~1.0 mg/mL with Buffer A.

4. Add 350 µL of DMSO to 1 mg of S-HyNic. Pipette up and down for 60 s, and then vortex until the dried pallet completely dissolves. This solution is stable if store in anhydrous conditions at −20℃, for up to 2 weeks.

5. Transfer 100 µL of the ~1.0 mg/mL antibody solution into a separate 1.5 mL tube. Add 6 µL of the S-HyNic solution. Mix thoroughly by pipetting up and down for 10 s. Incubate solution at RT for 2 hr.

6. After 1 hr and 50 min of incubation (i.e. 10 min before the incubation period ends), prepare Zeba spin column using exactly the same protocol shown in Step 2a-g, except that Buffer C should be used instead of Buffer A as the equilibration buffer, in order to exchange the antibodies into Buffer C this time.

7. After 2 hr of incubation, apply the S-HyNic-reacted antibody solution to the spin column, and complete the rest of buffer exchange (same as Step 2h-i). This S-HyNic-reacted antibody solution is not stable for more than a few hours, and need to react with 4FB-DNA immediately (within 1 hr), as instructed in Steps 1–3 in the next section.

## Part C: Conjugate 4FB-DNA with S-HyNic-reacted antibody

1. Add 4FB-DNA to the S-HyNic-reacted secondary antibody, by the amount calculated in Step 28 in Part A. (If the antibody tube started from 100 µL of the ~1 mg/mL stock in Buffer A, as instructed in Step 5 in Part B, then the antibody in solution should be around 1 mg/mL * 100 µL = 100 µg. The amount calculated in Step 28 in Part A was for 100 ug of antibody.)

2. Measure the total volume of the antibody-DNA mixture. Divide the total volume by 9, and then add this amount of 10x Turbolink Catalyst Buffer to the reaction tube.

3. Mix thoroughly by pipetting up and down for 10 s. Incubate the tube at RT overnight.

4. Purify unreacted 4FB-DNA from the DNA-conjugated antibodies, by using an Amicon Ultra 0.5 mL centrifugal filter with 100 kDa molecular weight cutoff (Amicon UFC510024). To do this,

a. Transfer all content of the reaction tube into the spin column.
b. Add 1x PBS to the column until volume is at 500 µL. Thoroughly mix the content, by pipetting up and down for ~5 times. Centrifuge the filter for 5 min at 14000 g. Discard the flow-through.
c. Repeat Step b for two more times, for a total of 3 spins.
d. Transfer the concentrate from the filter to a separate 1.5 mL tube. Calculate the amount of 1x PBS needed to bring the volume to the starting volume at the ~1 mg/mL stage (100 µL). Add this amount of 1x PBS to the filter to collect residual DNA-conjugated antibodies, pipette up and down to mix, and then transfer the entire content into the 1.5 mL tube.

5. The DNA-conjugated antibody is now ready for use, and is stable at 4℃ for at least 1 year.

## Methods to track the surface of the hydrogel on which the worms are located

During the initial gelation step of any ExCel protocol, the worms, which have diameters in the range of ~10–60 µm, settle to the bottom surface of the casted gel, which has a thickness of ~180 µm (set by the #1.5 cover glass, which serves the spacer of the gelation chamber). Then, during the repeated washes throughout the ExCel protocols, the hydrogel sample could get flipped, which results in worms located on the top surface of the hydrogel. If the hydrogel is in such up-side-down orientation during a re-embedding process, or during the final imaging step, a short-working-distance objective (<1–1.5 mm; most high NA and magnification objectives are in this category) might not be able to reach part or all of the expanded animals. Thus, it is necessary to ensure that the hydrogel sample is in the correct orientation (i.e. worms are on the bottom surface), at the following time points:

a. Prior to any re-embedding step
b. Prior to the final imaging step

If the re-embedding or the final imaging step is preceded by hydrogel expansion, such orientation check needs to be performed prior to the expansion process, because if the gel is on the wrong side, expanded gels are typically too fragile to get flipped.

To track the orientation of the hydrogel (i.e. which surface of the gel the animals are located on), use either of following two procedures after Gel#1 gelation, and before the gel is lifted off from the chamber.

1. Take a low magnification image of the gelled sample **before** opening the chamber, to record the orientation of worms in the gel. Image can be based on transillumination, or fluorescence (if the fluorescent protein(s) are expressed in a way that allows delineation of the border of the worm). Later, to determine whether the gelled sample has been flipped or not, take another image of the gel, and compare such image to the initial image.
2. During gel trimming, trim the gel into a shape that makes it feasible to identify the side of the gel. For example, a clearly-trimmed 4-edge parallelogram shape (with inner angles 90 deg, 90 deg, 135 deg, 45 deg, consecutively) allows one to distinguish whether the gel has been flipped or not, because if flipped, the gel can never resume its original shape by rotation.

Both methods work well when executed properly (and are also mutually compatible; i.e. it is possible to do both on the same sample, if desired). Method (1) requires microscopy access whenever gel orientation needs to be checked, and might also require more experience and practice with microscopy (when checking the gel orientation, one has to be able to find expanded worms by transillumination and/or dim residual fluorescence.) but is easier to control gel dimension, since such method involves no constraints on the gel shape.

