## [Decision Letter]

Thank you for submitting your article "Expansion microscopy of *C. elegans*" for consideration by *eLife*. Your article has been reviewed by two peer reviewers, and the evaluation has been overseen by a Reviewing Editor and David Ron as the Senior Editor. The reviewers have opted to remain anonymous.

The reviewers have discussed the reviews with one another and the Reviewing Editor has drafted this decision to help you prepare a revised submission.

Summary:

This work builds on the method of Expansion Microscopy (ExM), a groundbreaking technique developed by the senior author (Boyden) that enables embedding and controlled expansion of biological tissue to enhance microscopic analysis by increasing spatial resolution. Boyden and colleagues had previously found that the tough outer cuticle precludes isotropic expansion of *C. elegans* larvae or adults. Given the widespread use of *Caenorhabditis elegans* as an experimental model for cell biology and neuroscience, such a technique would likely be widely useful. Here, the authors have developed a version of ExM to address this issue, which they term ExCel (Expansion of *C. elegans*). This required significant effort to overcome the barrier of the outer cuticle that surrounds *C. elegans*, which acts as a barrier to entry of macromolecules. To circumvent this obstacle, a procedure involving disulfide reduction and extended protease digestion, prior to expansion, was developed to digest the cuticle prior to expansion. The method results in successful, near-isotropic expansion by a linear factor of ~3.3, with some distortions observed around particular anatomical structures (portions of the gonad and pharynx). The authors show that this method is compatible with immunolocalization of abundant, fluorescently-tagged proteins, as well as a number of transgenic and endogenous mRNAs.

This is an important technical advance that will be of interest to the *C. elegans* research community, and may accelerate discovery. It represents a nontrivial adaptation of an existing technique. The work also introduces a novel and likely useful method for counterstaining anatomical structures in expanded worms using amine-reactive dyes. However, the reviewers' enthusiasm was tempered by the current limitations of the methodology, and they felt that the authors should explore some additional parameters before publication. In its current form, the reviewers agreed that the paper would be suitable for publication in a more specialized journal, but the reviewers were divided on its suitability for *eLife*.

Essential revisions:

The major limitation of the current procedure is that the extensive Proteinase K treatment of the intact worms (required to digest the cuticle) also destroys most internal proteins. In the current work, only fluorescent-protein tagged proteins expressed from high-copy arrays, which typically result in marked overexpression of a protease resistant fluorescent protein, were successfully localized. Although there are clearly some cases where this will be useful to other researchers, the technique would be much more widely adapted if untagged proteins and/or proteins expressed at endogenous levels could be detected, particularly since overexpression may perturb the normal localization, and genome editing methods now allow proteins to be tagged at their endogenous loci, reducing potential overexpression artifacts. While we appreciate that other researchers will likely build on and enhance the method once it is published, we are also concerned that in its current form it may prove to be of very limited use.

An obvious way to increase the sensitivity of the method would be to target the cuticle for degradation more selectively. β-mercaptoethanol has been known for many years to facilitate disruption of the disulfide-crosslinked collagen network, and the authors have exploited this in adapting their protocol. Prior work has also shown that elastase is useful to digest the outer layer, and the pharyngeal lining is sensitive to pronase; while the latter may not be markedly more selective than Proteinase K, elastase may prove useful. The authors also suggest the possibility of using chitinase, which can help to digest pharyngeal structures (Zhang et al., 2005). Thus, we feel that further optimization of the procedure should be attempted before further consideration of the paper for publication in e*Life*. If the authors feel that this is not attainable within a reasonable period of time, we welcome their comments.

---

## [Author Response]

[…] This is an important technical advance that will be of interest to the *C. elegans* research community, and may accelerate discovery. It represents a nontrivial adaptation of an existing technique. The work also introduces a novel and likely useful method for counterstaining anatomical structures in expanded worms using amine-reactive dyes. However, the reviewers' enthusiasm was tempered by the current limitations of the methodology, and they felt that the authors should explore some additional parameters before publication. In its current form, the reviewers agreed that the paper would be suitable for publication in a more specialized journal, but the reviewers were divided on its suitability for eLife.

We have now addressed the limitations of the original version of ExCel, as remarked by the reviewers, in particular its inability to detect untagged proteins, and its undemonstrated ability to detect proteins expressed at endogenous levels.

First, we include additional experimental data to demonstrate that the original version of ExCel could indeed detect proteins expressed at endogenous levels. We first constructed a transgenic strain in which the endogenous copy of *che-7* (encoding an innexin, which marks electrical synapses in *C. elegans*) is fused to a fluorescent protein (TagRFP) via CRISPR/Cas9-mediated homologous recombination (and thus at endogenous levels, as opposed to classical overexpression). We then showed that CHE-7::TagRFP could be clearly visualized after applying the original version of ExCel (Figure 7), and the spatial pattern of the post-ExCel anti-TagRFP staining corresponded well to the spatial pattern of the pre-ExCel native TagRFP signal. These results demonstrate that the standard ExCel protocol has enough sensitivity to detect proteins expressed at endogenous levels.

Second, we developed an alternative ExCel protocol that can read out untagged, completely endogenous proteins in a more general case. This protocol, which we call epitope-preserving ExCel, replaces the epitope-disruptive Proteinase K treatment essential for the original form of ExCel, with a combination of collagenase and heat-denaturation treatments, which, among the 22 treatments that we screened, yielded the optimal immunostaining results observed against a general set of antigens (Figures 11-12 and Table 1). Then, antibodies could be delivered directly into the processed worm for direct visualization of endogenous proteins. We formalized this protocol (Figure 13), and showed that although the omission of the strong 2-day Proteinase K treatment from the original form of ExCel reduced expansion isotropy (Figure 14), conclusions about qualitative spatial relationships could still be made. For example, we demonstrated that this protocol enables multiplexed imaging of endogenous proteins at nanoscale resolution, using examples including the identification of previously unreported protein localization at the interface between developing vulval precursor cells (Figure 15), and mapping of peri-active and active zones of chemical synapses (Figure 16). These results demonstrate that epitope-preserving ExCel, an alternative ExCel protocol, enables multiplexed and nanoscale-resolved imaging of untagged proteins expressed at endogenous levels, through direct antibody staining.

In order to develop the epitope-preserving ExCel protocol, we explored extensive parameters for an alternative post-gelation treatment, which would replace the strong 2-day Proteinase K treatment in the standard ExCel protocol with a gentler, epitope-preserving formula, and thus achieve greater preservation of general protein epitopes (Figures 11-12). In particular, we explored non-Proteinase-K proteases with various substrate specificity (e.g. elastase, trypsin, pepsin, papain, collagenase), treatments derived from the antigen retrieval literature (which has been described to reverse formaldehyde-mediated crosslinks introduced by fixation, and could thus potentially improve tissue expandability and epitope stainability), and treatments that were derived from earlier tissue-expansion protocols had been shown to permit a wide range of antibody stains to be applied post-treatment (Ku, 2016). We found that the post-gelation treatment that yielded optimal immunostaining results was a combination between a collagenase type VII digestion, followed by a heat-mediated denaturation treatment (Figure 12). We incorporated this combination of treatments into our final protocol for epitope-preserving ExCel (Figure 13).

Essential revisions:The major limitation of the current procedure is that the extensive Proteinase K treatment of the intact worms (required to digest the cuticle) also destroys most internal proteins. In the current work, only fluorescent-protein tagged proteins expressed from high-copy arrays, which typically result in marked overexpression of a protease resistant fluorescent protein, were successfully localized. Although there are clearly some cases where this will be useful to other researchers, the technique would be much more widely adapted if untagged proteins and/or proteins expressed at endogenous levels could be detected, particularly since overexpression may perturb the normal localization, and genome editing methods now allow proteins to be tagged at their endogenous loci, reducing potential overexpression artifacts. While we appreciate that other researchers will likely build on and enhance the method once it is published, we are also concerned that in its current form it may prove to be of very limited use.An obvious way to increase the sensitivity of the method would be to target the cuticle for degradation more selectively. β-mercaptoethanol has been known for many years to facilitate disruption of the disulfide-crosslinked collagen network, and the authors have exploited this in adapting their protocol. Prior work has also shown that elastase is useful to digest the outer layer, and the pharyngeal lining is sensitive to pronase; while the latter may not be markedly more selective than Proteinase K, elastase may prove useful. The authors also suggest the possibility of using chitinase, which can help to digest pharyngeal structures (Zhang et al., 2005). Thus, we feel that further optimization of the procedure should be attempted before further consideration of the paper for publication in ELife. If the authors feel that this is not attainable within a reasonable period of time, we welcome their comments.

Please see the text above. In particular, we have shown that untagged proteins, and proteins expressed at endogenous levels, can be directly visualized via the new, epitope-preserving ExCel protocol (Figures 11-16). In addition, we show that genome editing methods that enable proteins to be tagged at their endogenous loci can also result in proteins that can be visualized by the original form of ExCel (Figure 7).

The authors’ suggestions were very helpful – indeed, we found that screening a wide variety of proteases, in addition to other chemical treatment conditions, was very useful (Figure 11 and Table 1). In particular, we did target the cuticle for degradation more selectively, by replacing the Proteinase K treatment, which non-specifically digests proteins (and thus disrupts most epitopes, and thus precludes antibody staining of most things), with a combination of collagenase (which targets a peptide sequence that is highly specific to collagen) and heat-mediated denaturation (which does not cleave peptides, and have been shown to preserve antigenicity in other tissue-expansion protocols). With this combined treatment, the cuticle appears to be degraded by a lesser extent than by the strong 2-day Proteinase K treatment in the standard ExCel protocol, as the spatial error increases from ~1-6% to ~8-25% (compare Figure 3 to Figure 14). However, the cuticle is sufficiently permeabilized for antibody access, as we showed that the protocol can clearly visualize ~70% of protein antigens that are stainable through non-IgM class antibodies, a capability that would not be achievable with the non-specific digestion by Proteinase K.

The novel epitope-preserving ExCel protocol can clearly visualize ~70% of antigens that are detectable by non-IgM-class antibodies (Figure 12), such as acetylated tubulin (expressed in touch receptor neurons), DLG-1 (adherens junction marker), DAO-5 (a nuclear protein), UNC-10 (a pre-synaptic protein that mediates release of synaptic vesicles), and DYN-1 (dynamin, for clathrin-mediated endocytosis). In addition, we demonstrated simultaneous readout of multiple endogenous proteins, including proteins at the junctions between developing vulval precursor cells (Figure 15) and in the peri-active and active zones of chemical pre-synapses (Figure 16).